JCB Journal of Cell Biology

# Dominant-negative isoform of TDP-43 is regulated by ALS-linked RNA-binding proteins

Minami Hasegawa-Ogawa[1], Asako Onda-Ohto[1,2], Takumasa Nakajo[1], Arisa Funabashi[1], Ayane Ohya[1], Ryota Yazaki[1], and Hirotaka James Okano[1]

**TDP-43, an RNA-binding protein (RBP) encoded by the *TARDBP* gene, is crucial for understanding the pathogenesis of neurodegenerative diseases like amyotrophic lateral sclerosis (ALS) and frontotemporal lobar degeneration. Dysregulated TDP-43 causes motor neuron loss, highlighting the need for proper expression levels. Here, we identify a dominant-negative isoform among the multiple *TARDBP* splicing variants and validate its endogenous expression using a developed antibody against its translated product. Furthermore, we revealed that ALS-associated RBPs regulate its expression: hnRNP K promotes its splicing and expression, while hnRNP A1 and FUS suppress these processes through distinct mechanisms. hnRNP A1 inhibits hnRNP K–mediated splicing, and FUS represses the dominant-negative isoform through both its translational inhibition and hnRNP K suppression. Notably, ALS-mutant FUS weakens this regulatory mechanism, leading to impaired repression of hnRNP K and the dominant-negative isoform. Our findings suggest a regulatory network involving ALS-linked RBPs that govern TDP-43 isoform expression and provide new insights into how disruptions in this network contribute to ALS pathogenesis.**

## Introduction

Cellular homeostasis is tightly regulated by multiple RNA-binding proteins (RBPs), which collectively control key RNA-related processes, such as RNA splicing, stability, and translation, as well as transcription (Ravanidis et al., 2018; Zhou et al., 2014). These include ubiquitous RBPs such as heterogeneous nuclear RNPs (hnRNPs), fused in sarcoma (FUS), and TAR DNA-binding protein 43 kDa (TDP-43), as well as neuron-specific RBPs, such as neuronal ELAV-like proteins and neuro-oncological ventral antigens (NOVAs). RBPs contain RNA-binding domains, including K-homology (KH) domains and RNA-recognition motifs (RRMs), which enable them to bind target DNA/RNA and regulate its functions (Zhou et al., 2014). Aberrations in these RBPs are associated with several neurodegenerative diseases, including amyotrophic lateral sclerosis (ALS) and frontotemporal lobar degeneration (FTLD) (Kapeli et al., 2017; Krach et al., 2022; Zhou et al., 2014). Nearly 30 causative genes have been associated with (ALS), some of which encode RBPs such as hnRNP A1, hnRNP A2/B1, FUS, and TDP-43 (Akçimen et al., 2023). TDP-43 (encoded by the TAR DNA-binding protein [TARDBP] gene) is particularly critical for understanding ALS pathogenesis. TDP-43 is a multifunctional protein involved in transcription, RNA stability, and splicing

regulation of numerous binding targets (Buratti and Baralle, 2001; Buratti et al., 2001; Ou et al., 1995; Polymenidou et al., 2011; Tollervey et al., 2011a; Volkening et al., 2009). Mislocalization of TDP-43 is a hallmark of ALS pathology, observed in 97% of ALS patients (Arai et al., 2006; Ling et al., 2013; Neumann et al., 2006). Furthermore, aberrant TDP-43 function has been linked to cryptic exon (CE) inclusion in target RNAs. The incorporation of CEs often results in premature stop codons and transcript degradation or premature polyadenylation, contributing to the dysregulation of RNA processing (Ling et al., 2015). Indeed, CEs within the RNA targets of TDP-43, such as G protein signaling modulator (*GPSM2*), autophagy-related 4B cysteine peptidase (*ATG4B*), and *UNC13A*, have been detected in the frontal cortex of patients with sporadic ALS and FTLD due to TDP-43 proteinopathy (Brown et al., 2022; Ling et al., 2015; Ma et al., 2022), suggesting that the splicing activity of TDP-43 is impaired in these patients.

Both excessive and insufficient TDP-43 expression have been shown to cause the loss of motor neurons, suggesting that maintaining TDP-43 expression within a moderate range is crucial for the homeostasis of motor neurons (Igaz et al., 2011; Wu et al., 2012). TDP-43 has been reported to autoregulate its

[1]Division of Regenerative Medicine, Research Center for Medical Sciences, The Jikei University School of Medicine, Tokyo, Japan;   [2]Department of Neurology, The Jikei University School of Medicine, Tokyo, Japan.

Correspondence to Hirotaka James Okano: hjokano@jikei.ac.jp.

expression levels by modulating the proportions of canonical and truncated splicing isoforms, which in turn influence RNA stability through alternative polyadenylation (Avendaño-Vázquez et al., 2012; Ayala et al., 2011; Bembich et al., 2014; Koyama et al., 2016; Sugai et al., 2018). These splicing isoforms primarily result from alternative splicing of exon 6, the final exon (D'Alton et al., 2015; Wang et al., 2004; Wang et al., 2002). Recently, endogenous expression of certain shortened TDP-43 splicing isoforms has been detected in human cells (Shenouda et al., 2022; Weskamp et al., 2020). However, the functions of these splicing isoforms, the mechanisms underlying their alternative splicing, and their regulatory factors remain poorly understood.

In this study, we cloned several truncated TDP-43 isoforms, which were translated into proteins lacking the C-terminal glycine-rich regions but containing unique sequences at the C terminus. Interestingly, some of these shortened isoforms exhibited dominant-negative effects on the functional TDP-43 derived from the canonical splicing isoform, suggesting a novel mechanism to prevent the hyperactivity of TDP-43. Further investigation revealed that ALS-associated RBPs, hnRNP K and hnRNP A1, regulate the alternative splicing of *TARDBP*. Cytoplasmic mislocalization and nuclear depletion of hnRNP K, as well as the loss of hnRNP A1, were observed in postmortem neural tissues of ALS and FTLD patients (Bampton et al., 2021; Braems et al., 2022; Honda et al., 2015; Sidhu et al., 2022), indicating that these RBP abnormalities may contribute to TDP-43 dysregulation. Additionally, we demonstrated that pathological mutations in FUS disrupt the suppression of dominant-negative TDP-43 isoforms expression. Our work suggests that a network of RBPs regulates TDP-43, and abnormalities within any component of this regulatory network may lead to TDP-43 dysfunction, ultimately contributing to neurodegenerative diseases such as ALS and FTLD.

## Results

### One of the shortened splicing isoforms of TDP-43 exhibits dominant-negative activity

The canonical isoform of TDP-43, known as full-length TDP-43 (TDP-FL), consists of six exons. Exon 6 contains multiple splice donor and acceptor sites and generates various truncated isoforms through alternative splicing of cryptic intron 6, producing isoforms with an additional stop codon in exon 7 (Fig. 1 A). These isoforms lack the C-terminal glycine-rich domain, with the C-terminal region substituted by unique amino acids (D'Alton et al., 2015; Shenouda et al., 2022; Weskamp et al., 2020). We designated these isoforms TDP-marginal peptides (TDP-MPs). Specifically, TDP-MP13, TDP-MP18, and TDP-MP20 contain 13, 18, or 20 unique C-terminal amino acids, corresponding to the sequences FISFQMFMEEALH, VHLISNVYGRSTSLKVVL, and ILSTCFLIQEFVITHHRPRL, respectively. To investigate their function, we cloned TDP-MP18 and TDP-MP20, which have been previously characterized in earlier studies (D'Alton et al., 2015; Dammer et al., 2012; Seyfried et al., 2012; Weskamp et al., 2020). Each isoform utilizes alternative donor sites, resulting in two variants per isoform: MP18 (118), MP18 (127), MP20 (118),

and MP20 (127). These variants were cloned into N-terminal FLAG-tagged vectors, and their expression was confirmed in HEK293T cells using an anti–TDP-43 antibody (Fig. S1 A).

Overexpression of TDP-43 represses its endogenous expression via negative feedback (Ayala et al., 2011). To address the influence of MPs on endogenous TDP-43, we performed western blot (WB) analysis of HEK293T cells overexpressing C-terminal Venus-fused MPs. As previously reported, the protein levels of full-length endogenous TDP-43 (FL-endo: 43 kDa) were dramatically reduced to 43% by FL overexpression (Fig. 1 B). MP20s and MP18s also reduced FL-endo expression to approximately half and roughly 30%, respectively. For a detailed analysis of the involvement of MPs in the negative feedback system of FL-endo expression, immunocytochemical analysis was performed by overexpressing N-terminal FLAG-tagged MPs (FLAG-MPs) in HeLaS3 cells. Given that the TDP-43 antibody (G400) recognizes the C-terminal amino acid of TDP-43 that is absent in MPs, specific detection of endogenous TDP-43 was achieved. In FLAG-MPs expressing cells, endogenous TDP-43 signals were strikingly attenuated (Fig. 1 C, arrowheads) and tend to localize to the cytoplasm. Immunocytochemical analysis in HeLaS3 cells overexpressing FLAG-MPs revealed that some MP20 variants localized to both the cytoplasm and the nucleus, while TDP-FL predominantly resided in the nucleus (Fig. S1 B). No significant differences were observed between the two MP20 variants with distinct splicing donor sites, MP20 (118) and MP20 (127).

TDP-43 depletion promotes the inclusion of CEs in target RNAs like *GPSM2* and *ATG4B* (Brown et al., 2022; Klim et al., 2019; Ling et al., 2015; Ma et al., 2022; Melamed et al., 2019; Seddighi et al., 2024). To investigate the involvement of TDP-MPs in cryptic splicing, we performed RT-PCR and quantitative RT-PCR (RT-qPCR) on CEs of these targets in HEK293T cells overexpressing FLAG-MPs. Consistent with previous findings, our results confirmed that TDP-FL knockdown (FL KD) led to substantial CE inclusion in both *GPSM2* and *ATG4B* (Fig. 2 A; and Fig. S1, C and D). Interestingly, MP20s, but not FL or MP18, induced the inclusion of CEs by 70.9-fold for MP20 (118) and 42.4-fold for MP20 (127) in *GPSM2*, and by 116.3-fold for MP20 (118) and 93.9-fold for MP20 (127) in *ATG4B* (Fig. 2 A), suggesting that MP20 exerts a dominant-negative influence on the exclusion of CEs by TDP-43. Furthermore, canonical TDP-43 has been reported to regulate alternative splicing by facilitating exon exclusion in pyruvate dehydrogenase phosphatase catalytic subunit 1 (*PDP1*) and inclusion in BCL2 like 11 (*BCL2L11*) (Polymenidou et al., 2011; Tollervey et al., 2011a). Specifically, TDP-43 KD promotes the inclusion of alternative exon in *PDP1* and their exclusion in *BCL2L11*. Consistent with these findings, our RT-PCR analysis also showed that FL KD led to an increase in exon inclusion in *PDP1* and exon exclusion in *BCL2L11* (Fig. 2 A). In contrast, FL overexpression resulted in a reduction in exon inclusion in *PDP1*, as indicated by a decreased exon inclusion/exclusion ratio, while no significant changes in exon inclusion were observed for *BCL2L11*. Upon overexpressing MP20s, a significant increase in the exon inclusion/exclusion ratio for *PDP1* (~1.4-fold increase) was observed, while the ratio for *BCL2L11* was significantly decreased by ~77% for MP20s, aligning with the effects of FL KD. In contrast, MP18s had different effects:

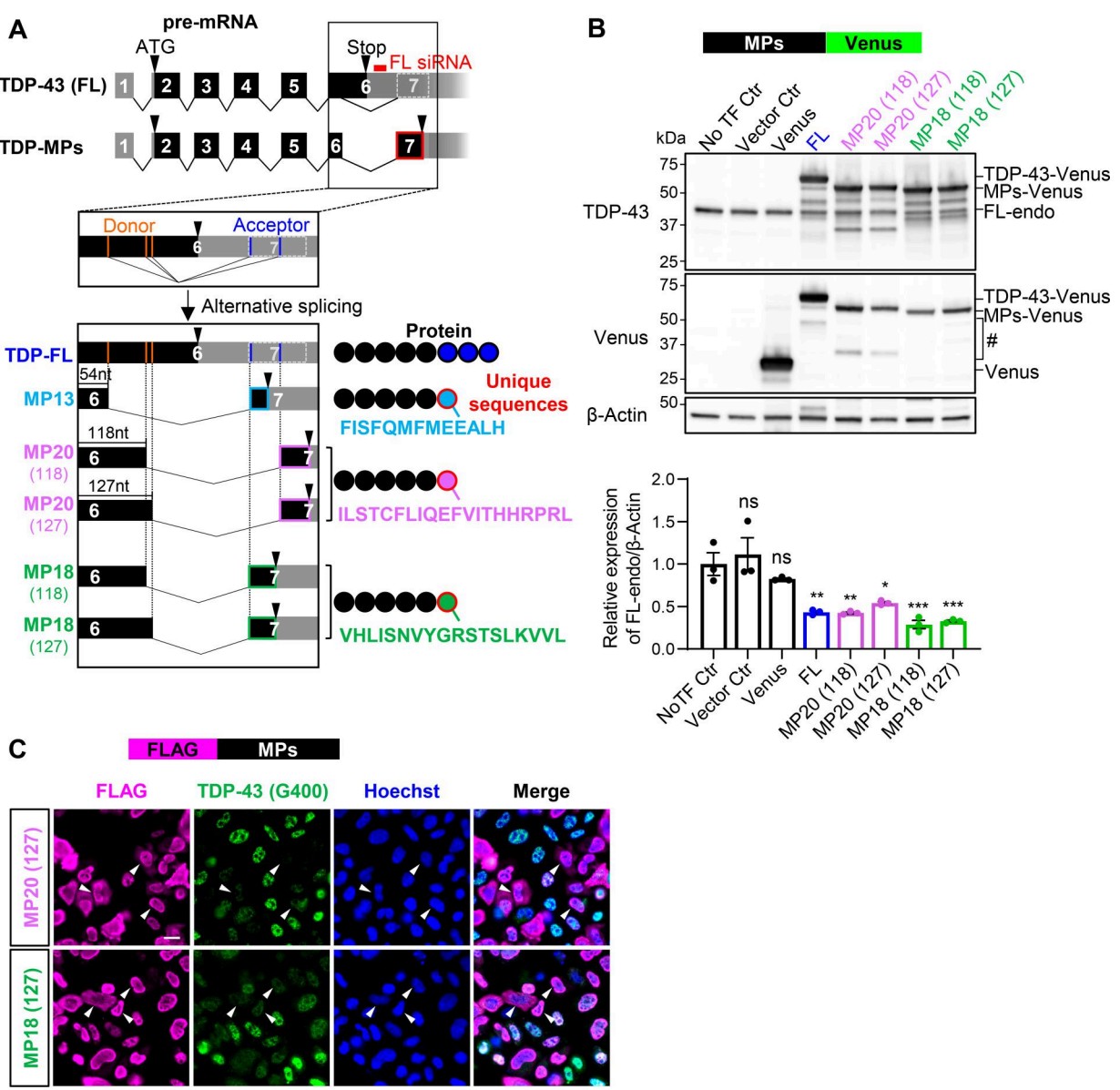

Figure 1. **Shortened TDP-43 isoforms suppress endogenous TDP-43 expression. (A)** Schematic representation of alternative splicing isoforms of *TARDBP*. Black or gray boxes indicate coding or untranslated regions, respectively. The pale gray box with a dotted outline represents the region that becomes exon 7 through alternative splicing. Orange and blue lines indicate alternative donor and acceptor sites, respectively. Alternative splicing of *TARDBP* generates TDP-MPs by excluding part of exon 6 from the TDP-FL, resulting in unique C-terminal sequences (light blue, purple, or green circle with red outlines). The red box marks the siRNA target site specific to TDP-FL. **(B)** WB analysis of HEK293T cells overexpressing TDP-MPs fused to Venus (MPs-Venus). Quantification of endogenous TDP-FL (FL-endo) levels is normalized to β-actin (*n* = 3 for each group). Controls include no transfection (NoTF Ctr) and empty vector (Vector Ctr). Data represent the mean ± SEM. Statistical significance was evaluated using one-way ANOVA followed by Dunnett's test. *P < 0.05, **P < 0.01, and ***P < 0.001 compared with the NoTF Ctr group. **(C)** Immunocytochemical images of HeLa cells overexpressing FLAG-tagged MP20 (127) or MP18 (127) (FLAG-MPs). Endogenous TDP-43 was detected using an antibody against Gly400 (G400), absent in MPs (green). FLAG (magenta) and Hoechst (blue). Arrowheads indicate FLAG-positive cells. Scale bars: 20 µm. Source data are available for this figure: SourceData F1.

MP18 (118) showed no significant impact on PDP1, whereas MP18 (127) significantly reduced the ratio by 46.8%, similar to FL overexpression. For *BCL2L11*, both MP18s also significantly decreased the exon inclusion/exclusion ratio by ~73%, aligning with the direction of FL KD. These data suggest that MP20 appears to exert a dominant-negative effect on alternative splicing targets, while MP18 appears to play a partial role in splicing regulation, similar to FL. To briefly summarize up to this point, although all MPs suppressed the protein levels of FL-endo, they

exhibited markedly distinct effects on splicing targets, notably regarding dominant-negative activity. These observations raised the possibility that differences in the RNA-binding abilities of MPs contribute to their distinct splicing effects. The RRM1 domain of TDP-43 is known to play a crucial role in nucleic acid–binding activity, and previous studies have indicated that the substitution of Phe 147 and 149 with Leu results in the loss of RNA-binding and -splicing regulatory activity of TDP-43 (Buratti and Baralle, 2001; D'Ambrogio et al., 2009). In our

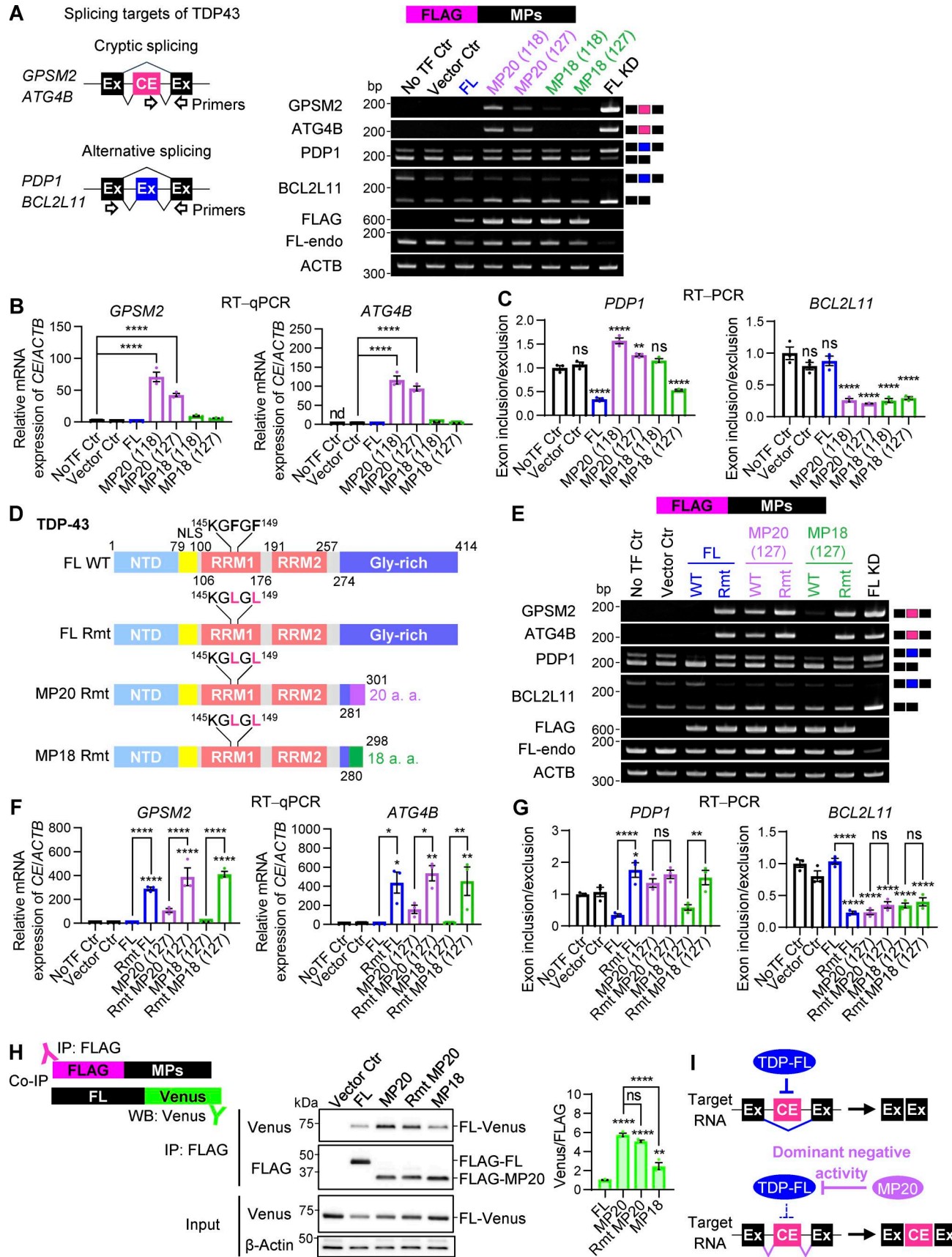

Figure 2. **Characterization of dominant-negative TDP-43 isoforms. (A)** Schematic representation of RT-PCR analysis for TDP-43–associated cryptic splicing targets (*GPSM2* and *ATG4B*) and alternative splicing events (*PDP1* and *BCL2L11*). Representative RT-PCR results from HEK293T cells overexpressing

FLAG-MPs. TDP-FL KD was used as a positive control for CE inclusion and splicing exon inclusion (*PDP1*) or exclusion (*BCL2L11*). **(B)** RT-qPCR analysis of CE inclusion in *GPSM2* and *ATG4B* transcripts in cells expressing FLAG-MPs. *ACTB* was used for normalization (*n* = 3 for each group). nd, not detected. **(C)** Ratios of exon inclusion to exon exclusion for *PDP1* and *BCL2L11* transcripts were determined via RT-PCR (*n* = 3 for each group) in cells expressing FLAG-MPs. **(D)** Schematic representation of TDP-FL, MP20, MP18, and their RRM domain mutants (Rmt). **(E)** Representative RT-PCR results of the TDP-43–associated splicing targets in HEK293T cells overexpressing FLAG-Rmt isoforms. **(F)** RT-qPCR analysis of CE inclusion in *GPSM2* and *ATG4B* transcripts in cells expressing FLAG-Rmt isoforms. *ACTB* was used for normalization (*n* = 3 for each group). **(G)** Ratios of exon inclusion to exon exclusion for *PDP1* and *BCL2L11* transcripts in cells expressing FLAG-Rmt isoforms (*n* = 3 for each group). **(H)** Co-IP analysis of FLAG-MPs and TDP-FL fused to Venus (FL-Venus) in HEK293T cells. FLAG IP was performed, and Venus levels were normalized to FLAG signals (*n* = 3 for each group). **(I)** Proposed model of dominant-negative effects exerted by TDP-MP20 on splicing regulation. In B and C, data were analyzed using one-way ANOVA followed by Dunnett's test. In F, G, and H, data were analyzed using one-way ANOVA followed by Tukey's test. Error bars represent mean ± SEM. *$P < 0.05$, **$P < 0.01$, and ****$P < 0.0001$. Gly-rich, glycine-rich region; NLS, nuclear localization signal. Source data are available for this figure: SourceData F2.

study, we introduced F147/149L mutations into the RRM1 domain of MP20 (127), as well as TDP-FL and MP18 (127) variants (Fig. 2 D), and confirmed their expression in HEK293T cells (Fig. S1 E). The F147/149L mutant of each splice variant is referred to below as RNA-binding–deficient mutant (Rmt). The overexpression of all Rmt variants of FL, MP20, and MP18 resulted in increased CE inclusion in *GPSM2* and *ATG4B*, suggesting that the suppression of CEs depends on the functional RRM domain of TDP-43 (Fig. 2, E and F). FL Rmt altered the alternative splicing patterns of *PDP1* and *BCL2L11* in the opposite direction to that observed with WT FL (Fig. 2, E and G). Similarly, MP18 Rmt altered *PDP1* splicing in a manner consistent with Rmt FL, with no significant changes in *BCL2L11* splicing compared with WT MP18. Notably, WT MP20 showed similar changes in either *PDP1* or *BCL2L11* splicing compared with MP20 Rmt. Taken together, RNA-binding mutations appear to enhance dominant-negative activity of all TDP43 variants; however, WT MP20 exerts the same level of dominant-negative activity as the RNA-binding mutants.

To further explore the differences among these isoforms, we next investigated their protein–protein interactions with FL. Previous studies have also demonstrated that TDP-FL homodimer/oligomer formation is essential for its splicing function (Afroz et al., 2017; Mompeán et al., 2017). We performed a co-immunoprecipitation (IP) assay in HEK293T cells co-overexpressing FLAG-MP20 and FL-Venus (Fig. 2 H). The degree of protein–protein interaction was quantified by the ratio of FL-Venus to FLAG in the immunoprecipitated samples. Consistent with previous reports, an interaction between FLAG-FL and FL-Venus was confirmed. Interestingly, FLAG-MP20 also showed a significant interaction with FL-Venus, which was stronger than that observed with FLAG-FL or FLAG-MP18 (by 5.7-fold relative to FLAG-FL and by 2.3-fold relative to FLAG-MP18). Notably, MP20 Rmt did not reduce this interaction, suggesting that MP20 forms a complex with FL through protein–protein interactions independent of its RNA-binding capacity. Taken together, our findings suggest that MP20 exhibits stronger protein–protein interactions compared with FL and MP18, which may be relevant to heterodimer/oligomer formation.

## hnRNP K induces alternative splicing events within *TARDBP*, culminating in the generation of the TDP-MP20 isoform

To investigate the mechanisms underlying the MP20 isoform production, which exhibits dominant-negative activity, we focused on identifying RBPs that regulate the alternative splicing

of cryptic intron 6 within endogenous *TARDBP*. We systematically overexpressed various RBPs in HEK293T cells and analyzed their effects on the splicing patterns of cryptic intron 6 using RT-PCR. The resulting RT-PCR products included TDP-FL (1,240 bp), MP18s (289/298 bp), MP20s (219/228 bp), MP13 (225 bp), and short MP13 (sMP13, 171 bp) (Fig. 2 A). Notably, due to the similar sizes of MP20s and MP13, these isoforms merged into a single band, designated as MP20+. We strategically selected three cohorts of candidate *TARDBP* splicing regulators: (1) RBPs implicated as causative genes in ALS (TDP-43, FUS, and hnRNP A1) (Akçimen et al., 2023; Zhou et al., 2014), (2) those exclusively expressed in neurons (ELAV like RBP 3 [ELAVL3] and NOVA alternative splicing regulator 1 [NOVA1]) (Zhou et al., 2014), and (3) RBPs purportedly associated with ALS pathophysiology (hnRNP K, hnRNP E1, and hnRNP E2) (Bampton et al., 2021; Braems et al., 2022; Honda et al., 2023; Sidhu et al., 2022; Yoshimura et al., 2021). These RBPs exhibited distinct regulatory patterns in the alternative splicing of *TARDBP* (Fig. 3, B and C). Among the ALS-causative RBPs, TDP-FL reduced the levels of endogenous FL while promoting splicing toward MP18. FUS had a marginal effect on endogenous FL but subtly shifted splicing toward MP20+ at the expense of reducing splicing to MP18. In contrast, hnRNP A1 did not significantly impact endogenous FL but markedly suppressed both MP18 and MP20+, resulting in increased production of sMP13 (Fig. 3, B and C; and Fig. S1 F). In the context of neuron-specific RBPs, a marginal increase in splicing toward MP20+ was observed. Notably, among RBPs implicated in ALS pathogenesis, hnRNP K induced a pronounced reduction in endogenous FL and a significant increase in MP20+, followed by hnRNP E1, while hnRNP E2 had a weaker effect (Fig. 3, B and C). hnRNP E1 promoted splicing toward sMP13, although to a lesser extent than hnRNP A1. Sequencing of PCR products from the MP20+ bands, which increased upon hnRNP K overexpression, revealed that all were MP20 (127). In controls, 21.1% were MP20 (127), 10.5% MP20 (118), and 68.4% MP13 (Fig. 3 D). To further quantify these effects, we performed RT-qPCR using TaqMan MGB probes designed to target the specific sequences of TDP-FL and MP20 (127) (Fig. 3 A).

Consistent with RT-PCR results, RT-qPCR confirmed that hnRNP K overexpression reduced endogenous FL mRNA levels to 62.0% of nontransfected control and induced a 19.4-fold increase in MP20 (127) (Fig. 3 E). These findings suggest that hnRNP K promotes alternative splicing of TDP-43 exon 6, generating short isoforms, especially MP20 (127) from FL (Fig. 3 H). Endogenous FL mRNA levels remained unchanged, and MP20

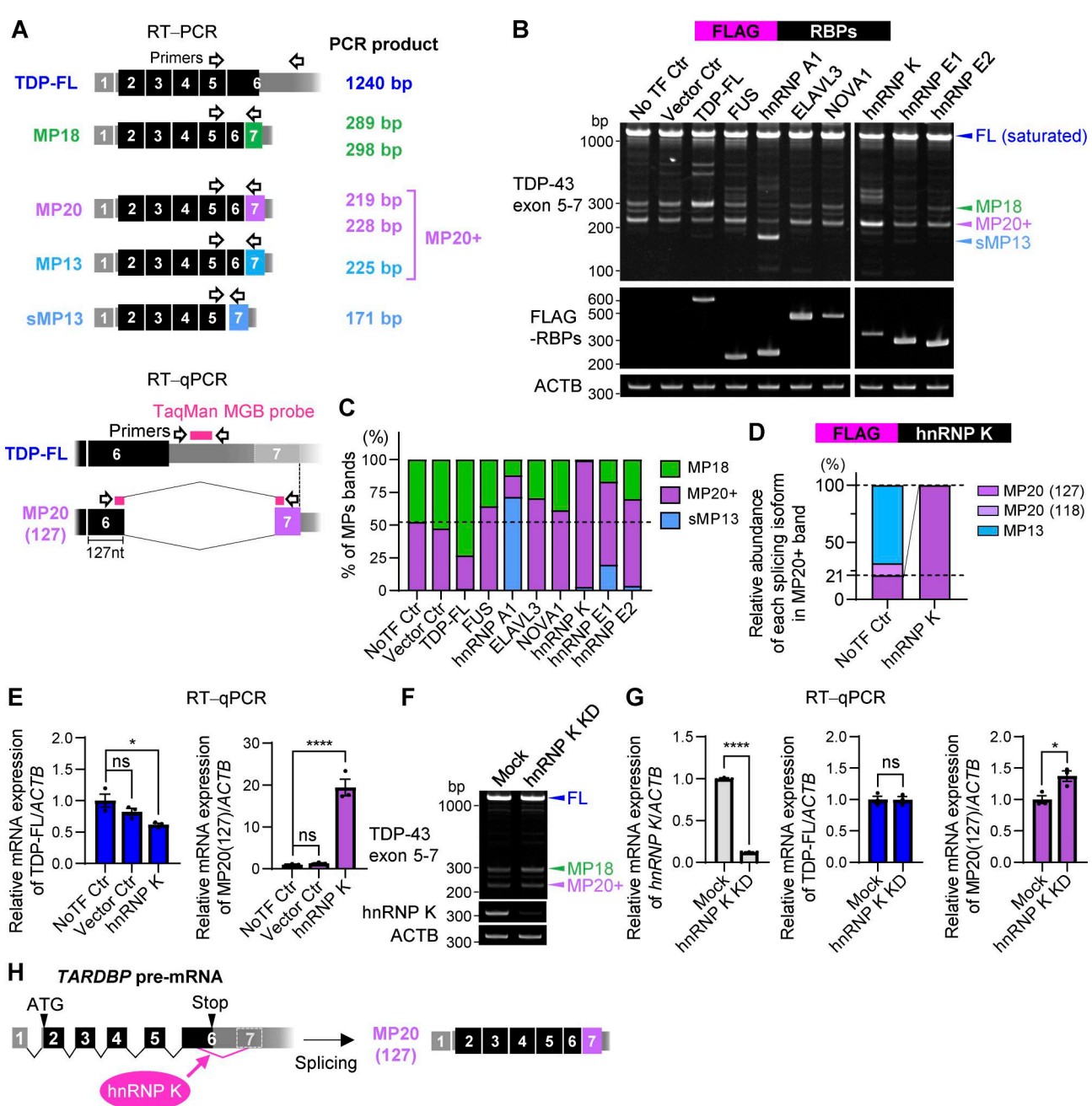

Figure 3. **ALS pathophysiology–associated RBPs induce the production of a dominant-negative splicing isoform of *TARDBP*. (A)** Schematic representation of alternative splicing isoforms of *TARDBP* and corresponding PCR product sizes. Black-framed arrows indicate the primer locations used for RT-PCR analysis of exons 5–7. For RT-qPCR, TaqMan MGB probes (magenta boxes) and primer sets specifically targeting TDP-FL or MP20 (127) transcripts are shown. **(B)** Representative RT-PCR results of *TARDBP* splicing in HEK293T cells overexpressing FLAG-RBPs, including ALS-causative RBPs (TDP-FL, FUS, and hnRNP A1), neuron-specific RBPs (ELAVL3 and NOVA1), and RBPs associated with ALS pathophysiology (hnRNP K, hnRNP E1, and hnRNP E2). A schematic of the sMP13 isoform is also provided in Fig. S1 F. **(C)** Quantification of MP20+ band ratios from B based on RT-PCR analysis. The band intensities were quantified from three independent experiments. **(D)** Relative proportions of MP20+ isoforms determined through sequencing analysis of PCR bands in cells with or without FLAG-hnRNP K overexpression. Analysis included 19 cloned colonies per condition. Dashed lines indicate the MP20+ proportion observed in the NoTF control. **(E)** RT-qPCR analysis of TDP-FL and MP20 (127) mRNA levels in HEK293T cells overexpressing FLAG-hnRNP K. Data are normalized to *ACTB* expression (n = 3 for each group). **(F)** Representative RT-PCR images of *TARDBP* splicing in HEK293T cells with hnRNP K KD. **(G)** RT-qPCR quantification of *hnRNP K*, TDP-FL, and MP20 (127) mRNA levels in hnRNP K KD cells. Data are normalized to *ACTB* expression (n = 3 for each group). Statistical analyses were performed using one-way ANOVA followed by Dunnett's test (E) or Welch's *t* test (G). All graphs display mean ± SEM. *P < 0.05 and ****P < 0.0001. **(H)** Schematic illustrating how hnRNP K promotes the splicing of *TARDBP* to generate the MP20 (127) isoform. Source data are available for this figure: SourceData F3.

(127) unexpectedly increased by 1.37-fold upon hnRNP K KD, suggesting the involvement of another splicing regulator (Fig. 3, F and G).

### Production and evaluation of a polyclonal antibody against endogenous MP20

To study endogenous MP20 (MP20-endo), we generated a polyclonal antibody targeting a unique 20-amino acid sequence at the C terminus of MP20 (Fig. 4 A). The antibody specifically recognized overexpressed FLAG-MP20 (Fig. 4 B). In HeLaS3 cells, MP20-endo predominantly localized to the nucleus, with a minor cytoplasmic presence, similar to overexpressed FLAG-MP20 (Fig. 4, B and C; and Fig. S1 B). At high overexpression levels, FLAG-MP20 tended to localize to the cytoplasm. Antibody specificity was confirmed by the complete loss of immunoreactivity with the immunizing peptide, while the C-terminal peptide of MP18 had no effect (Fig. 4 C). IP in FLAG-MP20–overexpressing HEK293T cells detected a band smaller than 37 kDa with FLAG, TDP-43, and MP20 antibodies (Fig. 4 D). However, excessive FLAG-MP20 overexpression saturated the antibody, hindering the detection of MP20-endo and TDP-43–positive endogenous short variants (SVs-endo).

### MP20 exhibits physiological expression in neurons in vivo

To address the physiological relevance of the spliced isoform MP20, we examined the presence of MP20 in the brain cortex and spinal cord of adult mice. RT-PCR analysis revealed significant differences in TDP-43 splicing patterns between the mouse CNS and human cells, with MP18 splicing being dominant in mouse tissues, while MP20+ splicing was observed in human-derived cells (nonneural HEK293T and neural SH-SY5Y cell lines) (Fig. 4 E). This species-specific difference highlights the unique splicing regulation of TDP-43 in humans. WB and immunofluorescent staining using the MP20-specific antibody confirmed the expression and localization of MP20 in mouse CNS (Fig. 4, F and G). MP20 signals were predominantly nuclear in MAP2-positive neurons (arrowhead), with weaker signals detected in MAP2-negative cells (arrow), possibly glial or other non-neuronal populations (Fig. 4 G). The localization pattern of MP20, primarily nuclear with slight cytoplasmic presence, was consistent with the MP20-endo localization observed in HeLaS3 cells. These findings suggest that MP20 is expressed in neurons in vivo and may play a physiological role, despite its RNA levels remaining low in mice.

### Upregulation of MP20-endo expression is induced by hnRNP K

We investigated whether hnRNP K–mediated facilitation of MP20 splicing increases MP20 protein expression by overexpressing FLAG-hnRNP K in HeLaS3 cells, followed by immunocytochemistry using the newly generated MP20-specific antibody. We compared hnRNP K with hnRNP E1 and hnRNP E2, members of the PCBP family (Bomsztyk et al., 2004; Makeyev and Liebhaber, 2002), as they similarly increased the ratio of MP20+ splicing (Fig. 3 C). hnRNP A1 was included due to its role in promoting a distinct isoform, sMP13 (Fig. 3 C). FLAG-hnRNP K localized in both the nucleus and cytoplasm, with cytoplasmic leakage observed in 18.4% of cells (Fig. 5 A and Fig. S1 G). In

contrast, FLAG-hnRNP E1 and FLAG-hnRNP E2 predominantly localized in the cytoplasm, while FLAG-hnRNP A1 was exclusively nuclear (Fig. 5 A and Fig. S1 G). Quantification of nuclear MP20 fluorescence intensity in immunocytochemistry showed a 1.4-fold increase in the FLAG-hnRNP K–positive cells (arrowheads) compared with FLAG-negative cells (arrow), suggesting that hnRNP K enhances nuclear MP20 expression (Fig. 5 B). A similar increase in nuclear MP20 was observed in FLAG-hnRNP E1– and FLAG-hnRNP E2–positive cells, while FLAG-hnRNP A1–positive cells showed no detectable change in MP20 levels (Fig. 5 B). Notably, in cells where FLAG-hnRNP K exhibited cytoplasmic leakage, the nuclear fluorescence intensity of MP20 was significantly increased by 1.2-fold compared with cells with exclusively nuclear FLAG-hnRNP K localization (Fig. 5 C). WB analysis in HEK293T cells further confirmed that overexpression of hnRNP K and hnRNP E1 increased MP20-endo and SVs-endo levels (Fig. 5 D). A similar trend was observed with hnRNP E2, although the increase was not statistically significant. In contrast, overexpression of hnRNP A1 led to a reduction in both SVs-endo and MP20, a result that did not align with the immunocytochemistry data (Fig. 5 D). Consistent with WB results, hnRNP A1 KD induced a 2.71-fold increase in MP20 (127) mRNA levels and a 2.02-fold increase in protein levels, as confirmed by immunocytochemistry, while FL mRNA and protein levels remained unchanged (Fig. S1, H–K). hnRNP K KD in HEK293T cells showed a slight increase in SVs-endo and MP20-endo levels in WB analysis, though this change was not statistically significant (Fig. 5 E). To confirm that the band increased by hnRNP K overexpression in WB corresponded to MP20, we performed IP with the MP20 antibody in FLAG-hnRNP K–overexpressing HEK293T cells, detecting SVs-endo and MP20-endo bands. The interaction was blocked by the MP20-immunizing peptide but not by the MP18 peptide, further supporting that hnRNP K promotes MP20 splicing and protein expression (Fig. 5 F).

### Increased hnRNP K expression induces the expression of hnRNP A1 through a posttranscriptional mechanism

Based on the findings in Fig. 3, both hnRNP K and hnRNP A1 significantly influence the TARDBP exon 6 splicing (Fig. 3, B and C), suggesting their involvement as key regulators. To determine whether hnRNP K influences hnRNP A1 expression, we examined endogenous hnRNP A1 protein and mRNA levels in FLAG-hnRNP K–overexpressing HEK293T cells. hnRNP K increased hnRNP A1 protein levels by 6.6-fold, while mRNA levels decreased to 68.0%, with no significant changes in hnRNP A2/B1 protein levels (Fig. 6 A and Fig. S2 A). Consistently, immunocytochemistry in HeLaS3 cells confirmed increased hnRNP A1 fluorescence in FLAG-hnRNP K–positive cells, while hnRNP A2/B1 showed a nonsignificant trend toward increased levels (Fig. 6, B and C). In addition, cells with cytoplasmic leakage of FLAG-hnRNP K exhibited significantly increased hnRNP A1 and hnRNP A2/B signals compared with cells with nuclear FLAG-hnRNP K (Fig. 6 D). Conversely, hnRNP K KD reduced hnRNP A1 protein levels to 58.9%, without affecting hnRNP A1 mRNA levels (Fig. 6 E and Fig. S2 B). To assess reciprocal effects, we examined the influence of FLAG-hnRNP A1 overexpression

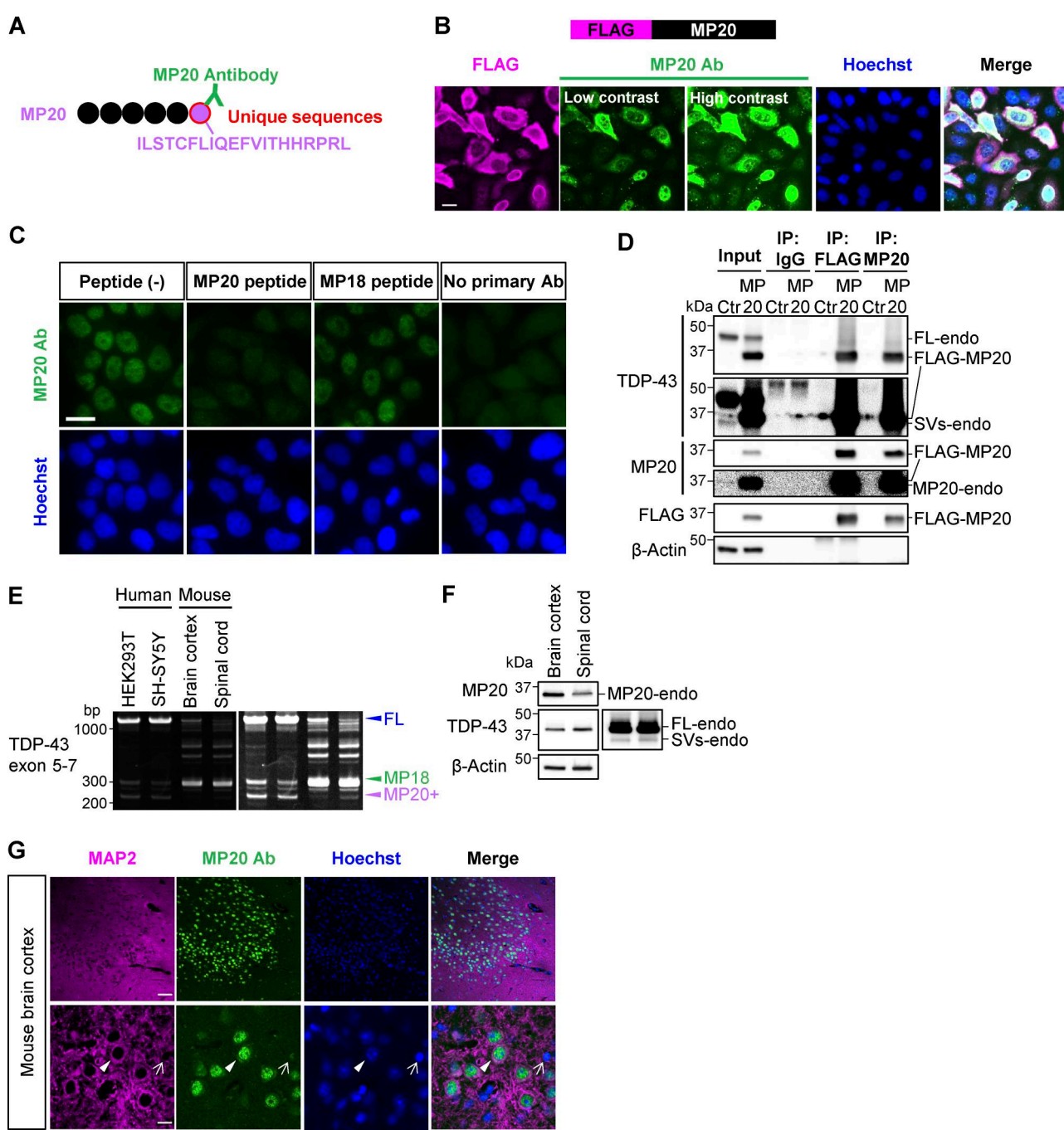

Figure 4. **Quality evaluation of a polyclonal antibody against endogenous TDP-MP20. (A)** Schematic representation of MP20 antibody (MP20 Ab) recognizing a unique C-terminal peptide sequence of MP20. **(B)** Immunocytochemistry results showing FLAG-MP20 overexpression in HeLaS3 cells. MP20 Ab (green), FLAG (magenta), and Hoechst (blue). Scale bar: 20 μm. **(C)** Immunocytochemistry showing MP20-endo staining in HeLaS3 cells, using a peptide-absorbed MP20 Ab (green) and Hoechst (blue). MP18 peptide served as a negative control. Scale bar: 20 μm. **(D)** IP analysis of FLAG-MP20–overexpressing HEK293T cells using FLAG or MP20 Ab. An anti-IgG Ab was used as a negative control. Endogenous TDP-43 SVs (SVs-endo) include cleaved FL-endo and variants derived from short isoforms. **(E)** RT-PCR results comparing *TARDBP* splicing patterns in human cell lines (HEK293T and SH-SY5Y), adult mouse (11 mo old, male) brain cortex, and spinal cord. **(F)** WB analysis of MP20 expression in adult mouse (22 mo old, male) brain cortex and spinal cord. **(G)** Fluorescent immunostaining of adult mouse (22 mo old, male) brain cortex, stained with MP20 (green), MAP2 (magenta), and Hoechst (blue). Arrowheads and arrows denote MAP2-positive and MAP2-negative cells, respectively. Scale bar: 50 μm (upper panel), 10 μm (lower panel). Source data are available for this figure: SourceData F4.

on hnRNP K protein and mRNA levels. FLAG-hnRNP A1 did not affect either the protein or mRNA levels of endogenous hnRNP K (Fig. S2, C–E). hnRNP A1 KD slightly increased hnRNP K mRNA levels by 1.19-fold, with no changes in

hnRNP K protein levels (Fig. S2, F and G). These findings suggest that hnRNP K primarily acts upstream of hnRNP A1. IP experiments revealed an interaction between FLAG-hnRNP A1 and endogenous hnRNP K (Fig. S2 H). Our results

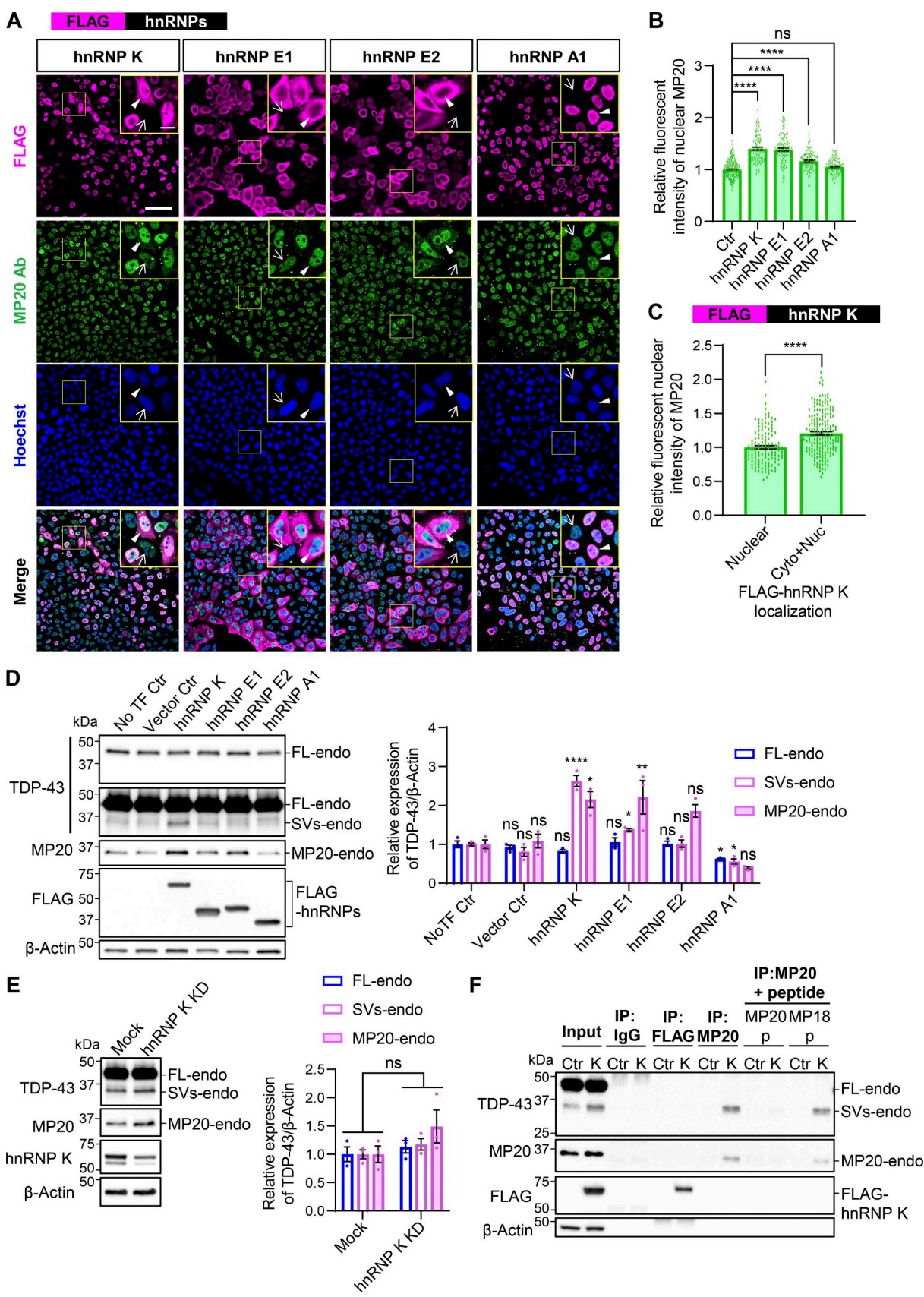

Figure 5. **hnRNP K induces the protein expression of endogenous TDP-MP20. (A)** Immunocytochemistry of HeLaS3 cells overexpressing FLAG-hnRNPs. MP20-endo (green), FLAG (magenta), and Hoechst (blue). Yellow boxes indicate magnified images (scale bar: 10 μm). Arrowheads and arrows denote FLAG-positive and FLAG-negative cells, respectively. Scale bar: 50 μm. **(B)** Quantification of nuclear MP20 fluorescence intensity in FLAG-hnRNPs–positive cells. A

total of 100 FLAG-positive and 100 FLAG-negative cells per group were analyzed. Fluorescence signals were normalized to the average intensity of 400 FLAG-negative cells. Statistical analysis was performed using one-way ANOVA followed by Dunnett's test. **(C)** Comparison of nuclear MP20 signals in cells with FLAG-hnRNP K localized in the nucleus only (Nuc) versus in both the cytoplasm and nucleus (Cyto+Nuc). Nuc: 123 cells; Cyto+Nuc: 177 cells. Welch's $t$ test was used for statistical analysis. **(D and E)** WB analysis of HEK293T cells overexpressing FLAG-hnRNPs (D) or with hnRNP K KD (E). Quantification of the relative expression of FL-endo, SVs-endo, and MP20-endo ($n = 3$ for each group). Data are normalized to β-actin. Statistical analyses were performed using one-way ANOVA followed by Dunnett's test (D) or Welch's $t$ test (E). **(F)** IP of FLAG-hnRNP K overexpressed in HEK293T cells using MP20 or FLAG Ab, with or without peptide absorption by MP20- or MP18-immunizing peptides. An anti-IgG Ab was used as a negative control. All graphs show the mean ± SEM. *P < 0.05, **P < 0.01, and ****P < 0.0001. Ab, antibody. Source data are available for this figure: SourceData F5.

suggest that translational or posttranslational regulatory mechanisms are likely involved.

## hnRNP A1 inhibits the effect of hnRNP K on the alternative splicing of *TARDBP* in an RNA recognition capacity-dependent manner

RBPs often interact in complex networks, exhibiting inhibitory or synergistic effects on shared splicing targets (Fu and Ares, 2014). To investigate the relationships between hnRNP K and hnRNP A1 in *TARDBP* splicing regulation, we performed RT-PCR analysis using a FLAG-tagged *TARDBP* mini-gene construct encompassing exon 5, intron 5, and exon 6 1–2,100 nt (Ex5-Int5-Ex6) (Fig. 7 A). The Ex5-Int5-Ex6 construct predominantly produced TDP-FL, MP18s, and MP20+ isoforms (Fig. 7 B). Coexpression of the mini-gene with FLAG-hnRNP K or FLAG-hnRNP A1 replicated the splicing patterns observed for endogenous TDP-43 in Fig. 3 B. Specifically, hnRNP K promoted MP20+ splicing, while hnRNP A1 shifted splicing toward sMP13 (Fig. 7 B). Interestingly, when both hnRNP K and hnRNP A1 were coexpressed, splicing shifted toward several shorter isoforms in an hnRNP A1 dose-dependent manner, with reductions in both MP20+ and sMP13 (Fig. 7 B). Consistently, similar effects were observed in endogenous *TARDBP* exon 6 splicing (Fig. 7 C). RT-qPCR analysis showed that MP20 (127) mRNA, which increased 7.51-fold with hnRNP K overexpression, was significantly reduced upon coexpression with hnRNP A1 (13.4-fold decrease), while FL mRNA levels were reduced to a degree comparable with hnRNP K overexpression alone (Fig. 7 D). These results suggest that hnRNP K and hnRNP A1 cooperate to drive splicing toward shorter isoforms. At the protein level, the combination of FLAG-hnRNP K and FLAG-hnRNP A1 reduced SVs-endo and MP20-endo levels by 73.6% and 60.0%, respectively, compared with hnRNP K overexpression alone (Fig. 7 E). To determine whether these effects depend on the RNA-binding capacity of hnRNP A1, we generated Rmt's (R1mt and R2mt) of hnRNP A1 (Mayeda et al., 1994) (Fig. 7 F). Overexpression of R1mt or R2mt alone failed to promote the splicing of *TARDBP* into sMP13, indicating that RNA-binding activity of hnRNP A1 is essential for the splicing regulation (Fig. 7 G). Coexpression of either mutant with hnRNP K abolished the WT hnRNP A1 effect; the reductions in MP20+ and sMP13, as well as the increased production of shorter isoforms, were not observed. Quantitative analysis confirmed that MP20 (127) mRNA levels, which were reduced by coexpression of hnRNP K and WT hnRNP A1, were restored in cells expressing the mutants (Fig. 7 H). FL mRNA levels, however, remained similar to those observed with WT hnRNP A1 coexpression. Furthermore, at the protein level, the reduction in MP20-endo and SV-endo observed with WT hnRNP A1 and

hnRNP K coexpression were absent when either R1mt or R2mt was coexpressed with hnRNP K (Fig. 7 I). These findings demonstrate that the RNA-binding activity of hnRNP A1 is crucial for its cooperative effects with hnRNP K in regulating *TARDBP* splicing. Collectively, these results suggest that hnRNP A1 modulates *TARDBP* alternative splicing by counteracting the hnRNP K-mediated increase in MP20 and the production of shorter isoforms.

To further explore the specific region of *TARDBP* RNA involved in hnRNP A1–mediated splicing regulation, we utilized the *TARDBP* mini-gene (Ex5-Int5-Ex6) with targeted deletions of potential hnRNP A1–binding sites (Fig. S3 A). Despite the presence of a putative hnRNP A1–binding site in intron 5 ($_{414}$UAGGGU$_{419}$), deletion of this site did not affect the splicing regulation by hnRNP A1 (Fig. S3 B). We next generated a series of systematic deletion mutants within intron 5, covering nt positions 51–1,474 (Fig. S3 C). These deletion mutants also showed no change in splicing patterns (Fig. S3 D), indicating that the hnRNP A1 target sequence is not within this region. We then assessed the role of cryptic intron 6, but deletion of regions within this intron also did not affect hnRNP A1–dependent splicing (Fig. S3, E and F). These results suggest that hnRNP A1 may interact with sequences outside of these regions or that weaker binding sites may compensate for deletions.

In summary, excess hnRNP K promotes alternative splicing, generating MP20 (127) and subsequently increasing MP20 protein levels (Fig. S2 I), whereas hnRNP A1 promotes splicing that excises intron 5 to exon 7, resulting in sMP13 with undetectable protein levels (Fig. S2 J). Coexpression of hnRNP K and hnRNP A1 impedes the formation of MP20 and promotes the generation of shorter isoforms, which are not detected by protein expression (Fig. S2 K). These findings highlight the critical role of hnRNP A1 in modulating *TARDBP* splicing, particularly in counteracting the hnRNP K-mediated increase in MP20 production.

## Induction of endogenous TDP-MP20 by hnRNP K is correlated with hnRNP A1

To identify the regions of hnRNP K required for regulating TDP-43 splicing and inducing hnRNP A1, we generated deletion mutants lacking either the KH domains or the K protein–interactive region (KI) and K nuclear–shuttling (KNS) domains (Fig. 8 A). Deletion of the KH1, KH2, or KI/KNS domains abolished the ability of hnRNP K to promote MP20+ splicing, while deletion of KH3 had no significant effect (Fig. 8 B). Similarly, RT-qPCR analysis showed that MP20 (127) mRNA levels were not increased in the KH1, KH2, or KI/KNS deletion mutants, while FL mRNA levels remained largely unchanged (Fig. 8 C). Consistently, protein analysis confirmed that MP20 and hnRNP A1

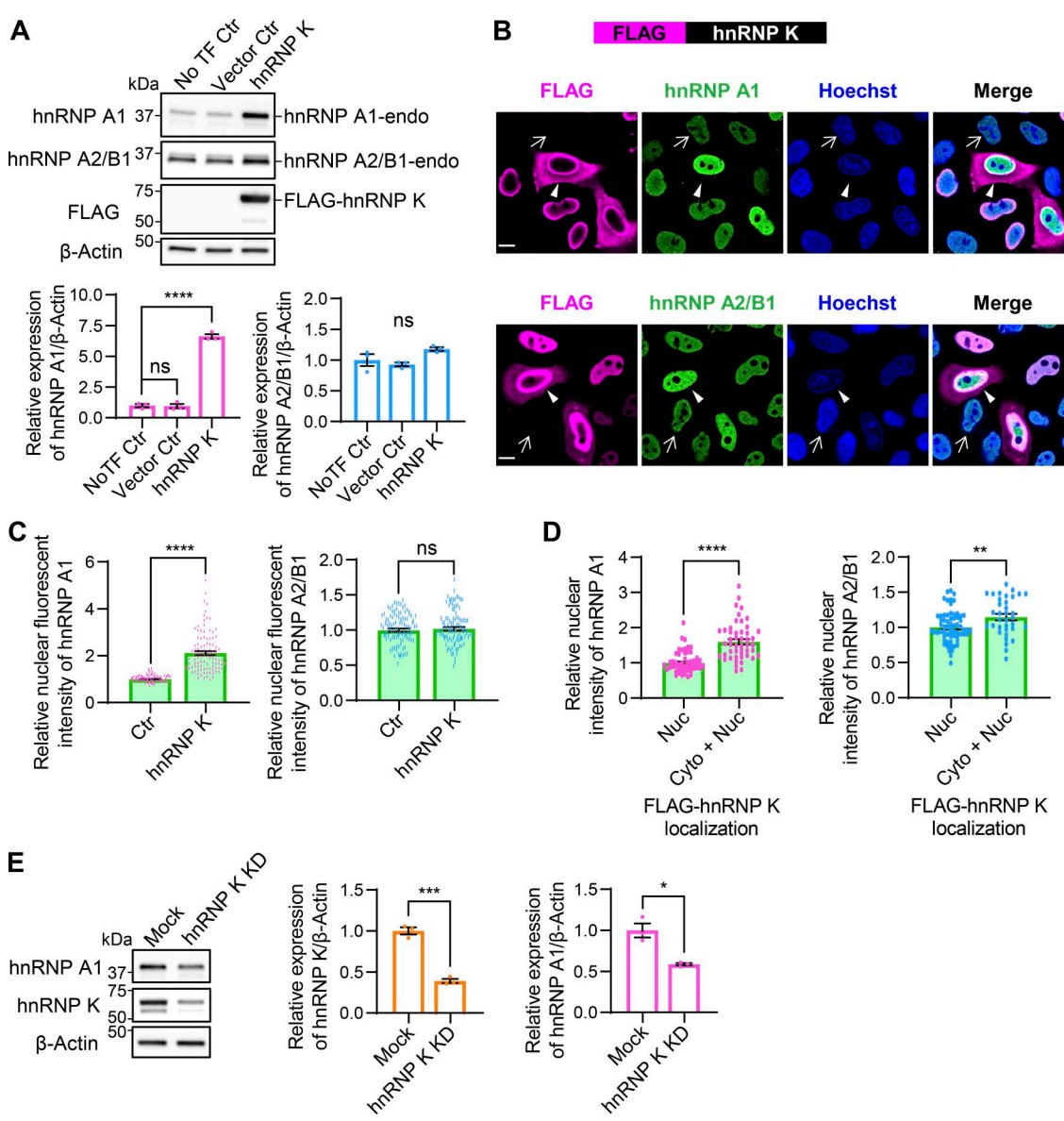

Figure 6. **hnRNP K enhances hnRNP A1 expression. (A)** WB analysis of HEK293T cells overexpressing FLAG-hnRNP K. Quantification of relative expression levels of hnRNP A1 and hnRNP A2/B1 is shown (*n* = 3 for each group). **(B)** Immunocytochemistry of HeLaS3 cells overexpressing FLAG-hnRNP K. FLAG (magenta), hnRNP A1 or hnRNP A2/B1 (green), and Hoechst (blue). Arrowheads indicate FLAG-positive cells, while arrows indicate FLAG-negative cells. Scale bar: 10 μm. **(C)** Quantification of nuclear hnRNP A1 or hnRNP A2/B1 fluorescence intensity in FLAG-hnRNP K–positive and –negative cells. A total of 100 FLAG-positive and 100 FLAG-negative cells per group were analyzed. Statistical analyses were performed using Welch's *t* test. **(D)** Comparison of nuclear hnRNP A1 or hnRNP A2/B1 fluorescence intensity in cells with FLAG-hnRNP K localized only in the nucleus (Nuc) versus in both the cytoplasm and nucleus (Cyto+Nuc). For hnRNP A1, Nuc: 52 cells, Cyto+Nuc: 48 cells; for hnRNP A2/B1, Nuc: 64 cells, Cyto+Nuc: 35 cells. Statistical analyses were performed using Welch's *t* test. **(E)** WB analysis of HEK293T cells with hnRNP K KD. Quantification of relative expression levels of hnRNP K or hnRNP A1 is shown (*n* = 3 for each group). In A and E, data are normalized to β-actin. Statistical analyses were performed using one-way ANOVA followed by Dunnett's test (A) or Welch's *t* test (E). All graphs show the mean ± SEM. *P < 0.05, **P < 0.01, ***P < 0.001, and ****P < 0.0001. Source data are available for this figure: SourceData F6.

levels were not elevated in any mutants, except for the KH3 deletion mutant, while hnRNP A2/B1 expression was unaffected (Fig. 8 D). These results highlight the critical role of the KH1, KH2, and KI/KNS domains in regulating *TARDBP* splicing and inducing MP20 and hnRNP A1 expression. To further assess the contribution of DNA/RNA binding, we introduced mutations (GxxG→GDDG) into the KH domains, preserving protein structure while impairing binding activity (Fig. 8 A) (Backe et al., 2005; Braddock et al., 2002; Hollingworth et al., 2012; Yin

et al., 2020). Mutants of KH1, KH2, or KH3 (KH1mt, KH2mt, and KH3mt) retained the ability to promote MP20 splicing and induce MP20 and hnRNP A1 expression (Fig. 8, E–G). Consistently, RT-qPCR analysis of these mutants showed increased MP20 (127) mRNA levels with no significant changes in FL mRNA (Fig. 8, E and F). However, combined mutations in KH1/2 or KH1/2/3 completely abolished these effects. These findings underscore the essential role of DNA/RNA binding in the KH1 and KH2 domains for hnRNP K–mediated splicing regulation and

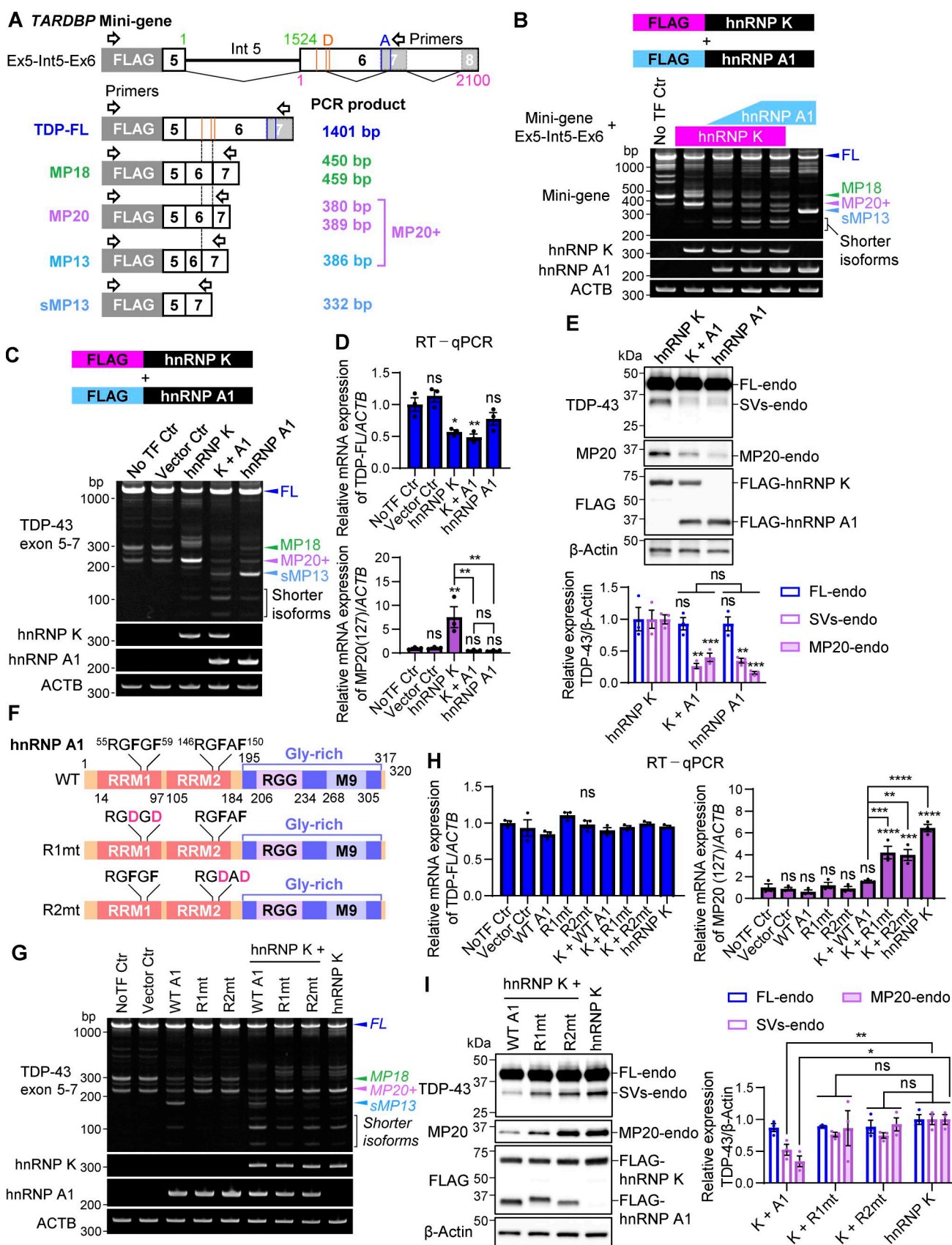

Figure 7. **hnRNP A1 inhibits the effect of hnRNP K on the alternative splicing of *TARDBP*. (A)** Schematic representation of the mini-gene containing *TARDBP* exon 5 to exon 6 (1–2,100 nt) (mini-gene Ex5-Int5-Ex6) and the spliced PCR product length. Black-framed arrows indicate the primer locations for FLAG amplification to exon 7 of *TARDBP*. Exons and intron 5 (Int 5) are shown as white boxes with black lines, while the pale gray box with a dotted line represents the

region that becomes exon 7 through alternative splicing. Orange and blue lines represent alternative donor (D) and acceptor (A) sites, respectively. **(B)** RT-PCR images showing *TARDBP* mini-gene splicing in cells overexpressing FLAG-hnRNP K and FLAG-hnRNP A1 at different doses. **(C and D)** RT-PCR images of endogenous *TARDBP* splicing (C) or RT-qPCR analysis of TDP-FL and MP20(127) mRNA levels (D) in HEK293T cells co-overexpressing FLAG-hnRNP K and FLAG-hnRNP A1. **(E)** WB analysis of HEK293T cells co-overexpressing FLAG-hnRNP K and FLAG-hnRNP A1. **(F)** Schematic diagram of hnRNP A1 RRM1 or RRM2 domain mutants (R1mt and R2mt). **(G)** RT-PCR images of endogenous *TARDBP* splicing in HEK293T cells coexpressing FLAG-hnRNP A1 mutants (R1mt or R2mt) with or without FLAG-hnRNP K. **(H)** RT-qPCR analysis of TDP-FL and MP20 (127) mRNA expression levels. In D and H, data are normalized to *ACTB*. Statistical analyses were performed using one-way ANOVA followed by Tukey's test ($n = 3$ for each group). **(I)** WB analysis of HEK293T cells co-overexpressing FLAG-hnRNP A1 RRM domain mutants (R1mt or R2mt) and FLAG-hnRNP K. In E and I, quantification of relative expression levels of FL-endo, SVs-endo, and MP20-endo is shown ($n = 3$ for each group). Data are normalized to β-actin, and statistical analyses were performed using one-way ANOVA followed by Tukey's test (E) or Dunnett's test (I). All graphs show the mean ± SEM. *$P < 0.05$, **$P < 0.01$, ***$P < 0.001$, and ****$P < 0.0001$. Gly-rich, glycine-rich region; RGG, arginine-glycine-glycine repetitive region. Source data are available for this figure: SourceData F7.

hnRNP A1 induction. Collectively, our data support a model in which the induction of MP20 and hnRNP A1 by hnRNP K is mechanistically linked.

### The main functional domain of hnRNP K involved in alternative splicing of *TARDBP* exon 6 is not observed within cryptic intron 6

To investigate the specific region on *TARDBP* RNA where hnRNP K exerts its function, we utilized a *TARDBP* exon 6 mini-gene (Ex6) containing nt's 1–2,100 of exon 6 (Fig. S4 A). Unlike the previously described mini-gene (Ex5-Int5-Ex6) (Fig. 7 A and Fig. S3), this construct excludes exon 5 and intron 5. Coexpression of FLAG-hnRNP K enhanced MP20+ splicing of this mini-gene, indicating that exon 5 and intron 5 are not required for this effect. Previous studies have demonstrated that the hnRNP K preferentially binds to 5′-GCCCA-3′ sequences (Van Nostrand et al., 2020) and stem-loop structures (Dominguez et al., 2018). Based on these findings, we predicted the secondary structure of *TARDBP* exon 6 and identified two potential consensus hnRNP K–binding sites, $_{262}$GCCCA$_{266}$ (located on a stem structure, K-1) and $_{329}$GCCCA$_{333}$ (K-2) (Fig. S4 A). However, deletion of these sequences from the *TARDBP* mini-gene did not abolish the effects of hnRNP K on MP20+ splicing, suggesting these sites are not essential for its regulation (Fig. S4 B). To identify broader regions of *TARDBP* RNA important for hnRNP K–mediated splicing regulation, we generated systematic deletion mutants of the Ex6 mini-gene, each removing ∼100 bp within cryptic intron 6 (Fig. S4 C). Among these mutants, Del 866–985, which deleted a region just upstream of the predicted polypyrimidine tract, showed the least response to FLAG-hnRNP K coexpression (Fig. S4 D). However, this modest response suggests that this region is not critical for hnRNP K–mediated regulation. Notably, none of the mini-genes entirely abolished the effects of hnRNP K, indicating that hnRNP K does not exert its effect via these regions.

Unexpectedly, our analysis instead revealed a potential regulatory region between exon 6 residues 485–584 that may modulate hnRNP K activity. Specifically, the Del 485–584 mini-gene failed to sustain FL splicing and disrupted splicing to short isoforms upon hnRNP K coexpression (Fig. S4 D). This suggests that the 485–584 region may serve as a binding site for an RBP that suppresses hnRNP K activity, thereby, influencing its regulatory effect on *TARDBP* splicing.

### The downstream GU-rich region following the FL stop codon is crucial for *TARDBP* exon 6 splicing

We further analyzed *TARDBP* exon 6 residues 485–584, including the FL stop codon (FL-stop, 529–531), using the mini-genes (Ex6) with narrower deletions (Fig. S4 E). Deletion of 48-bp upstream of FL-stop (Del 485–531) did not significantly affect hnRNP K–mediated splicing (Fig. S4 F). In contrast, a 54-bp deletion downstream of FL-stop (Del 532–584) disrupted the production of short isoforms (MP13, MP20, and MP18), highlighting the importance of this region in maintaining the normal splicing patterns. Upon coexpression with hnRNP K, the Del 532–584 mini-gene showed a dramatic reduction in FL levels and aberrant production of short splicing isoforms, indicating increased sensitivity to hnRNP K (Fig. S4 F). The downstream region of FL-stop contains both guanine–uracil (GU)-rich (535–555) and adenine–uracil (AU)-rich (569–584) sequences. Deletion of the GU-rich region (Del 535–555) disrupted splicing patterns similar to Del 532–584, with a comparable response to hnRNP K. In contrast, deletions of the AU-rich region (Del 569–584) or the intervening interval region (Del 556–568) had minimal effects, suggesting that the GU-rich region is primarily responsible for these effects. The GU-rich sequence (535–555) is highly conserved in mice (Fig. 9 A). Notably, this region contains FUS-binding motifs, such as GUGGU (Lagier-Tourenne et al., 2012) and GGUG (Iko et al., 2004; Lerga et al., 2001) (Fig. 9 A). UV cross-linking followed by IP using an anti-FUS antibody confirmed that endogenous FUS associates with *TARDBP* RNA (Fig. S5, A–C). These findings raise the possibility that FUS may contribute to modulating hnRNP K sensitivity through its interaction with this region.

### The ALS-causative FUS mutant exhibits reduced inhibitory effects on hnRNP K and MP20 expression

To explore whether FUS regulates hnRNP K and subsequent MP20, we investigated the effects of ALS-associated P525L FUS mutant, which is a mutation in the nuclear localization signal of FUS, disrupting its nuclear localization and inducing cytoplasmic aggregation (Lenzi et al., 2015; Liu et al., 2015) (Fig. 9 B). This allowed us to examine how the loss of FUS nuclear function and/or its abnormal cytoplasmic localization might influence hnRNP K and MP20 expression. WT FUS significantly increased MP20 (127) mRNA levels by 1.8-fold while reducing FL to 58.0% (Fig. 9 C). In contrast, the P525L FUS caused a marked increase in MP20 (127) mRNA levels (3.15-fold compared with nontransfected control, 1.73-fold compared with WT FUS) but did not reduce FL

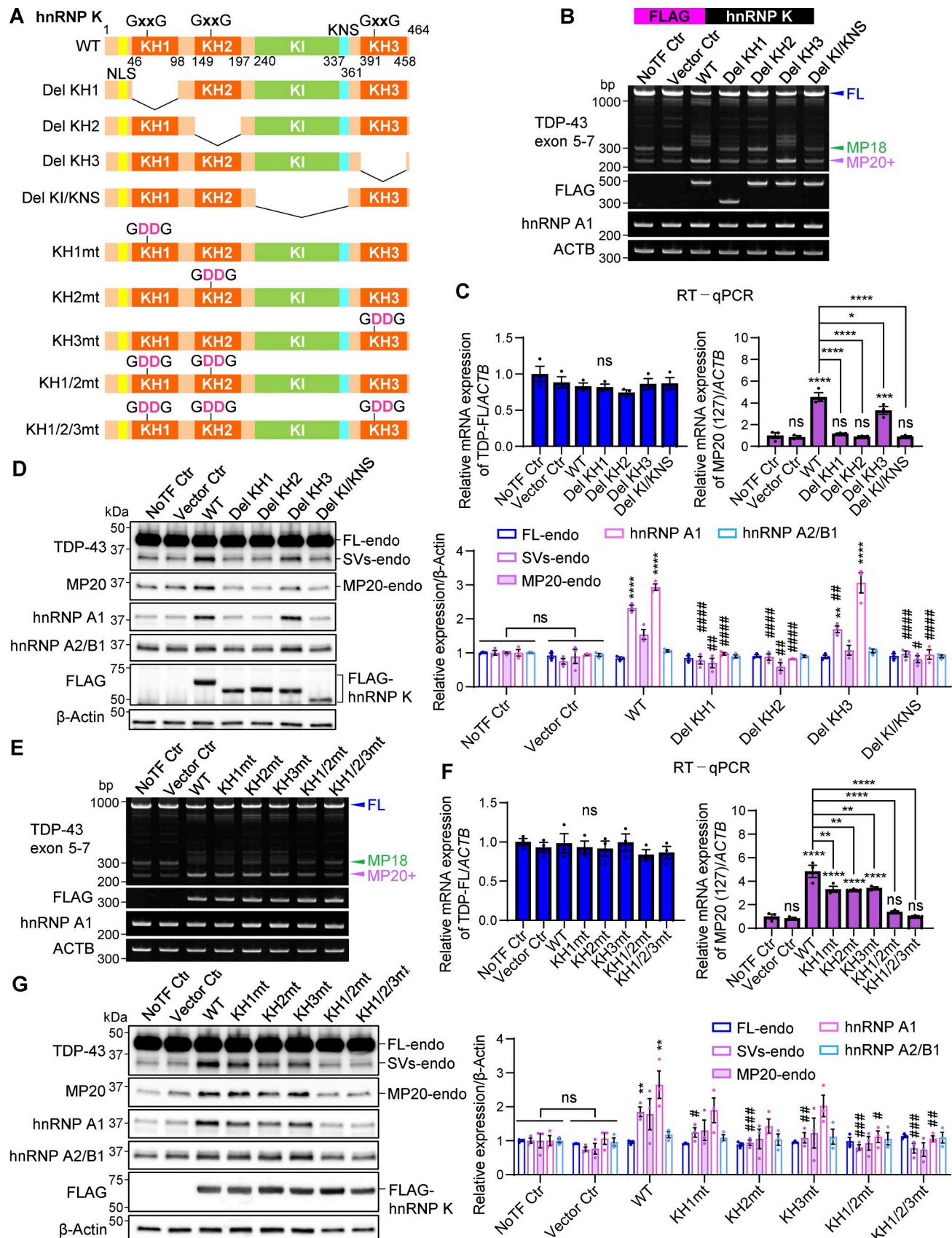

Figure 8. **The induction of endogenous TDP-MP20 expression by hnRNP K correlates with that of hnRNP A1. (A)** Schematic representation of hnRNP K with KH or KI/KNS domain deletion (*Del* KH, *Del* KI/KNS) or with KH domain mutations (KHmt). **(B and C)** RT-PCR images of *TARDBP* splicing (B) or RT-qPCR

analysis of TDP-FL and MP20 (127) mRNA expression levels (C) in HEK293T cells overexpressing hnRNP K deletion mutants. **(D)** WB of HEK293T cells overexpressing deletion mutants of hnRNP K. **(E and F)** RT-PCR images of *TARDBP* splicing (E) or RT-qPCR analysis of TDP-FL and MP20 (127) mRNA expression levels (F) in HEK293T cells overexpressing KHmts. In C and F, data are normalized to *ACTB*. Statistical analyses were performed using one-way ANOVA followed by Tukey's test. **(G)** WB of HEK293T cells overexpressing KHmts. In D and G, quantification of the relative expression of FL-endo, SVs-endo, MP20-endo, hnRNP A1, and hnRNP A2/B1 is shown, respectively. Data are normalized to β-actin. Statistical analyses were performed using one-way ANOVA followed by Tukey's test (*n* = 3 for each group). All graphs show the mean ± SEM. *P < 0.05, **P < 0.01, ***P < 0.001, and ****P < 0.0001 compared with the NoTF Ctr group, and #P < 0.05, ##P < 0.01, ###P < 0.001, and ####P < 0.0001 compared with the WT hnRNP K group. NoTF Ctr, nontransfected control; NLS, nuclear localization signal. Source data are available for this figure: SourceData F8.

levels. Both WT and P525L FUS reduced hnRNP K mRNA levels by ∼30%. Consistently, WB analysis revealed a reduction of hnRNP K protein levels by around 25% and MP20 protein levels by ∼50% following overexpression of WT and P525L FUS (Fig. 9, D and F). However, FL-endo and SVs-endo protein levels were unaffected by the overexpression of either WT or P525L FUS (Fig. S5 D). Immunocytochemistry confirmed that FLAG-WT FUS predominantly localized to the nucleus, while FLAG-P525L FUS was mainly cytoplasmic (Fig. 9, E and G). The relative nuclear intensity of hnRNP K in FLAG-positive cells was reduced by 76.1% with WT FUS, while P525L FUS showed minimal suppression (Fig. 9 E). Similarly, the nuclear intensity of MP20 decreased by 56.2% with WT FUS, but only by 14.5% with P525L (Fig. 9 G). These findings suggest that FUS suppresses hnRNP K at both RNA and protein levels, with the P525L mutation impairing this suppression and leading to enhanced MP20 expression.

Interestingly, while FUS increased MP20 (127) mRNA levels, MP20 protein levels were paradoxically reduced, suggesting that FUS may regulate MP20 expression not only indirectly via hnRNP K but also directly by modulating MP20 translation. The 3′UTR of MP20 contains multiple FUS-binding motifs (GUGG), including sequences downstream of the stop codon, conserved between humans and mice (Fig. 9 H and Fig. S5 B). We hypothesized that FUS regulates MP20 translation via its 3′UTR and constructed a reporter by inserting the MP20 3′UTR (*TARDBP* exon 6) downstream of Venus (Fig. S5 E). This construct included the FUS consensus motifs (GUGG and GUGGU) immediately downstream of the MP20 stop codon, along with additional motifs located further downstream. This allowed for a more comprehensive evaluation of FUS-mediated translational regulation. Coexpression of FLAG-WT FUS resulted in a dose-dependent decrease in Venus expression containing the MP20 3′UTR, while Venus without the 3′UTR was unaffected (Fig. S5 F). Similarly, P525L FUS modulated the translation of Venus containing the 3′UTR but had no effect on Venus without it (Fig. S5 G). These results confirm that FUS regulates MP20 expression both indirectly through hnRNP K and directly by modulating MP20 translation via its 3′UTR (Fig. 10, A–C).

## Discussion

A previous study showed that overexpression of the MP20 isoform (referred to as H5) reduces endogenous TDP-FL levels and alters TDP-43 splicing targets, mimicking patterns seen under TDP-43 KD conditions (D'Alton et al., 2015). Consistently, we observed not only similar splicing changes in *BCL2L11* but also aberrant CE inclusion in *GPSM2* and *ATG4B* following MP20

overexpression as well as TDP-43 KD (Fig. 2 A and Fig. S1 D). MP20 shares functional domains from the N-terminal domain (NTD) to RRM2 with FL but lacks the C-terminal region, which is essential for interactions with other splicing regulatory proteins. For instance, TDP-43 splicing activity in CFTR exon 9 skipping relies on protein–protein interactions between its C-terminal residues 321–366 and hnRNP A2 and A1 (Buratti et al., 2005; D'Ambrogio et al., 2009). Given that MP20 lacks this region, it is likely unable to regulate splicing independently.

While all MPs analyzed in this study similarly reduced FL-endo protein levels, their impact on CE inclusion varied (Fig. 1, B and C; and Fig. 2 A). FL forms functional homodimers/polymers thorough its NTD, which are critical for splicing regulation (Afroz et al., 2017; Mompeán et al., 2017). MPs retain the NTD, allowing them to form heterocomplexes with FL. Indeed, MP20, which significantly promoted CE inclusion, demonstrated higher complex formation with FL either FL or MP18 did, as shown by co-IP (Fig. 2 H). Therefore, MP20 may inhibit functional FL homodimer/polymer formation by competitively forming heterocomplexes, contributing to its dominant-negative activity. Despite reducing FL-endo levels, MP18 overexpression did not induce CE inclusion, and changes in splicing targets were similar to FL overexpression (Fig. 2 A). These findings suggest that the MP18-FL heterocomplex may partially retain the splicing function of FL homocomplexes.

The expression of MP18 (also described as sTDP-43) has been shown to be induced by hyperexcitability in iPSC-derived neurons treated with potassium channel blocker (Weskamp et al., 2020). This study demonstrated that an endogenous TDP-43 splicing isoform can be upregulated by exogenous stimuli, and detected the endogenous protein using a specific antibody. Future investigations with our anti-MP20 antibody in ALS postmortem tissues will offer further observation. In addition, recent work from the same research group showed that MP18 is generated through TDP-43 autoregulation and may exert a potent dominant-negative effect (Dykstra et al., 2025). Our findings are consistent with this concept, demonstrating that alternative splicing of TDP-43 produces isoforms with dominant-negative potential. In particular, MP20 induced CE inclusion more clearly than MP18 (Fig. 2, A and B), suggesting a more pronounced dominant-negative activity.

Our results indicate that MP20 expression is particularly elevated in cells with cytoplasmic leakage of hnRNP K (Fig. 5 C), suggesting the promotion of MP20 splicing by hnRNP K (Fig. 3, B–E) does not appear to result from direct interaction with *TARDBP* RNA. This is supported by the absence of a specific *TARDBP* region regulated by hnRNP K, as shown by experiments using *TARDBP* exon 6 mini-gene constructs with systematic

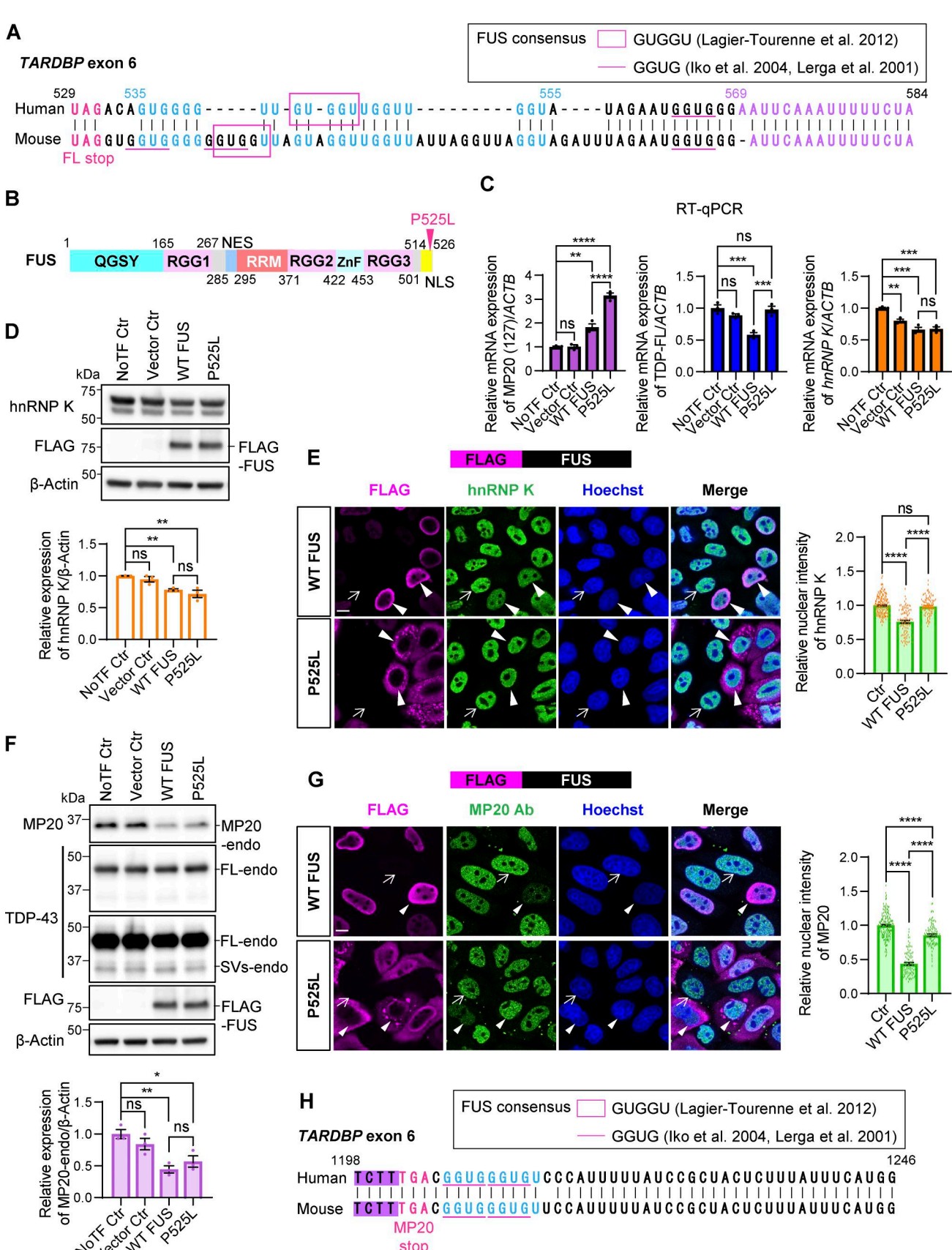

Figure 9. **ALS-associated mutant FUS exhibits a restricted capacity to suppress the expression of hnRNP K and MP20. (A)** Comparison of sequences downstream of the FL-stop in exon 6 529–585 in *TARDBP* between humans and mice. Annotated with FL-stop (magenta), GU-rich (light blue), and AU-rich region (purple), respectively. Magenta lines and square box indicate the FUS-binding consensus reported previously. **(B)** Schematic representation of the FUS with

P525L mutation. **(C)** RT-qPCR analysis of relative mRNA expression of MP20 (127), TDP-FL, and *hnRNP K* in HEK293T cells overexpressing FLAG-FUS. Data are normalized to *ACTB*. Statistical analyses were performed using one-way ANOVA followed by Tukey's test (*n* = 3 for each group). **(D)** WB of HEK293T cells overexpressing FLAG-FUS. Quantification of relative expression of hnRNP K (*n* = 3 for each group). **(E)** Immunocytochemistry of HeLaS3 cells overexpressing FLAG-FUS. HnRNP K (green), FLAG (magenta), and Hoechst (blue). Arrowheads or arrows indicate FLAG-positive or FLAG-negative cells, respectively. Scale bar: 10 µm. Quantification of the fluorescence intensity of nuclear hnRNP K in FLAG-FUS–positive cells (right). **(F)** WB of HEK293T cells overexpressing FLAG-FUS. Quantification of relative expression of MP20-endo (*n* = 3 for each group). In D and F, data are normalized to β-actin. Statistical analyses were performed using one-way ANOVA followed by Tukey's test. **(G)** Immunocytochemistry of HeLaS3 cells overexpressing FLAG-FUS. MP20 (green), FLAG (magenta), and Hoechst (blue). Arrowheads or arrows indicate FLAG-positive or FLAG-negative cells, respectively. Scale bar: 10 µm. Quantification of the fluorescence intensity of nuclear MP20 in FLAG-FUS–positive cells (right). In E and G, a total of 100 FLAG-positive and 100 FLAG-negative cells per group were analyzed. Fluorescence signals were normalized to the average intensity of 200 FLAG-negative cells. Statistical analysis was performed using Welch's *t* test. **(H)** Comparison of sequences downstream of the MP20 stop codon in exon 6 1,198–1,246 in *TARDBP* between humans and mice. Annotated with MP20 stop codon (magenta) and GU-rich (light blue), respectively. The magenta lines and square box indicate the FUS-binding consensus reported previously. All graphs show the mean ± SEM. *P < 0.05, **P < 0.01, ***P < 0.001, and ****P < 0.0001. Ab, antibody; RMM, RNA recognition motif; ZnF, zinc finger; NES, nuclear export signal; NLS, nuclear localization signal; QSYG, glutamine-glycine-serine-tyrosine-rich domain. Source data are available for this figure: SourceData F9.

deletion (Fig. S4, A–D) and by the trend of increased MP20 RNA and protein levels observed following hnRNP K KD (Fig. 3 G and Fig. 5 E). Nuclear hnRNP K likely acts through other protein binding to *TARDBP* RNA, potentially by suppressing MP20 splicing promoters and/or enhancing MP20 splicing suppressors. However, when hnRNP K translocates to the cytoplasm, its influence on these factors may be diminished, resulting in enhanced MP20 splicing and subsequent MP20 expression. Indeed, hnRNP K cytoplasmic mislocalization has been reported in FTLD, Alzheimer's disease, and C9orf72-related ALS (Braems et al., 2022; Sidhu et al., 2022), suggesting a link to TDP-43 functional impairment in these neurodegenerative diseases.

hnRNP A1–mediated splicing promotes sMP13 production by recognizing the 5′ splice site (ss) of intron 5 (Fig. S2 H). While hnRNP K–mediated splicing instead utilizes the 5′ ss of cryptic intron 6, coexpression with hnRNP A1 redirected this to the 5′ ss of intron 5, suggesting that hnRNP A1 facilitates its recognition as a donor site (Fig. S2, I–K). Notably, hnRNP A1 Rmt did not induce this shift, indicating that the RNA-binding capacity of hnRNP A1 is required for this activity (Fig. 7 G). Unexpectedly, no specific regulatory sequences for hnRNP A1 were identified within *TARDBP* intron 5 or cryptic intron 6 (Fig. S3). As hnRNP A1 is known to act not only on introns but also on exonic silencers (Pan et al., 2024; Zhu et al., 2001) and to interact with splice sites on target RNAs (David et al., 2010; Liu et al., 2020; Oh et al., 2013), it is possible that *TARDBP* exon 5 or exon 6, or their respective splice sites, may serve as regulatory elements. If hnRNP A1 does not act through direct binding to *TARDBP* RNA, an alternative possibility is that hnRNP A1 may exert its effect on *TARDBP* RNA via a protein complex associated with another RNA.

A recent study suggested the importance of splicing regulation by multiple RBPs in neuronal homeostasis, demonstrating that an ALS-associated synaptic ras GTPase activating protein 1 (*SYNGAP1*) 3′ UTR variant recruits excessive FUS and hnRNP K, with hnRNP K driving aberrant splicing that *SYNGAP1* isoform α1 over γ, leading to dendritic spine loss (Yokoi et al., 2022). Our study demonstrated that FUS interacts with *TARDBP* RNA exon 6 (Fig. S5, A–C) and downregulates both RNA and protein levels of hnRNP K (Fig. 9, C–E). The GU-rich sequence downstream of the FL-stop, containing FUS-binding sites (Fig. 9 A) (Iko et al., 2004; Lagier-Tourenne et al., 2012; Lerga et al., 2001), appears to

influence hnRNP K sensitivity in *TARDBP* exon 6 splicing (Fig. S4, E and F). These suggest that FUS may regulate hnRNP K activity through *TARDBP* RNA interaction as part of its nuclear function.

In ALS, cytoplasmic FUS leakage is linked to loss of nuclear function, while its aggregation suggests gain of toxic function (An et al., 2019; Scekic-Zahirovic et al., 2016). Our data also showed that the P525L mutant FUS exhibited reduced nuclear localization and formed aggregates in the cytoplasm (Fig. 9, E and G). Loss of nuclear FUS likely impairs its ability to regulate hnRNP K activity via *TARDBP* RNA interaction, which could explain the failure of P525L FUS to suppress nuclear hnRNP K expression (Fig. 9 E).

As for cytoplasmic FUS, while WT FUS forms stress granules containing GC-rich RNA under stress, mutant FUS stress granules are enriched with unstructured AU-rich RNA (Mariani et al., 2024). Although we observed no significant difference in translational inhibition of MP20 between WT and P525L mutant FUS (Fig. S5, E–G), the altered RNA-binding properties of the mutant may influence FUS-mediated translational control.

Our findings suggest that FUS plays a multifaceted role in regulating TDP-43, providing insights into its direct effects on TDP-43 and the complex mechanism involving other RBPs (Fig. 10, A–C).

Aging has been shown to induce changes in splicing events (Angarola and Anczuków, 2021), with transcriptome analyses revealing numerous age-related alternative splicing events in the human brain (Mazin et al., 2013). These events are observed in 95% of FTLD and Alzheimer's disease patients regardless of age, alongside disease-specific splicing changes absent in healthy individuals (Tollervey et al., 2011b). In mice, aging reduces TDP-FL transcript levels and increases MP18 levels (Weskamp et al., 2020), suggesting that a similar shift may occur in humans. Interestingly, while MP18 predominates in mice (Weskamp et al., 2020) (Fig. 4 E), humans exhibit additional short isoforms, reflecting an evolutionarily complex splicing machinery. This acquired complexity may enable stricter regulation of TDP-43 activity but also increase the risk of dysregulation.

Loss of hnRNPs has been observed in postmortem tissues from patients with neurodegenerative diseases. Disruption of hnRNP K expression has been reported in spinal motor neurons in both familial ALS and sporadic ALS patients (Moujalled et al.,

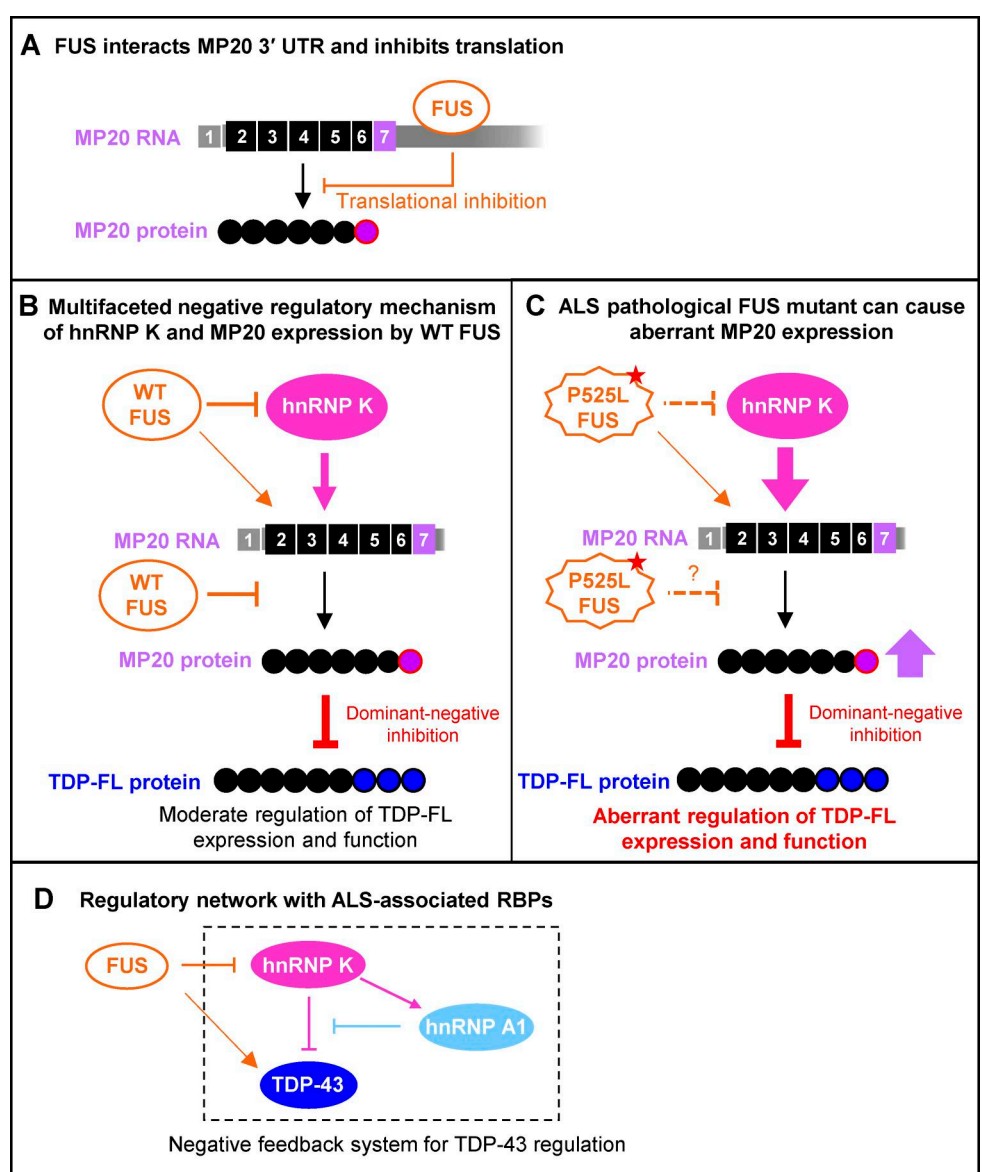

Figure 10. **Schematic representation of the negative feedback system regulating TDP-43 function through FUS, hnRNP K, and hnRNP A1. (A)** The binding consensus of FUS is widely present in *TARDBP* exon 6, including the 3′ UTR of MP20. FUS may inhibit the translation of MP20 by interacting with *TARDBP* RNA. **(B)** HnRNP K promotes splicing to MP20 while inhibiting FL, leading to the induction of MP20 expression with dominant-negative activity. WT FUS inhibits MP20 expression via multifaceted mechanism: by inhibiting hnRNP K expression and directly suppressing the posttranscriptional process of MP20. FUS seems to promote the increase of MP20 RNA levels, although its exact mechanism remains unclear. **(C)** The ALS-causing mutant FUS (P525L) not only increases MP20 RNA but also leads to the dysrepression of MP20 at each regulatory point, potentially affecting TDP-FL expression and function due to its aberrant dominant-negative activity. **(D)** Our study indicates a negative feedback mechanism in which hnRNP A1 is induced to counteract the effects of hnRNP K, which inhibits canonical TDP-43. These findings suggest that abnormalities in any of the RBPs involved in this regulatory system could result in dysfunction of TDP-43.

2015). Dysregulation of hnRNP A1, hnRNP A2B1, and hnRNP K has also been noted in tauopathies, with significant cellular mislocalization and unique pattern observed for each hnRNP (Kavanagh et al., 2024). Additionally, nuclear loss of hnRNP D, hnRNP C, and hnRNP A1 has been observed in the frontal cortex of FTLD (Pinkerton et al., 2024). These findings highlight the significance of hnRNPs regulated networks in the pathogenesis of neurodegenerative diseases. This study elucidates a part of the TDP-43 regulatory mechanism involving hnRNP K, hnRNP A1, and FUS (previously known as hnRNP P2). In addition, hnRNP K

and hnRNP A1 could participate in a negative feedback system to correct TDP-43–related abnormalities (Fig. 10 D).

In summary, we demonstrated the existence of the dominant-negative TDP-43 isoform, which can physiologically regulate canonical TDP-43 activity. This regulatory process appears to be mediated by multiple RBPs linked to ALS. Our study provides a molecular model illustrating how these RBPs form a regulatory network to control TDP-43. Our findings also suggest that abnormalities in any RBP involved in TDP-43 regulation could disrupt this network, potentially leading to TDP-43 dysfunction

and contributing to the pathology observed in most ALS cases. Furthermore, the regulatory process described may become uncontrollable during the decades preceding ALS onset, leading to an irreversible vicious cycle and offering important insights for future ALS research.

## Materials and methods

### Cell culture and transfection
HEK293T and HeLaS3 cells were obtained from ATCC, and SH-SY5Y (EC94030304-F0) cells were purchased from KAC Co., Ltd. The cells were cultured in DMEM (Sigma-Aldrich) supplemented with 10% FBS (GE Healthcare), 1% penicillin–streptomycin (FU-JIFILM), and MEM nonessential amino acids (Thermo Fisher Scientific). Cells were transfected with 0.5–4 µg of plasmid DNA or 60 pmol of siRNA for 48–72 h using Lipofectamine 3000 or RNAiMAX (Thermo Fisher Scientific) following the manufacturer's instructions. siRNAs targeting TDP-FL (Hasegawa-Ogawa and Okano, 2021) and hnRNP A1 (Li et al., 2021), were synthesized by Hokkaido System Science. siRNAs targeting hnRNP K and a mock control were purchased from OriGene (SR302173 and SR30004, respectively). Human-induced pluripotent stem cells (hiPSCs) were purchased from the Cell Bank and cultured in StemFit medium (Ajinomoto).

### Ethical approval for animal experiments
All animal experiments in this study were approved by the Institutional Animal Care and Use Committee of the Jikei University. Total RNA, protein extracts, and frozen sections for staining were obtained from the brain cortex and spinal cord of C57BL/6N mice.

### Genomic DNA extraction
Human genomic DNA was extracted from hiPSCs using a QIAamp DNA Mini Kit (QIAGEN) following the manufacturer's protocols.

### Plasmid vector construction
Shortened TDP-43 isoforms, FUS, and hnRNP A1 were cloned from HEK293T cells using primers with restriction sites and PrimeSTAR MAX DNA polymerase (Takara) according to the manufacturer's instructions. Similarly, ELAVL3, NOVA1, hnRNP E1, and hnRNP E2 were cloned from SH-SY5Y cells. The cloned DNA fragments were inserted into the pFLAG-CMV2 vector (Sigma-Aldrich). A plasmid containing hnRNP K in the pFLAG-CMV2 vector was kindly provided by Dr. Yano, Jikei University School of Medicine, Tokyo, Japan. FLAG-TDP-FL and TDP-FL-Venus plasmids were constructed in a previous study (Hasegawa-Ogawa and Okano, 2021). To construct the FLAG-tagged TARDBP mini-gene, TARDBP exon 6 (1–2,100) or exon 5 to exon 6 (1–2,100) was cloned from hiPSCs. A DNA fragment encoding a FLAG-tag with a restriction site was amplified from the pFLAG-CMV2 vector. Both the FLAG-tag and TARDBP fragments were inserted into the pcDNA3 vector (Thermo Fisher Scientific). For Venus-MP20 3′UTR construction, Venus and TARDBP exon 6 (1,205–1,940) fragments were amplified from the TDP-FL-Venus construct or TARDBP mini-gene

(Ex6), respectively, using KOD Fx Neo (TOYOBO) according to the manufacturer's instructions. Constructs with deletions or point mutations of the FLAG-hnRNP K, FLAG-hnRNP A1, FLAG-FUS, and TARDBP mini-genes were generated using PrimeSTAR MAX DNA polymerase following the manufacturer's protocols. Primer sets used for construction are listed in Table S3.

### RT-PCR and RT-qPCR
Total RNA was extracted from cells and tissues using TRIzol reagent (Thermo Fisher Scientific) and an RNeasy Plus Mini Kit (QIAGEN), following the manufacturer's instructions. cDNA synthesis was performed using ReverTra Ace -α- (TOYOBO) or ReverTra Ace qPCR RT Master Mix (TOYOBO). Semiquantitative PCR was conducted using EmeraldAmp PCR Master Mix (Takara) on a C1000 Touch Thermal Cycler (Bio-Rad). For qPCR, reactions were performed using TaqMan Fast Advanced Master Mix for qPCR (Thermo Fisher Scientific) or PowerTrack SYBR Green Master Mix for qPCR (Thermo Fisher Scientific) on a Quant Studio 5 Real-Time PCR System (Thermo Fisher Scientific) according to the manufacturer's instructions. TaqMan probes and primers specific for TDP-FL and MP20 (127) were custom-designed and synthesized (Thermo Fisher Scientific). The TDP-FL probe set targeted a unique 3′UTR sequence absent in MPs, while MP20 (127) probe targeted the junction between exon 6 (127 nt) and the spliced exon 7 (Fig. 3 A).

For mRNA quantifications of hnRNP K, hnRNP A1, and ACTB, we used predesigned TaqMan Gene Expression Assays (Thermo Fisher Scientific). For GPSM2 and ATG4B CE, we utilized the SYBR Green system. All qPCR results were normalized to ACTB. Primer sets and TaqMan probes used in this study are listed in Tables S1 and S2, respectively.

### Band sequencing of RT-PCR products
The RT-PCR bands were subjected to gel extraction using a NucleoSpin Gel and PCR Clean-up Kit (MACHEREY-NAGEL) and then inserted into the pCR2.1 plasmid vector (Thermo Fisher Scientific). After the selection of blue and white Escherichia coli colonies, the plasmid DNA from the white colonies was sequenced using a BigDye Terminator v3.1 Cycle Sequencing Kit (Thermo Fisher Scientific) following the manufacturer's instructions.

### Protein extraction and western blotting
Protein lysates were prepared from cultured cells 48 h after plasmid or siRNA transfection, or from mouse tissues using Tissue Extraction Reagent I (Thermo Fisher Scientific) with Protease Inhibitor Cocktail (Roche) and Phosphatase Inhibitor Cocktail (Nacalai). Lysates were sonicated for 20 s and centrifuged at 15,000 rpm for 5 min at 0°C. The supernatant was mixed with 2× Laemmli sample buffer (Bio-Rad) containing 5% β-mercaptoethanol and boiled at 95°C for 3 min and separated by SDS-PAGE using a TGX FastCast Acrylamide Kit, 10% (Bio-Rad). Proteins were transferred to polyvinylidene difluoride (PVDF) membranes, blocked with 1% nonfat skim milk/Tris-buffered saline with Tween 20 (TBST) or Blocking One (Nacalai), and incubated with primary antibodies overnight (ON) at 4°C. Primary antibodies used were anti–DDDDK-tag (for FLAG-tag

detection) (1:5,000, M185-3L; MBL), anti–TDP-43 (1:1,000, 10782-2-AP; Proteintech), anti-hnRNP A1 (1:1,000, RN114PW; MBL), anti-hnRNP A2/B1 (1:1,000, sc-374053; Santa Cruz), anti-hnRNP K (1:1,000, RN019P; MBL), anti-MP20 (self-produced, 1:500), anti-FUS (1:1,000, sc-47711; Santa Cruz), anti-GFP (1:1,000, 598; MLB or 1:1,000, A11120; Thermo Fisher Scientific), anti-GAPDH (1:1,000, #5174; Cell Signaling), and anti–β-Actin (1:4,000, A1978; Sigma-Aldrich). HRP-conjugated secondary antibodies (1:2,000, AP307P for rabbit IgG or AP308P for mouse IgG; Millipore) were applied for 2 h at RT. Signals were detected by Western BLoT Chemiluminescence HRP Substrate (Takara) and imaged with LuminoGraph I (ATTO) using ImageSaver6 software.

## IP experiments

Primary antibodies were added to extracted protein samples: anti–DDDDK-tag (for FLAG-tag detection, M185-3L; MBL, 2 μg/sample), anti-MP20 (self-produced, 2.7 μg/sample), anti-FUS (sc-47711; Santa Cruz, 2.5 μg/sample), normal rabbit IgG (#2729; Cell Signaling, 2.7 μg/sample), and mouse (G3A1) mAb IgG1 Isotype Control (#5415; Cell Signaling, 2.5 μg/sample). Samples were incubated at 4°C for 3 h (anti-DDDDK) or ON (anti-MP20 and normal rabbit IgG) with rotation. Protein G Mag Sepharose beads (GE Healthcare) were added and incubated for 2–4 h at 4°C. The beads were washed three times with low-salt buffer and once with high-salt buffer, prepared according to the Simple ChIP Chromatin IP Buffers protocol (Cell Signaling). After washing, bound proteins were eluted by mixing the beads with 2× Laemmli sample buffer containing 5% β-mercaptoethanol, followed by heating at 95°C for 3 min. WB was performed as described above, using HRP-conjugated secondary antibodies (ab131368; Abcam, 1:500 for mouse IgG, AP307P for rabbit IgG) or HRP-conjugated VeriBlot for IP Detection Reagent (1:2,000, ab131366; Abcam).

## UV cross-linking IP and RT-PCR

HEK293T cells were cultured in a 10-cm dish and subjected to UV cross-linking at 400 J/cm². Protein lysates were prepared using Tissue Extraction Reagent I (Thermo Fisher Scientific) supplemented with protease (Roche) and phosphatase inhibitors (Nacalai). IP was performed as described above using anti-FUS antibody (Santa Cruz) or mouse mAb IgG1 Isotype Control (Cell Signaling). Following IP, bound protein was digested with Protease K (Takara), and total RNA was extracted using TRIzol reagent (Thermo Fisher Scientific) and RNeasy Plus Mini Kit (QIAGEN). Extracted RNA was subjected to RT-PCR. The primer sets for *TARDBP* detection are listed in Table S1.

## Generation of antibodies recognizing shortened TDP-43 isoforms

Generation of polyclonal antibodies against the unique C-terminal peptides of MP20 and affinity column purification was contracted to Scrum Inc. Rabbits were immunized four times with a synthetic peptide (ILSTCFLIQEFVITHHRPRL) conjugated N-terminal Cys bound to the carrier protein keyhole-limpet hemocyanin. Sera were affinity purified using the same immunization peptide. The specificity of the antibody was evaluated by confirming the recognition of overexpressed MP20 and a decrease in the signal of this antibody by unique C-terminal peptide adsorption (Fig. 4, A–C). In addition, we demonstrated that FLAG-MP20 immunoprecipitated with this antibody was positive for TDP-43 (against the N-terminal 1–260 amino acids), FLAG, and MP20 antibody (Fig. 4 D).

## Immunocytochemistry

Cells were fixed with 4% PFA/PBS for 10 min at 4°C, followed by blocking and permeabilization with 5% BSA/PBS containing 0.3% Triton X-100 for 1 h at RT. Primary antibodies were incubated ON at 4°C: anti–DDDDK-tag (for FLAG-tag detection) (1:1,000, M185-3L; MBL), anti–TDP-43 (G400) (1:50, #3448; CST), anti-MP20 (self-produced, 1:25), anti-hnRNP K (1:1,000, RN019P; MBL), anti-hnRNP A1 (1:200, RN114PW; MBL or 1:400, 67844-1-Ig; Proteintech), and anti-hnRNP A2/B1 (1:200, sc-374053; Santa Cruz). Secondary antibodies (1:200; Thermo Fisher Scientific) and Hoechst 33342 (1:1,000, H3570; Thermo Fisher Scientific) were applied for 1 h at RT. Secondary antibodies included Alexa Fluor goat anti-rabbit IgG 488 (A11034), Alexa Fluor goat anti-mouse IgG2b 488 (A21141), Alexa Fluor donkey anti-mouse IgG 594 (A21203), Alexa Fluor goat anti-mouse IgG2a 594 (A21135), Alexa Fluor goat anti-mouse IgG2b 594 (A21145), and Alexa Fluor goat anti-mouse IgG 633 (A21050). Images were acquired using an LSM880 confocal laser microscope (Zeiss) equipped with Plan-Apochromat 20×/0.8 M27, Plan-Apochromat 63×/1.4 Oil DIC M27, or alpha Plan-Apochromat 100×/1.46 Oil DIC M27 objective lenses and ZEN software (Zeiss) (Fig. 5 A; Fig. 6 B; Fig. 9, E and G; Fig. S1, B and J; and Fig. S2 E). Additional images were obtained using an IX73 fluorescence microscope (Olympus) equipped with a UPlanSApo 40×/0.95 objective lens and Metaview software (Olympus) (Fig. 1 C; and Fig. 4, B and C).

## Analysis of fluorescence images

Fluorescent images were analyzed using ImageJ software (Rueden et al., 2017). In the present study, nuclear fluorescence intensities of MP20, hnRNP K, hnRNP A1, and hnRNP A2/B1 were measured (Fig. 5, B and C; Fig. 6, C and D; Fig. 9, E and G; and Fig. S1 J). Nuclear regions of interest were defined based on Hoechst staining using ROI Manager. Mean fluorescence intensities in FLAG-positive and -negative cells were measured and compared.

## Immunofluorescent staining

Mice were perfusion fixed with 4% PFA, and frozen sections were prepared. Antigen retrieval was performed by autoclaving in PBS at 105°C for 5 min. Sections were then blocked and permeabilized in 5% BSA/PBS containing 0.3% Triton X-100 for 1 h at RT. Primary antibodies included anti-MAP2 (1:200, M4403; Sigma-Aldrich) and anti-MP20 (self-produced, 1:25), incubated ON at 4°C. Secondary antibodies (1:200; Thermo Fisher Scientific) and Hoechst 33342 (1:1,000, H3570; Thermo Fisher Scientific) were applied for 2 h at RT. Secondary antibodies included Alexa Fluor goat anti-rabbit IgG 488 (A11034) and Alexa Fluor goat anti-mouse IgG1 594 (A21125). Samples were mounted with Fluoromount (Diagnostic BioSystems) and imaged using an

LSM880 confocal laser microscope (Zeiss) equipped with Plan-Apochromat 20×/0.8 M27 or alpha Plan-Apochromat 100×/1.46 Oil DIC M27 objective lenses, and images were acquired using ZEN software (Zeiss).

### Identification of potential hnRNP A1– and hnRNP K–binding motifs

hnRNP A1 is known to selectively bind the consensus sequence 5′-UAGGGA/U-3′ (Burd and Dreyfuss, 1994; Ishikawa et al., 1993), and more recent studies suggest its preference for single-stranded elements within stable secondary structures (Jain et al., 2017; Levengood and Tolbert, 2019; Morgan et al., 2015). Previous studies have also shown that hnRNP K preferentially recognizes the consensus sequence 5′-GCCCA-3′ and tends to bind within stem-loop structures (Dominguez et al., 2018; Van Nostrand et al., 2020). To assess the presence of the hnRNP A1– and hnRNP K–binding motifs, secondary structure prediction was performed using the UNAFold software package (Markham and Zuker, 2008) on a frame with a length of 300 nt's. The frame was iteratively moved from the 5′ end at increments of three nt's until it reached the 3′ end. Then, the existence probabilities (PGlobal) of each predicted structure were calculated based on the free energies according to the Maxwell–Boltzmann statistics, as previously described (Morishita, 2023; Nakamura et al., 2019). All structural motifs with PGlobal values above 50% were examined for the presence of the abovementioned hnRNP A1 and hnRNP K consensus sequences in loops and stems, respectively.

### Statistical analyses

All the statistical analyses were performed using GraphPad Prism 9.2.0 software. The level of significance was set to 0.05, and Welch's $t$ test (1 variable, 2 groups) and one-way ANOVA (1 variable, >2 groups) were used. Analysis after ANOVA was subsequently performed utilizing Tukey's test or Dunnett's test, and the details are provided in the figure legends.

### Online supplemental material

Fig. S1 presents the effect of hnRNP A1 downregulation on MP20 expression. Fig. S2 shows that the hnRNP K levels are not affected by hnRNP A1 expression. Fig. S3 demonstrates that the region of hnRNP A1 responsible for *TARDBP* exon 6 splicing is not located in *TARDBP* intron 5 or cryptic intron 6. Fig. S4 presents the importance of the GC-rich sequence immediately downstream of the TDP-FL-stop in controlling sensitivity to hnRNP K. Fig. S5 demonstrates that the FUS interacts with *TARDBP* RNA and regulates translation through the MP20 3′UTR. Table S1 is a list of PCR primer sets used in this study (for RT-PCR). Table S2 is a list of TaqMan Probes and primer sets used in this study (for RT-qPCR). Table S3 is a list of PCR primers used in this study (for plasmid vector construction).

### Data availability

All data generated or analyzed during this study are included in this published article and its supplementary information files.

## Acknowledgments

We express our gratitude to Dr. Shingo Nakamura and Dr. Ella Morishita for their efforts in predicting potential consensus sequences of hnRNP K and A1. Additionally, we appreciate the guidance provided by Prof. Shushi Nagamori and Prof. Yumi Kanegae from the Advisory Board for Mass Spectrometry at Core Research Facilities for Basic Science in Center for Medical Science, The Jikei University School of Medicine, regarding the biochemical analysis.

This work was supported by Japan Society for the Promotion of Science KAKENHI Grant Numbers JP16H07218, JP17K09766, JP17K14967, JP20K15904, JP21K07302, and JP23K06812; The Uehara Memorial Foundation; and The Jikei University Graduate Research Fund. Open Access funding provided by The Jikei University School of Medicine.

Author contributions: M. Hasegawa-Ogawa: conceptualization, data curation, formal analysis, funding acquisition, investigation, methodology, project administration, supervision, validation, visualization, and writing—original draft, review, and editing. A. Onda-Ohto: investigation and methodology. T. Nakajo: investigation and methodology. A. Funabashi: investigation and methodology. A. Ohya: investigation and methodology. R. Yazaki: investigation and methodology. H.J. Okano: conceptualization, data curation, funding acquisition, project administration, supervision, validation, visualization, and writing—original draft, review, and editing.

Disclosures: The authors declare no competing interests exist.

Submitted: 17 June 2024

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

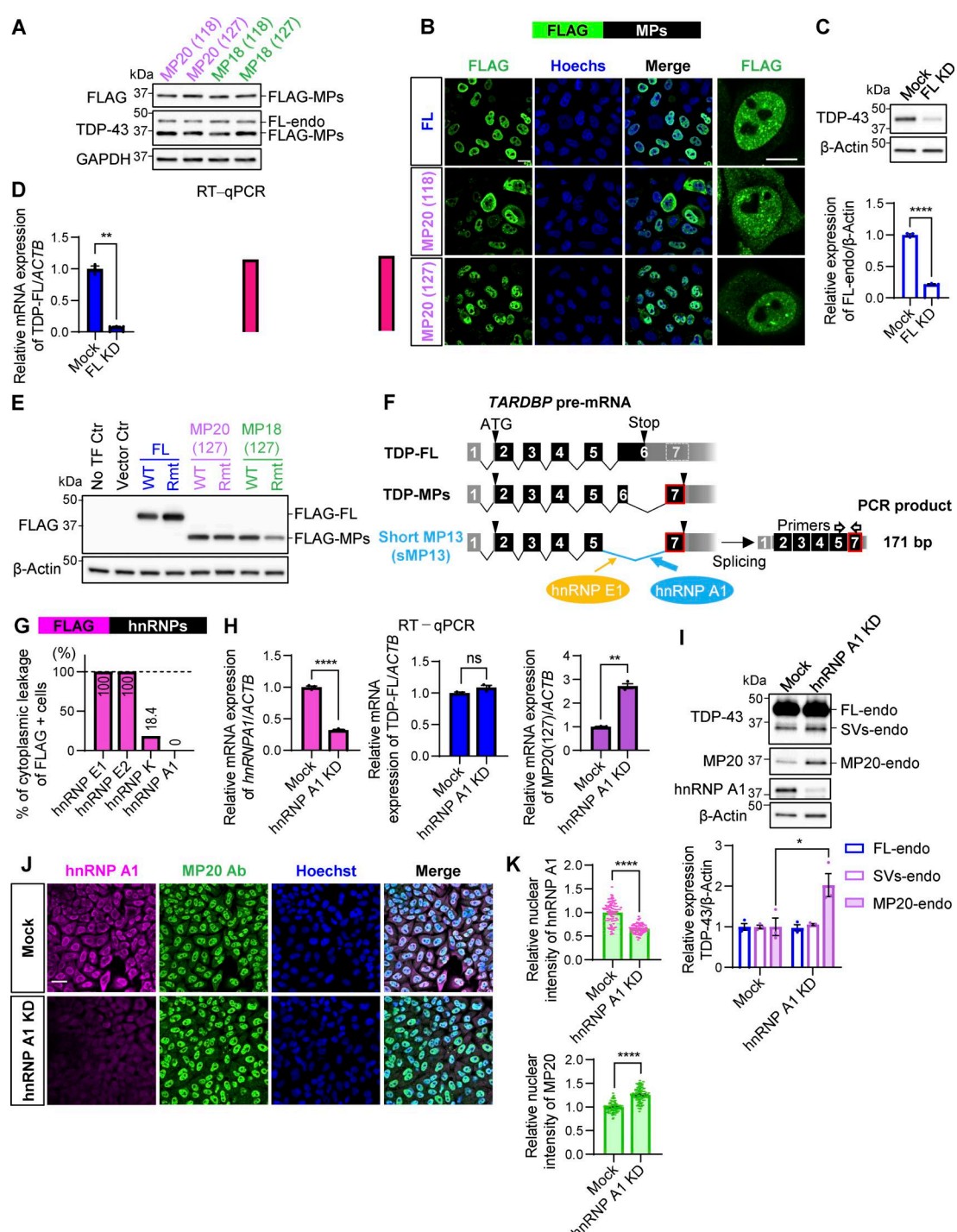

Figure S1. **hnRNP A1 downregulation promotes MP20 expression. (A)** WB analysis confirming the expression of FLAG-MPs in HEK293T cells. The anti–TDP-43 antibody detected endogenous TDP-FL (FL-endo) and FLAG-MPs. **(B)** Subcellular localization of FLAG-MPs in HeLaS3 cells. FLAG (green) and Hoechst (blue). Magnified images of FLAG are shown in the right panels. Scale bars: 20 μm (left) and 10 μm (right). **(C)** WB analysis of TDP-43 protein levels in HEK293T with TDP-FL KD (FL KD). Data are normalized to β-actin. Statistical analysis was performed using Welch's t test. **(D)** RT-qPCR analysis of TDP-FL, *GPSM2 CE*, and *ATG4B CE* mRNA levels in HEK293T with FL KD. Data are normalized to *ACTB*. Statistical analysis was performed using Welch's t test (n = 3 for each group). **(E)** WB analysis of FLAG-MPs with Rmt's (see Fig. 2 E). **(F)** Schematic representation of the sMP13 isoform of *TARDBP*, induced by hnRNP A1 and E1. **(G)** Percentage of cytoplasmic leakage in FLAG-hnRNPs–positive cells. Data present 564 cells per group. **(H)** RT-qPCR analysis of *hnRNP A1*, TDP-FL, and MP20 (127) mRNA levels in HEK293T cells with hnRNP A1 KD. Data are normalized to *ACTB*. Statistical analyses were performed using Welch's t test (n = 3 for each group). **(I)** WB analysis of TDP-43 protein levels in HEK293T cells with hnRNP A1 KD. Quantification of FL-endo, SVs-endo, and MP20-endo protein levels is normalized to β-actin. Statistical analyses were performed using Welch's t test. **(J)** Immunocytochemistry of HeLaS3 cells with hnRNP A1 KD. hnRNP A1 (magenta), MP20 Ab (green), and Hoechst (blue). Scale bar: 50 μm. **(K)** Quantification of nuclear hnRNP A1 or MP20 intensity. Data present 100 cells per group. Statistical analyses were performed using Welch's t test. All graphs show the mean ± SEM. *P < 0.05, **P < 0.01, ***<0.001, and ****P < 0.0001. Ab, antibody. Source data are available for this figure: SourceData FS1.

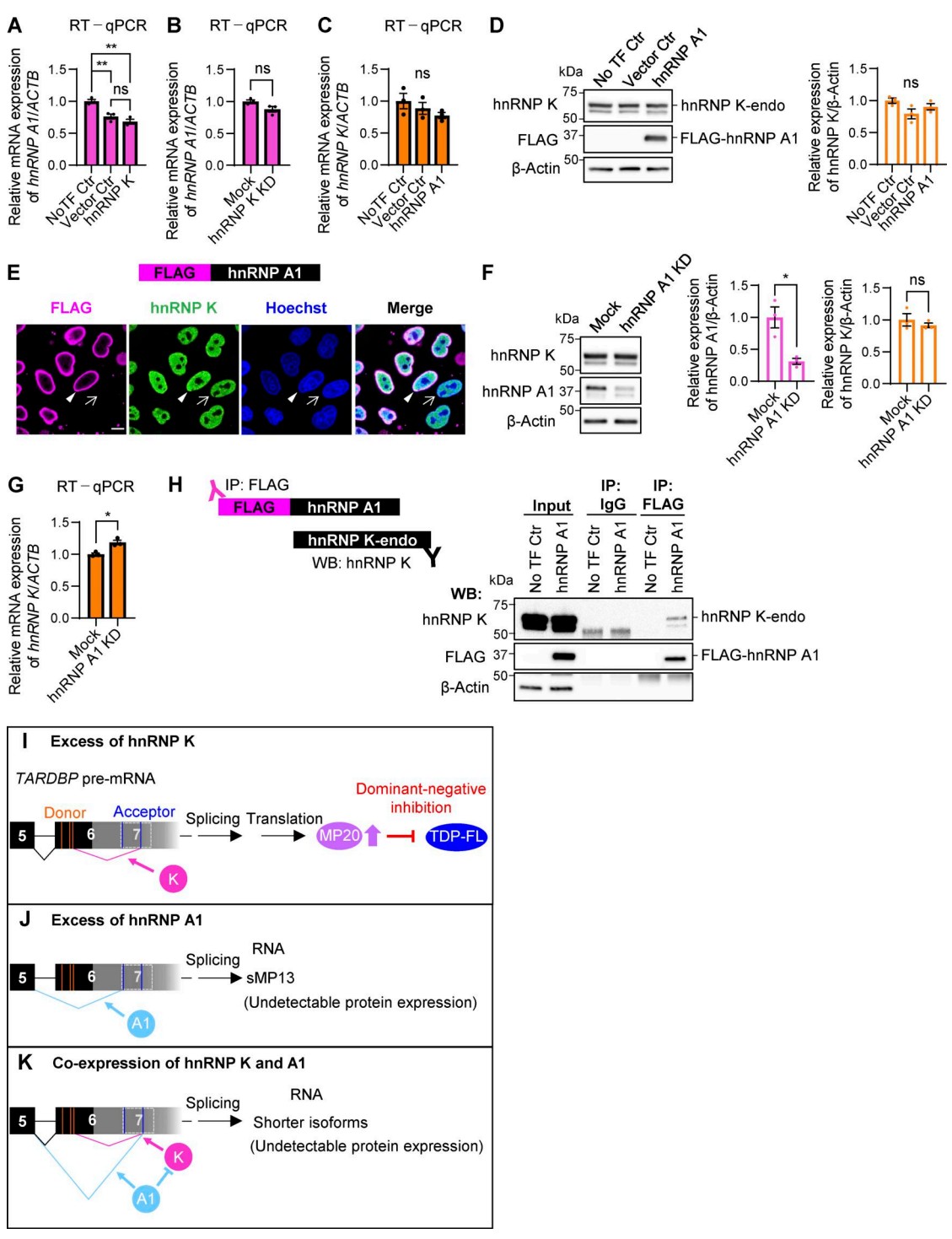

Figure S2. **hnRNP K levels are not influenced by hnRNP A1 expression. (A–C)** RT-qPCR analysis of *hnRNP A1* (A and B) and *hnRNP K* (C) mRNA levels in HEK293T cells overexpressing FLAG-hnRNP K (A) or with hnRNP K KD (B) or overexpressing hnRNP A1 (C). Data are normalized to *ACTB*. Statistical analyses were performed using one-way ANOVA followed by Tukey's test (A and C) or Welch's *t* test (B), respectively (*n* = 3 for each group). **(D)** WB analysis of HEK293T cells overexpressing FLAG-hnRNP A1. Quantification of hnRNP K is normalized to β-actin. Statistical analysis was performed using one-way ANOVA followed by Tukey's test (*n* = 3 for each group). **(E)** Immunocytochemistry of HeLaS3 cells overexpressing FLAG-hnRNP K. FLAG (magenta), hnRNP A1 (green), and Hoechst (blue). Arrowheads and arrows indicate FLAG-positive and FLAG-negative cells, respectively. Scale bar: 10 μm. **(F)** WB analysis of HEK293T cells with hnRNP A1 KD. Quantification of hnRNP A1 and hnRNP K is normalized to β-actin. Statistical analyses were performed using Welch's *t* test (*n* = 3 for each group). **(G)** RT-qPCR analysis of *hnRNP K* mRNA levels in HEK293T cells overexpressing FLAG-hnRNP A1. Data are normalized to *ACTB*. Statistical analysis was performed using Welch's *t* test (*n* = 3 for each group). **(H)** IP of FLAG-hnRNP A1–overexpressing HEK293T cells. FLAG antibody was used for IP, and endogenous hnRNP K was detected by WB. An anti-IgG antibody was used as a negative control for IP. **(I–K)** Schematic diagram showing conditions with excess hnRNP K (I), hnRNP A1 (J), or hnRNP K coexpressed with hnRNP A1 (K). All graphs show the mean ± SEM. *P < 0.05, **P < 0.01. Source data are available for this figure: SourceData FS2.

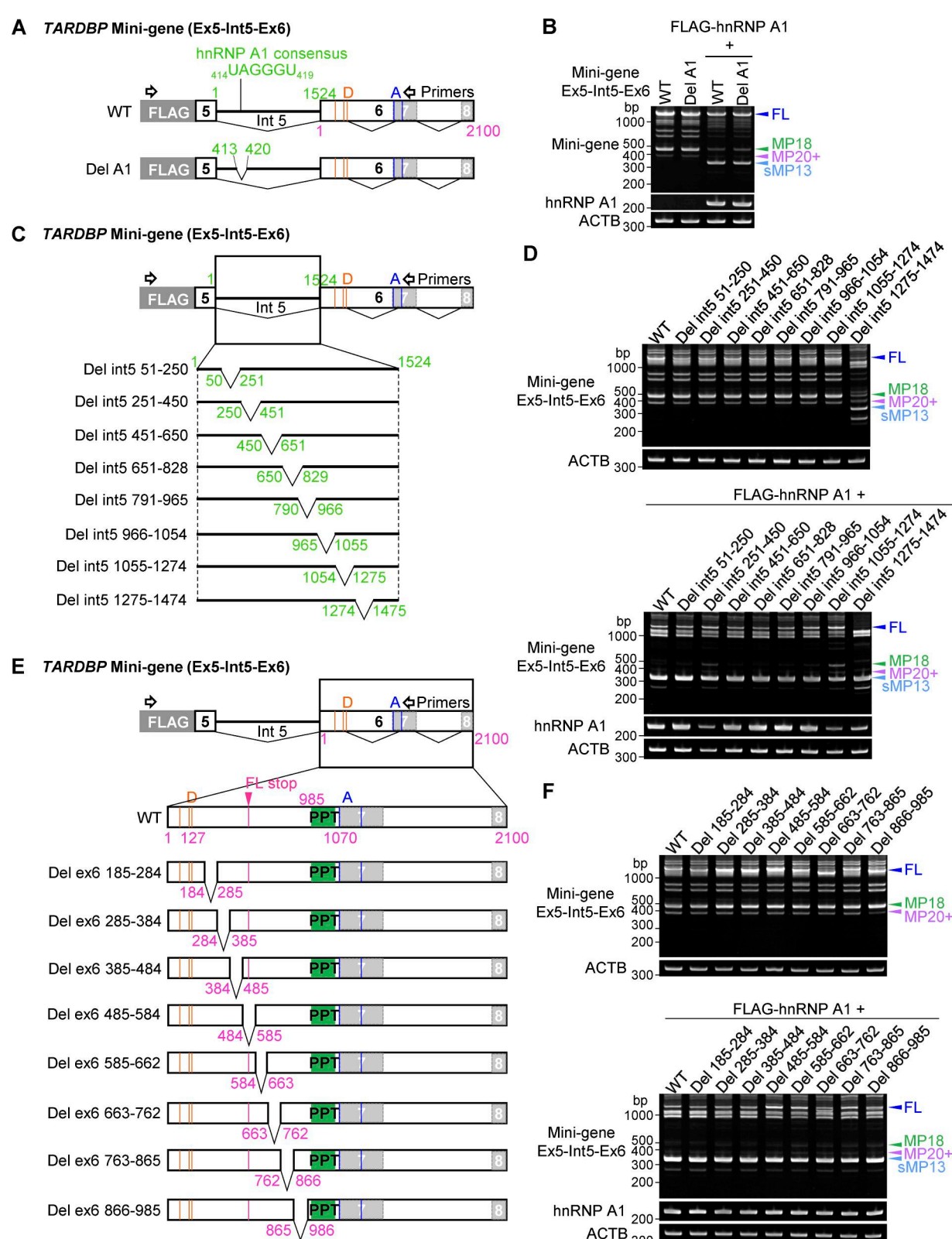

Figure S3. **Responsible region of hnRNP A1 on *TARDBP* exon 6 splicing is not found in *TARDBP* intron 5 and cryptic intron 6. (A)** Schematic representation of a mini-gene containing exon 6 (1–2,100) (Ex6) or exon 5 to exon 6 (Ex5-Int5-Ex6) of *TARDBP*. A potential hnRNP A1 consensus site is indicated in the Ex5-Int5-Ex6 mini-gene. Orange or blue lines show alternative donor (D) or acceptor (A) sites, respectively, and magenta lines indicate the stop codon of FL. **(B)** RT-PCR images showing the effect of hnRNP A1 on splicing of the mini-gene (Ex5-Int5-Ex6) with deletion of the potential hnRNP A1 consensus sequence in HEK293T cells. **(C and E)** Schematic representation of deletion mutants of the *TARDBP* mini-gene (Ex5-Int5-Ex6). Green-filled box indicates polypyrimidine tract (PPT). **(D and F)** RT-PCR images of HEK293T cells coexpressing *TARDBP* mini-genes (Ex5-Int5-Ex6) with or without hnRNP A1. Source data are available for this figure: SourceData FS3.

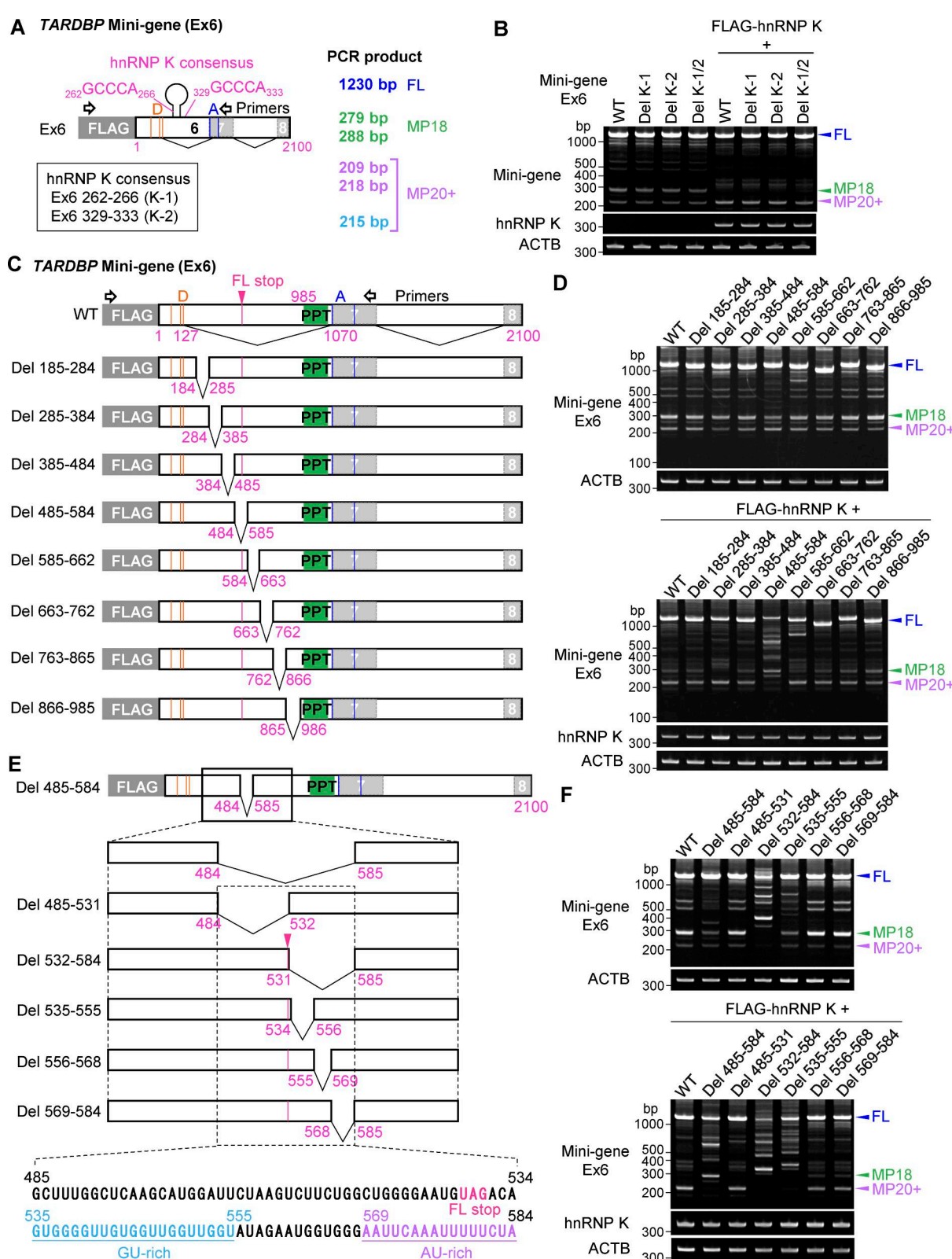

Figure S4. **The GC-rich sequence immediately downstream of the TDP-FL-stop in *TARDBP* is crucial for controlling sensitivity to hnRNP K.** **(A)** Schematic representation of a mini-gene containing exon 6 (1–2,100) (Ex6) of *TARDBP*. Two potential hnRNP K consensus sites (K-1 and K-2) are described. Orange or blue lines show alternative donor (D) and acceptor (A) sites, and magenta lines indicate the stop codon of FL. **(B)** RT-PCR images of the effect of hnRNP K on splicing of the mini-gene (Ex6) with deletion of the potential hnRNP K consensus sequence in HEK293T cells. **(C and E)** Schematic representation of systematic deletion mutants of the *TARDBP* mini-gene (Ex6). Green box indicates polypyrimidine tract (PPT). A dotted square in E indicates 485–584 residues in exon 6, annotated with FL-stop (magenta), GU-rich (light blue), and AU-rich (purple) residues, respectively. **(D and F)** RT-PCR images of HEK293T cells co-expressing the *TARDBP* mini-genes (Ex6) with or without hnRNP K. Source data are available for this figure: SourceData FS4.

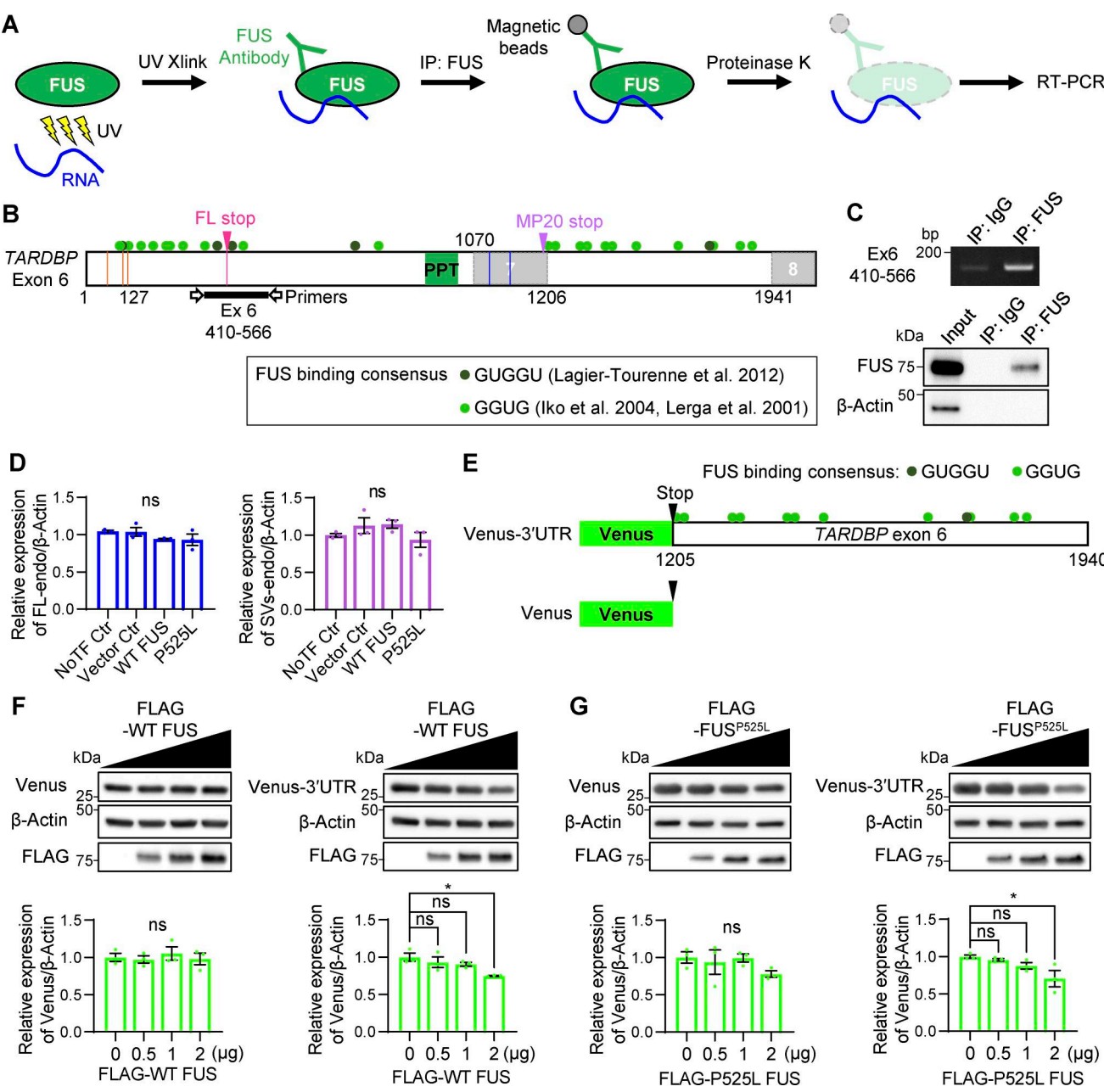

Figure S5. **FUS interacts with *TARDBP* RNA and regulates translation through the MP20 3'UTR. (A)** Schematic representation of UV cross-linking (UV Xlink) and IP followed by RT-PCR. UV Xlink was performed, and IP using an anti-FUS antibody isolated the FUS–RNA complex from HEK293T cells. Subsequent RT-PCR analysis showed whether endogenous FUS associates with *TARDBP* RNA. **(B)** Schematic diagram of FUS-binding consensus sequences on *TARDBP* exon 6. Green circles represent GGUG (Iko et al., 2004; Lerga et al., 2001), and dark green circles represent GUGGU (Lagier-Tourenne et al., 2012). Detection of *TARDBP* RNA after UV Xlink and IP was achieved using primers targeting exon 6 (410–566, black bold line). Orange and blue lines show alternative donor and acceptor sites, respectively, while magenta lines indicate the FL-stop. **(C)** UV Xlink followed by IP, and RT-PCR images showing IP validation. An anti-IgG antibody was used as a negative control for IP. **(D)** Quantification of FL-endo or SVs-endo protein levels from the WB analysis in Fig. 9 F. Data are normalized to β-actin. Statistical analyses were performed using one-way ANOVA followed by Tukey's test (*n* = 3 for each group). **(E)** Schematic diagram of the MP20 3'UTR construct. *TARDBP* exon 6 (1,205–1,940) was inserted downstream of Venus, while Venus alone was used as control construct. **(F and G)** WB analysis showing the dose-dependent effect of FLAG-WT FUS (F) or FLAG-P525L FUS mutant (G) on each Venus reporter construct. Quantification of Venus in F and G is normalized to β-actin. Statistical analyses were performed using one-way ANOVA followed by Dunnett's test. All graphs show the mean ± SEM. *P < 0.05. Source data are available for this figure: SourceData FS5.

Provided online are Table S1, Table S2, and Table S3. Table S1 is a list of PCR primer sets used in this study (for RT-PCR). Table S2 is a list of TaqMan Probes and primer sets used in this study (for RT-qPCR). Table S3 is a list of PCR primers used in this study (for plasmid vector construction).

