## [Peer Review File · The Journal of Cell Biology]

Dominant-negative isoform of TDP-43 is regulated by ALS-linked RNA-binding proteins

Minami Hasegawa-Ogawa, Asako Onda-Ohto, Takumasa Nakajo, Arisa Funabashi, Ayane Ohya, Ryota Yazaki, and Hiroataka Okano

Corresponding Author(s): Hiroataka Okano, Jikei University School of Medicine

Review Timeline:

Submission Date:	2024-06-17
Editorial Decision:	2024-09-04
Revision Received:	2025-05-02
Editorial Decision:	2025-06-12
Revision Received:	2025-06-18

Monitoring Editor: Richard Youle

Scientific Editor: Dan Simon

Transaction Report:

DOI: <https://doi.org/10.1083/jcb.202406097>

September 4, 2024

Re: JCB manuscript #202406097

Prof. Hirotaka James Okano
Jikei University School of Medicine
Division of Regenerative Medicine, Research Center for Medical Sciences
The Jikei University School of Medicine
3-25-8
Nishi-shimbashi, Minato-ku
Tokyo 1058461
Japan

Dear Prof. Okano,

Thank you for submitting your manuscript entitled "TDP-43 activity is controlled by regulatory networks involving ALS-related RNA-binding proteins." Your manuscript has been assessed by expert reviewers, whose comments are appended below. Thank you for your patience with the peer review process. Although the reviewers express potential interest in this work, significant concerns unfortunately preclude publication of the current version of the manuscript in JCB.

You will see that Reviewer #2 states that since the work was done using overexpression constructs in HeLa and HEK cells, it remains unclear whether the MP20 isoform exists in an appreciable amount in neuronal cells. We agree that this is an important question that must be addressed with new data demonstrating the significance of the MP20 isoform in neurons or in adult mouse brain. Both reviewers also ask for a few additional experiments, missing controls, quantifications, and more method details. The reviewers also noted that the presentation of the results was unclear and confusing in some places, including several instances of discrepancies between the data and the text descriptions. These points should be thoroughly and carefully addressed in a revised manuscript so that the text accurately reflects the data.

Please let us know if you are able to address the major issues outlined above and wish to submit a revised manuscript to JCB. Note that a substantial amount of additional experimental data likely would be needed to satisfactorily address the concerns of the reviewers. The typical timeframe for revisions is three to four months. If you anticipate any difficulties in meeting this aforementioned revision time limit, please contact us and we can work with you to find an appropriate time frame for resubmission. Please note that papers are generally considered through only one revision cycle, so any revised manuscript will likely be either accepted or rejected.

If you choose to revise and resubmit your manuscript, please also attend to the following editorial points. Please direct any editorial questions to the journal office.

GENERAL GUIDELINES:

Text limits: Character count is < 40,000, not including spaces. Count includes title page, abstract, introduction, results, discussion, and acknowledgments. Count does not include materials and methods, figure legends, references, tables, or supplemental legends.

Figures: Your manuscript may have up to 10 main text figures. To avoid delays in production, figures must be prepared according to the policies outlined in our Instructions to Authors, under Data Presentation, <https://jcb.rupress.org/site/misc/ifora.xhtml>. All figures in accepted manuscripts will be screened prior to publication.

*****IMPORTANT:** It is JCB policy that if requested, original data images must be made available. Failure to provide original images upon request will result in unavoidable delays in publication. Please ensure that you have access to all original microscopy and blot data images before submitting your revision. ***

Supplemental information: There are strict limits on the allowable amount of supplemental data. Your manuscript may have up to 5 supplemental figures. Up to 10 supplemental videos or flash animations are allowed. A summary of all supplemental material should appear at the end of the Materials and methods section.

Please note that JCB now requires authors to submit Source Data used to generate figures containing gels and Western blots with all revised manuscripts. This Source Data consists of fully uncropped and unprocessed images for each gel/blot displayed in the main and supplemental figures. Since your paper includes cropped gel and/or blot images, please be sure to provide one Source Data file for each figure that contains gels and/or blots along with your revised manuscript files. File names for Source Data figures should be alphanumeric without any spaces or special characters (i.e., SourceDataF#, where F# refers to the associated main figure number or SourceDataFS# for those associated with Supplementary figures). The lanes of the gels/blots

should be labeled as they are in the associated figure, the place where cropping was applied should be marked (with a box), and molecular weight/size standards should be labeled wherever possible. Source Data files will be made available to reviewers during evaluation of revised manuscripts and, if your paper is eventually published in JCB, the files will be directly linked to specific figures in the published article.

If you choose to resubmit, please include a cover letter addressing the reviewers' comments point by point. Please also highlight all changes in the text of the manuscript.

Regardless of how you choose to proceed, we hope that the comments below will prove constructive as your work progresses. We would be happy to discuss them further once you've had a chance to consider the points raised. You can contact the journal office with any questions at cellbio@rockefeller.edu.

Thank you for thinking of JCB as an appropriate place to publish your work.

Sincerely,

Richard Youle, PhD
Monitoring Editor
Journal of Cell Biology

Dan Simon, PhD
Scientific Editor
Journal of Cell Biology

Reviewer #1 (Comments to the Authors (Required)):

In this manuscript, Hasegawa-Ogawa et al. explore a network of RNA binding proteins regulating the splicing of TARDBP, a transcript and gene product that are intimately related to the pathogenesis of neurodegenerative disorders. TARDBP splice variants have been detected and described in prior studies, but remain inadequately characterized. The authors uncover an intricate interplay between several RNA binding proteins, including hnRNP A1, hnRNP K, and FUS, that actively control TARDBP splicing and the production of truncated protein variants. Notably, the authors generated a novel antibody that detects one of these splice variants and utilize this tool to confirm their results with regards to the endogenous protein. These findings have intriguing and potentially important implications for TARDBP and its gene product (TDP-43) in both healthy cells and disease.

While the data presented here are quite comprehensive and reflect a broad scope of molecular and biochemical experiments, there are some conceptual issues and mismatched data (i.e. immunofluorescence results that do not coincide with western blots) that should be addressed prior to publication. Moreover, despite the generally high quality of the data and the extensive characterization of splice isoforms, much of the the manuscript is written in a way that may confuse readers. Specific major and minor issues are listed below.

Major concerns:

It would be exceptionally helpful to have a model that incorporates all observations of how hnRNP A1, hnRNP K and FUS act in concert to regulate TARDBP splicing. Fig. 7h is one model included by the authors, but fails to adequately and simply summarize the results, while also introducing potentially confusing labels (i.e. ##). Fig 10c is perhaps a better and simpler model, but it currently appears too late in the manuscript to help readers make sense of the many experiments that are conducted.

Figure 1

There is a disconnect between the western blot and quantification in (b). The WB does not reflect a 50% reduction in TDP43-FL. The effect of RNA binding mutations is confusing (Line 127). RNA binding mutations appear to enhance dominant negative activity of all TDP43 variants, leading to enhanced cryptic exon inclusion. (d) should include confirmation of TDP43 knockdown if possible.

Figure 3

Can the MP20 ab detect endogenous MP20 by WB? Although an endogenous MP20 band appears in the WB in 4(d), there is no endogenous MP20 band in the Ctr input in 3(c).

Why is distribution of endogenous MP20 (nuclear) different from overexpressed M20 (cytosolic, nuclear excluded)?

Why is FL protein visible in (c)?

Figure 4
Quantification for hnRNP E1, E2 and A1 should be included.

Figure 5
In general, the way this figure is described in the text is very confusing.
The effect of hnRNP K on hnRNP A1 is only seen in cells with cytosolic hnRNP K. What does this mean?
Why does FLAG-hnRNP K staining in panel in 5(b) (top) look so different from 5(b) (bottom)?

Figure 6
Why weren't these results seen in Figure 4, by ICC?
The authors are encouraged to perform ICC using MP20 antibody to confirm result in 6(b) and (d).

Figure 7
In general, the way this figure is described in the text is very confusing.
The model in 7(h) could be confusing. The authors are encouraged to use a model like that in Fig 10c (but without FUS)

Figure 9
How is it that mRNA levels for MP20 and FL change (Fig. 9b) but there is no change in protein levels of FL-endo or SVs-endo (Sup Fig. 7)?
WB for FL-TDP43 are overexposed in (f)
Additional experiments are needed to determine if FUS acts through hnRNPK to affect MP20 and FL. For instance, overexpress FUS as well as hnRNP K

Minor concerns:
Immunocytochemistry experiments should include a nuclear stain (DAPI or Hoerchst), as well as a whole cell marker to distinguish subcellular compartments.
The labeling on all immunocytochemistry panels should be more specific to the images. Be sure to note which protein is FLAG-tagged when showing images of FLAG immunoreactivity.

Abstract
Line 31: remove "through", and "...alternative splicing of TARDBP," not TDP-43
Line 37: this sentence ends with the word, "suppressed." What is suppressed?
Line 38: mutant FUS "inadequately inhibited the dominant-negative isoform" of TDP43? and what does it do to hnRNP K?
Line 39: the last two sentences of the abstract are quite confusing as written.

Introduction
gene names and RNA should be italicized.
Line 74: remove "the" before TDP-43
Line 77: re-phrase in a way that includes less "RBPs" and "regulation"

Figure 1
Given the literature on the cytoplasmic localization of MP18s (Weskamp 2020, Shenouda 2022), it is surprising to see nuclear localization of MP18s here.
It should be clearer which overexpressed proteins are FLAG-tagged, and which are Venus-tagged, or both?
(d) contains both cryptic exon inclusion and exon skipping ('skiptic') events. It would be helpful to distinguish between cryptic and skiptic splicing events by separating into two separate panels. The authors could also consider including a primer diagram (such as the one currently shown for cryptic events) to clarify the method of detection for skiptic events.
Line 107: more references required for cryptic splicing (Melamed 2019, Klim 2019, Rosa Ma 2022, Seddhigi 2024, Brown 2022)
Line 118: discussion of MP20 localization is out of place, should occur above (line 104)
Line 129: I am not sure what the authors are trying to say ("Importantly...")
Line 134: this is out of place, and should come before RNA binding mutations (line 122)

Figure 3
Line 184: remove "indigenous"

Figure 4
Line 201: is this remarkable, or expected based on result in Figure 2(b)?
Line 205: shouldn't there be a decrease in MP20 after hnRNP A1 expression?

Figure 5
Line 227 refers to nonexistent Figure 5(d)

Figure 6

Line 241: "preventing TARDBP splicing to MP20"

Figure 8

Panels (d) and (g) are never referred to in the text.

Supplemental Figures 5-6

The text is very confusing when describing these data.

Line 352: awkward as written, "... although this region is not absolutely required for hnRNP K-dependent effects on TARDBP splicing."

Line 358-359: also very confusing and awkward. "Intriguingly" may not be the correct word. Maybe "accordingly"?

Discussion:

The authors spend time discussing inter-domain interactions of TDP43 RRMs, but not N-terminal driven oligomerization of TDP43

Line 424: why do these have to be unstable? Simply sequestered?

Line 435: needs references

The requirement for the C-terminus in TDP43 splicing should also be discussed

Line 524: is there evidence for TDP43 loss of function (i.e. cryptic splicing) in ALS-FUS?

Reviewer #2 (Comments to the Authors (Required)):

Retention of cryptic exons in neuronal genes has been identified as a common finding in ALS and FTD with TDP-43 proteinopathy. In this manuscript, Hasegawa-Ogawa and colleagues investigated the role of RNA binding proteins (RBPs) in the alternative splicing of TDP-43. The investigators first showed that an alternatively spliced TDP-43 isoform, TDP-43 marginal peptide 20 (or MP20) exhibited dominant negative effects on the retention of cryptic exons in TDP-43 target genes, GPSM2, ATG4B, PDP1, and BCL2L11. Next, the investigators used a candidate approach to show that RBPs HnRNPK and HnRNPA1 promote or suppress the splicing of TDP-43 M20 isoform, respectively. Finally, the authors showed that wild type FUS suppresses HnRNPK-mediated activation of M20 isoform splicing.

Overall, this is an interesting study that reveals new insights into the dominant negative role of TDP-43 spliced isoform M20 and how different RBPs, such as HnRNPK and HnRNPA1, can regulate the splicing of M20. However, there are several significant issues that need to be addressed to support the conclusions of this study. From a technical standpoint, the study requires more data to strengthen its conclusion. Without these additional data, this study is still quite preliminary and the conclusion premature. From a conceptual standpoint, the entire study was conducted using overexpression in heterologous cell lines. Hence, the physiological implications of the majority of data in this study remains unclear. Below are specific comments on how to improve this manuscript:

1. The entire study was conducted in HeLaS3 or HEK293 cells using overexpression of genetically engineered constructs. As such, the physiological relevance of the TDP-43 spliced isoform M20 and its reported upstream regulatory mechanisms by HnRNPK and HnRNPA1 remain unclear. For instance, the authors can provide data, using RT-qPCR combined with direct sequencing of PCR products to demonstrate the relative abundance of TDP-43 M20 (and M18, M13) isoforms in the adult mouse brain. Moreover, they can use their M20-specific antibody to detect the presence of M20 protein using immunostaining and western blots.
2. This study relied heavily on the use of TARDBP mini-gene and RT-qPCR to identify the presence of different TDP-43 spliced isoforms and characterize how different RBPs regulate TDP-43 splicing. However, the RT-qPCR approach did not have the "resolution" to distinguish TDP-43 M13 isoform (hence M20 and M13 were collectively grouped as "M20+"). To circumvent this issue, the authors should perform sequencing on the PCR products to figure out the relative abundance of each isoform.
3. Many key panels (including RT-qPCR, western blots) lack quantifications or proper control experiments to support the conclusion. Although the authors mentioned they used RT-qPCR, reading from Materials and Methods on this approach (page 22, lines 560-567), this reviewer still could not figure out how this approach is considered to be quantitative. In this context, it will be more convincing to quantify each PCR product (e.g., Figure 1d, 2b, 2d, 7f, and 8b) and show the consistency of these RT-PCR results.
4. Figure 4: Panel A compared the differences in the role of HnRNPK, HnRNPE1, HnRNPE2, and HnRNPA1. However, all subsequent studies in panels b, c and e will need to include results from E1, E2 and A1 to provide the effectiveness of the PCR reaction.
5. Figure 5: The images in panel 5b are quite confusing. Even after reading the figure legend for Figure 5b, it is still unclear what these images were intended for.

6. The data in Supplementary Figures 5 and 6 appeared to be incompatible with the central theme of the entire manuscript. As such, the inclusion of these two supplementary figures came across as out of place and very strange.

7. The findings of wild type FUS as an upstream regulator of HnRNPK in Figure 9 seem to contradict with those reported in Figure 2, which showed no obvious effects of wild type FUS in MP20 splicing (though these data were not quantified in any meaningful way). This inconsistency raises concerns regarding the validity of the overexpression system used in this study.

8. Finally, there are many areas in the manuscript where the data presentation and how they support the conclusion can be improved. For instance, the descriptions of TDP-43 RRM and RRM mutants in Page 5, lines 124-130, are very confusing. Moreover, the excessive use of adjectives and ("surprisingly", "intriguingly", "strongly", "importantly", "strikingly", etc) should be avoided.

Response Letter

Journal of Cell Biology

Dear

Monitoring Editor: Dr. Richard Youle,

Scientific Editor: Dr. Dan Simon

Thank you very much for reviewing our manuscript entitled, “TDP-43 activity is controlled by regulatory networks involving ALS-related RNA-binding proteins” by M. Hasegawa-Ogawa et al. We are resubmitting the manuscript (202406097R) after revising it extensively in accordance with your suggestions.

We are grateful to you for your review of our original manuscript and for your encouragement to us for resubmitting the manuscript after appropriate revision. We are pleased to learn that the reviewers greatly appreciated our work. We have revised the title to more appropriately reflect the main findings of the study: “Dominant-negative isoform of TDP-43 is regulated by ALS-linked RNA-binding proteins.” We hereby enclose our revised manuscript. We also attach point-by-point response to the reviewers' comments as summarized below.

We hope that you find the revised manuscript acceptable for publication in *Journal of Cell Biology*. Thank you for your consideration and for your suggestions, which were highly constructive and valuable for strengthening the quality of our manuscript.

Sincerely,

Hiroataka James Okano M.D., Ph.D. Professor

Division of Regenerative Medicine, The Jikei University School of Medicine

Address: 3-25-8 Nishi-Shimbashi, Minato-ku, Tokyo, 1058461, Japan

E-mail: hjokano@jikei.ac.jp

Tel: 03-3433-1111 or 03-5400-1200 (ext. 2350) Fax: 03-5400-1297

Reviewer #1 (Comments to the Authors (Required)):

In this manuscript, Hasegawa-Ogawa et al. explore a network of RNA binding proteins regulating the splicing of TARDBP, a transcript and gene product that are intimately related to the pathogenesis of neurodegenerative disorders. TARDBP splice variants have been detected and described in prior studies, but remain inadequately characterized. The authors uncover an intricate interplay between several RNA binding proteins, including hnRNP A1, hnRNP K, and FUS, that actively control TARDBP splicing and the production of truncated protein variants. Notably, the authors generated a novel antibody that detects one of these splice variants and utilize this tool to confirm their results with regards to the endogenous protein. These findings have intriguing and potentially important implications for TARDBP and its gene product (TDP-43) in both healthy cells and disease.

While the data presented here are quite comprehensive and reflect a broad scope of molecular and biochemical experiments, there are some conceptual issues and mismatched data (i.e. immunofluorescence results that do not coincide with western blots) that should be addressed prior to publication. Moreover, despite the generally high quality of the data and the extensive characterization of splice isoforms, much of the the manuscript is written in a way that may confuse readers. Specific major and minor issues are listed below.

Major concerns:

It would be exceptionally helpful to have a model that incorporates all observations of how hnRNP A1, hnRNP K and FUS act in concert to regulate TARDBP splicing. Fig. 7h is one model included by the authors, but fails to adequately and simply summarize the results, while also introducing potentially confusing labels (i.e. ##). Fig 10c is perhaps a better and simpler model, but it currently appears too late in the manuscript to help readers make sense of the many experiments that are conducted.

Response to reviewer's comments:

We sincerely thank Reviewer #1 for their thorough review and thoughtful feedback. We greatly appreciate the recognition of our findings and their potential implications, as well as the constructive suggestions for improvement. Below, we address the specific concerns raised.

Figure 1

There is a disconnect between the western blot and quantification in (b). The WB does not reflect a 50% reduction in TDP43-FL.

Thank you for your insightful suggestion.

It seems the misunderstanding may have been caused by the high contrast of the TDP-43 Western blot (WB) image, which made the 50% reduction in TDP-43-FL appear less evident. To address this, we have included below the original WB images of TDP-43 from three independent experiments, along with the corresponding quantitative values (normalized to β -Actin and expressed as ratios to the NoTF control). To avoid further confusion, we have replaced the TDP-43 image in the Revised Fig. 1 B with a version of lower contrast, which better represents the quantitative data.

Revised Fig. 1 B (Magnified image of TDP-43)

The effect of RNA binding mutations is confusing (Line 127). RNA binding mutations appear to enhance dominant negative activity of all TDP43 variants, leading to enhanced cryptic exon inclusion.

Thank you for pointing out the potential confusion in the description of the RNA binding mutations. To address this, we have revised the text at Line 151-162 (previously Line 127) to clarify the effect of the RNA binding mutations as follows.

"In our study, we introduced F147/149L mutations into the RRM1 domain of MP20 (127), as well as TDP-FL and MP18 (127) variants (Fig. 2 D), and confirmed their expression in HEK293T cells (Fig. S1 E). The F147/149L mutant of each splice variant is referred to below as RNA-binding deficient mutant (Rmt). The overexpression of all Rmt variants of FL, MP20, and MP18 resulted in increased CE inclusion in GPSM2 and ATG4B, suggesting that the suppression of CEs depends on the functional RRM domain of TDP-43 (Fig. 2, E and F). FL

Rmt altered the alternative splicing patterns of PDP1 and BCL2L11 in the opposite direction to that observed with wild-type (WT) FL (Fig. 2, E and G). Similarly, MP18 Rmt altered PDP1 splicing in a manner consistent with Rmt FL, with no significant changes in BCL2L11 splicing compared to WT MP18. Notably, WT MP20 showed similar changes in either PDP1 or BCL2L11 splicing compared to MP20 Rmt. Taken together, RNA binding mutations appear to enhance dominant negative activity of all TDP43 variants, however WT MP20 exerts the same level of dominant-negative activity as the RNA binding mutants.”

(d) should include confirmation of TDP43 knockdown if possible.

Thank you for your valuable suggestion. In response, we performed additional WB analysis in HEK293T cells with TDP-FL knockdown (FL KD). The results show a reduction of endogenous TDP-FL (FL-endo) expression to 21.5% relative to the Mock control (Revised Fig. S1 C).

Revised Fig. S1 C

In addition, to confirm the reduction of mRNA expression of FL-endo and the loss of the ability to suppress cryptic exon (CE) inclusion, we also conducted RT-qPCR for FL-endo, *GPSM2* CE, and *ATG4B* CE in HEK293T cells with FL KD. As expected, mRNA expression of FL-endo showed a 91.4% reduction, while *GPSM2* CE and *ATG4B* CE exhibited marked increases of 1918- and 3968-fold, respectively (Revised Fig. S1 D).

Revised Fig. S1 D

Accordingly, we have supplemented Revised Fig. S1, C and D revised the text at Line 126-

128.

Figure 3

Can the MP20 ab detect endogenous MP20 by WB? Although an endogenous MP20 band appears in the WB in 4(d), there is no endogenous MP20 band in the Ctr input in 3(c).

Thank you for the detailed suggestion. In our data, endogenous MP20 was successfully detected by WB not only in Initial Figure 4d but also in 4c, 4e, 6b, 6d, 7d, 7g, 8c, 8f, and 9f. Therefore, it is likely that in Initial Figure 3c, the expression levels of overexpressed MP20 were substantially higher than endogenous MP20, making the latter appear undetectable.

In initial Figure 3c, only low contrast WB images adjusted to highlight the exogenous MP20 band were provided. To address this, we have now included additional high contrast images of TDP-43 and MP20 in Revised Fig. 4 D (red frame), along with the original low contrast images.

Revised Fig. 4 D

Indeed, in experiments where MP20 was overexpressed (under these specific condition), endogenous MP20 appears to be less detectable. It is also possible that large amounts of overexpressed MP20 saturated the antibody binding, reducing the detectability of endogenous MP20 in these experiments.

Accordingly, we have supplemented Revised Fig. 4 D revised the text at Line 217-219 as follows.

“However, excessive FLAG-MP20 overexpression saturated the antibody, hindering the detection of endogenous MP20 (MP20-endo) and TDP-43-positive endogenous short variants (SVs-endo).”

Why is distribution of endogenous MP20 (nuclear) different from overexpressed M20 (cytosolic, nuclear excluded)?

Thank you for your thoughtful question. As shown in Revised Fig. S1 B (below), we observed two distinct patterns of subcellular localization for overexpressed FLAG-MP20: (1) a predominantly nuclear localization with some cytoplasmic leakage, which resembles the pattern observed for endogenous MP20, and (2) a primarily cytosolic localization with nuclear exclusion. In our experiment, cells with higher levels of FLAG-MP20 expression tended to exhibit the second pattern.

In contrast, endogenous MP20 expression is significantly lower than that of overexpressed MP20. Consequently, the second localization pattern is rarely observed in cells expressing endogenous MP20. However, it is possible that, under conditions of increased expression, endogenous MP20 might also localize to the cytoplasm.

Revised Fig. S1 B

Additionally, we have included high contrast images in Revised Fig. 4 B to highlight the substantial difference in MP20 signal between FLAG-MP20-positive (arrowhead) and negative cells (arrow).

Revised Fig. 4 B (high contrast images in lower panel, provided only in Response Letter)

Why is FL protein visible in (c)?

In the magnified image of TDP-43 below, molecular weights of bands in IP: FLAG and IP: MP20 lanes differ from the FL-endo band (magenta dotted line).

Revised Fig. 4 D (Magnified image of TDP-43)

However, previous studies have shown that TDP-FL forms homodimers/oligomers via N-terminal domain (NTD) (Afroz et al., 2017; Mompeán et al., 2017; Zhang et al., 2013). MP20 retains the NTD, suggesting the possibility of heterodimers/oligomers formation with TDP-FL. We further performed a co-immunoprecipitation (Co-IP) experiment in HEK293T cells co-overexpressing FLAG-MP20 and FL-Venus (Revised Fig. 2 H, below). As previously described, FLAG-FL and FL-Venus interaction was confirmed. Interestingly, FLAG-MP20 and FL-Venus also showed significant interactions compared with FLAG-FL or FLAG-MP18. Furthermore, an MP20 RRM domain mutant did not reduce this interaction, indicating that MP20 and FL form a complex through protein-protein interactions independent of RNA binding capacity.

Revised Fig. 2 H

Previous studies have also demonstrated that TDP-FL homodimer/oligomer formation is essential for its splicing function (Afroz et al., 2017; Mompeán et al., 2017). These findings suggest that the formation of heterodimers/oligomers between MP20 and FL may contribute to the dominant-negative inhibition of MP20, thereby leading to splicing dysregulation.

Accordingly, we have supplemented Revised Fig. 2 H and revised the text at Line 165-174, and 440-447 as follows.

Line 165-174

“We performed a co-immunoprecipitation (Co-IP) assay in HEK293T cells co-overexpressing FLAG-MP20 and FL-Venus (Fig. 2 H). The degree of protein-protein interaction was quantified by the ratio of FL-Venus to FLAG in the immunoprecipitated samples. Consistent with previous reports, an interaction between FLAG-FL and FL-Venus was confirmed. Interestingly, FLAG-MP20 also showed a significant interaction with FL-Venus, which was stronger than that observed with FLAG-FL or FLAG-MP18 (by 5.7-fold relative to FLAG-FL, and by 2.3-fold relative to FLAG-MP18). Notably, MP20 Rmt did not reduce this interaction, suggesting that MP20 forms a complex with FL through protein-protein interactions independent of its RNA-binding capacity. Taken together, our findings suggest that MP20 exhibits stronger protein-protein interactions compared to FL and MP18, which may be relevant to heterodimer/oligomer formation.”

Line 440-447

“FL forms functional homodimers/polymers thorough its NTD, which are critical for splicing regulation (Afroz et al., 2017; Mompeán et al., 2017). MPs retain the NTD, allowing them to

form heterocomplexes with FL. Indeed, MP20, which significantly promoted CE inclusion, demonstrated higher complex formation with FL than FL or MP18, as shown by Co-IP (Fig. 2 H). Therefore, MP20 may inhibit functional FL homodimer/polymer formation by competitively forming heterocomplexes, contributing to its dominant-negative activity. Despite reducing FL-endo levels, MP18 overexpression did not induce CE inclusion, and changes in splicing targets were similar to FL overexpression (Fig. 2 A). These findings suggest that the MP18-FL heterocomplex may partially retain the splicing function of FL homocomplexes.”

Figure 4

Quantification for hnRNP E1, E2 and A1 should be included.

Thank you for your constructive suggestion. To evaluate the effects of hnRNPs on MP20 expression, we performed additional immunocytochemistry (ICC) and WB experiments. In the ICC, overexpression of hnRNP K, E1 and E2 resulted in a significant increase in nuclear MP20 levels, while hnRNP A1 had no such effect compared to control (FLAG-negative cells) (Revised Fig. 5 C). In the WB experiments, hnRNP K and E1 significantly increased both SVs-endo and MP20-endo, consistent with the ICC findings (Revised Fig. 5 D). hnRNP E2 showed a tendency to increase MP20-endo, but the effect was not statistically significant. hnRNP A1 had no effect on MP20 expression in ICC the experiment.

Further details regarding the hnRNP A1 overexpression experiments are provided in the Response to Figure 6.

Revised Fig. 5 B

Revised Fig. 5 D

Accordingly, we have supplemented Revised Fig. 5, B and D, and revised the text in Line 244-248, and 250-254 as follows.

Line 244-248

“Quantification of nuclear MP20 fluorescence intensity in immunocytochemistry showed a 1.4-fold increase in the FLAG-hnRNP K-positive cells (arrowheads) compared to FLAG-negative cells (arrow), suggesting that hnRNP K enhances nuclear MP20 expression (Fig. 5 B). A similar increase in nuclear MP20 was observed in FLAG-hnRNP E1 and FLAG-hnRNP E2-positive cells, while FLAG-hnRNP A1-positive cells showed no detectable change in MP20 levels (Fig. 5 B).”

Line 250-254

“WB analysis in HEK293T cells further confirmed that overexpression of hnRNP K and hnRNP E1 increased MP20-endo and SVs-endo levels (Fig. 5 D). A similar trend was observed with hnRNP E2, although the increase was not statistically significant. In contrast, overexpression of hnRNP A1 led to a reduction in both SVs-endo and MP20, a result that did not align with the immunocytochemistry data (Fig. 5 D).”

Figure 5

In general, the way this figure is described in the text is very confusing.

Why does FLAG-hnRNP K staining in panel in 5(b) (top) look so different from 5(b) (bottom)?

We apologize for the confusion.

In the initial Figure 5b, different FLAG-tag antibodies were used to stain FLAG-hnRNP K for each panel due to the combination of antibodies selected for the ICC experiment. This resulted in differences in sensitivity between the antibodies, which likely caused the FLAG-

hnRNP K staining to appear differently across the panels.

In the present study, we performed additional ICC experiments using the same FLAG-tag antibody for all panels, after purchasing a new hnRNP A2/B1 antibody.

Quantification of the nuclear hnRNP A1 signal showed a marked increase with hnRNP K overexpression, while the hnRNP A2/B1 signal tended to increase but did not reach statistical significance (Revised Fig. 6B, C).

Revised Fig. 6, B and C

The effect of hnRNP K on hnRNP A1 is only seen in cells with cytosolic hnRNP K. What does this mean?

As you pointed out, when comparing the nuclear hnRNP A1 (and hnRNP A2/B1) signal in cells with FLAG-hnRNP K localized in the nucleus (Nuc) and those with FLAG-hnRNP K leaking into the cytoplasm (Cyto + Nuc), we found that the signal was significantly higher in the cells with cytoplasmic leakage (Revised Fig. 6 D). In addition, overexpression of hnRNP K appears to reduce the amount of hnRNP A1 transcript (Revised Fig. S2 A), despite an increase in hnRNP A1 protein levels (Revised Fig. 6 A), suggesting a post-transcriptional mechanism that enhances hnRNP A1 protein stability or translation. For example, hnRNP K might translocate proteins that repress hnRNP A1 to the cytoplasm.

Revised Fig. 6 D

Accordingly, we have supplemented Revised Fig. 6, B-D, and revised the text at Line 273-274 as follows.

“In addition, cells with cytoplasmic leakage of FLAG-hnRNP K exhibited significantly increased hnRNP A1 and hnRNP A2/B signals compared to cells with nuclear FLAG-hnRNP K (Fig. 6 D).”

Additionally, we performed further analysis to investigate the effect of FLAG-hnRNP K cytoplasmic leakage on the nuclear intensity of MP20. We found that the nuclear intensity of MP20 was also significantly higher when FLAG-hnRNP K leaked into the cytoplasm (Revised Fig. 5 C).

Revised Fig. 5 C

In our result, overexpression of hnRNP K induces increased expression of MP20, which is associated with enhanced splicing of MP20 (Fig. 3 B-E, Fig. 5 A, B, and D). Therefore, our findings suggest that the splicing promotion of MP20 by hnRNP K does not occur through direct interaction with *TARDBP* RNA. This is further supported by the lack of a clear

responsible region for hnRNP K, as shown by experiments using mini-gene constructs with systematic deletion of *TARDBP* exon 6 (Revised Fig. S4 A-D).

Accordingly, we have supplemented Revised Fig. 5 C, and revised the text at Line 248-250, and 458-463 as follows.

Line 248-250

“Notably, in cells where FLAG-hnRNP K exhibited cytoplasmic leakage, the nuclear fluorescence intensity of MP20 was significantly increased by 1.2-fold compared to cells with exclusively nuclear FLAG-hnRNP K localization (Fig. 5 C).”

Line 458-463

“Our results indicate that MP20 expression is particularly elevated in cells with cytoplasmic leakage of hnRNP K (Fig. 5C), suggesting the promotion of MP20 splicing by hnRNP K (Fig. 3 B-E) does not appear to result from direct interaction with TARDBP RNA. This is supported by the absence of a specific TARDBP region regulated by hnRNP K, as shown by experiments using TARDBP exon 6 mini-gene constructs with systematic deletion (Fig. S4 A-D), and by the trend of increased MP20 RNA and protein levels observed following hnRNP K KD (Fig. 3 G and 5 E).”

In addition, we also show the proportion of cytoplasmic leakage of FLAG-hnRNPs in HeLaS3 cells (Revised Fig. S1 G).

Revised Fig. S1 G

Despite the relatively low proportion of hnRNP K cytoplasmic leakage (18.4%), it appears to have a strong impact on the upregulation of MP20 and hnRNP A1 expression. Accordingly, we have supplemented Revised Fig. S1 G, and revised the text at Line 241-242 as follows

“FLAG-hnRNP K localized in both the nucleus and cytoplasm, with cytoplasmic leakage

observed in 18.4% of cells (Fig. 5 A and S1 G)."

Figure 6

Why weren't these results seen in Figure 4, by ICC?

The authors are encouraged to perform ICC using MP20 antibody to confirm result in 6(b) and (d).

Thank you for your constructive suggestion. In Revised Fig. 5 B, hnRNP A1 overexpression in HeLaS3 did not affect nuclear MP20 expression in ICC experiment. The discrepancy may be due to differences in overexpression efficiency between HEK293T (used for WB) and HeLaS3 (used for ICC), or the different plasmid amounts transfected (HEK293T; 2 μ g, HeLaS3; 1 μ g)

To validate this, we performed additional ICC experiments in which we transfected various amounts of hnRNP A1 plasmid (0, 1, 2 μ g) in both cell types. However, no change in nuclear MP20 signal was observed in either HeLaS3 or HEK293T cells, regardless of the plasmid dose (below).

Supporting Fig. (Response Letter only)

In contrast, in HeLaS3 cells, nuclear MP20 signal was increased following hnRNP A1 KD (Revised Fig. S1, J and K), which is consistent with the WB results in HEK293T (Initial Figure 6d (Revised Fig. S1 I)).

Revised Fig. S1, J and K

Regarding the WB analysis of hnRNP A1 overexpression, while the reduction in MP20-endo was not significant in the overall analysis (six groups: NoTF Ctr, Vector Ctr, hnRNP K, E1, E2, and A1), it became significant when restricted to three groups (NoTF Ctr, Vector Ctr, and hnRNP A1) (Revised Fig. 5 D).

Additional WB experiments in HEK293T, with different amounts of hnRNP A1 plasmid (0, 1, 2 μ g), confirmed a significant decrease in SVs-endo, MP20-endo, and FL-endo (data below), supporting the reproducibility of the significant decrease in MP20-endo previously observed in Figure 6b (Revised Fig. 5 D).

Supporting Fig. (Response Letter only)

One concern is that the results of hnRNP A1 overexpression did not match between WB and ICC experiments. This could be due to differences in the sensitivity of the two methods or the variation in transfection efficiency between the two cell lines.

Accordingly, we have supplemented Revised Fig. S1, J, K, and Revised Fig. 5 D, and revised the text at Line 254-256, and 250-256 as follows.

Line 250-256

“WB analysis in HEK293T cells further confirmed that overexpression of hnRNP K and hnRNP E1 increased MP20-endo and SVs-endo levels (Fig. 5 D). A similar trend was observed with hnRNP E2, although the increase was not statistically significant. In contrast, overexpression of hnRNP A1 led to a reduction in both SVs-endo and MP20, a result that did not align with the immunocytochemistry data (Fig. 5 D). Consistent with WB results, hnRNP A1 KD induced a 2.71-fold increase in MP20 (127) mRNA levels and a 2.02-fold increase in protein levels, as confirmed by immunocytochemistry, while FL mRNA and protein levels remained unchanged (Fig. S1 H-K).”

Figure 7

In general, the way this figure is described in the text is very confusing.

The model in 7(h) could be confusing. The authors are encouraged to use a model like that in Fig 10c (but without FUS)

We apologize for the confusion. To clarify, the schematic diagram has been updated (Revised Fig. S2 I-K) and revised the text at Line 326-331 as follows.

“In summary, excess hnRNP K promotes alternative splicing, generating of MP20 (127) and subsequently increasing MP20 protein levels (Fig. S2 I), while hnRNP A1 promotes splicing that excises intron 5 to exon 7, resulting in SMP13 with undetectable protein levels (Fig. S2 J). Coexpression of hnRNP K and hnRNP A1 impedes the formation of MP20 and promotes the generation of shorter isoforms, which are not detected protein expression (Fig. S2 K). These findings highlight the critical role of hnRNP A1 in modulating TARDBP splicing, particularly in counteracting the hnRNP K-mediated increase in MP20 production.”

Figure 9

How is it that mRNA levels for MP20 and FL change (Fig. 9b) but there is no change in protein levels of FL-endo or SVs-endo (Sup Fig. 7)?

Thank you for the detailed suggestion. FUS, as an RNA-binding protein, directly interacts with target RNAs, potentially leading to significant changes in RNA levels. Indeed, TDP-43 has been reported to be a target RNA of FUS (Lagier-Tourenne et al., 2012). On the other hand, proteins are subject to regulation by various factors and have specific half-lives, which often results in protein levels not directly correlating with mRNA levels.

In our study, FLAG-WT FUS was predominantly localized in the nucleus (Revised Figure 9 E, G), suggesting that the observed decrease in FL-endo and increase in MP20 (127) mRNA levels (Revised Fig. 9 C) is likely due to splicing regulation in nuclear. Additionally, FL protein levels are known to be autoregulated strictly through post-transcriptional mechanism (Avendaño-Vázquez et al., 2012; Ayala et al., 2011), which may explain why protein levels do not fully reflect mRNA changes

In addition, the P525L FUS mutant was predominantly localized in the cytoplasm (Revised Figure 9E, G), which likely limits its role in nuclear splicing regulation. Furthermore, it has been reported that the cytoplasmic mislocalization of mutant FUS can impair NMD (nonsense-mediated mRNA decay) regulation and protein translation (Kamelgarn et al., 2018). This could explain the increase in MP20 (127) mRNA levels following P525L

overexpression, potentially due to RNA stabilization in the cytoplasm by the P525L mutant.

Note that SVs-endo contains proteins derived from various splicing isoforms, not only truncated FL-endo but also other isoforms distinct from MP20. Thus, changes in protein levels within SVs-endo may not directly reflect the levels of MP20-endo (Initial Supplementary Fig. 7 (Revised Fig. S5 D)).

Accordingly, we have revised the text at Line 407-408 as follows.

“However, FL-endo and SVs-endo protein levels were unaffected by the overexpression of either WT or P525L FUS (Fig. S5 D).”

WB for FL-TDP43 are overexposed in (f)

Thank you for the detailed suggestion. A low contrast image was added to show the FL-endo and SVs-endo bands in proper contrast (Revised Fig. 9F, red flame).

Revised Figure 9F

Additional experiments are needed to determine if FUS acts through hnRNPK to affect MP20 and FL. For instance, overexpress FUS as well as hnRNPK

Thank you for the thoughtful suggestion. To determine whether FUS affects MP20 and FL through hnRNPK, we conducted additional ICC and WB experiments to examine the expression of MP20 and FL under co-overexpression of FLAG-FUS and FLAG-hnRNPK. In ICC experiments, nuclear MP20 signal increased significantly in FLAG-hnRNPK-positive cells and decreased in FLAG-FUS-positive cells, as expected. In cells co-expressing FLAG-

FUS and FLAG-hnRNP K, nuclear MP20 was elevated compared to control (FLAG-negative cells) but reduced compared to FLAG-hnRNP K overexpression alone (below).

Supporting Fig. (Response Letter only)

In WB experiments, the decrease in SVs-endo and MP20-endo levels observed with FUS overexpression was attenuated by co-expression with hnRNP K. However, FLAG-FUS protein levels were substantially reduced when co-expressed with FLAG-hnRNP K (below).

Supporting Fig. (Response Letter only)

FLAG **hnRNP K**
FLAG **FUS**

Co-transfection in HEK293T

To further explore whether hnRNP K affects FUS expression, we performed WB analysis in HEK293T cells overexpressing FLAG-hnRNP K. Protein levels of endogenous FUS (FUS-endo) were not altered by hnRNP K overexpression (below).

Supporting Fig. (Response Letter only)

Additionally, ICC experiments in HeLaS3 cells overexpressing FLAG-hnRNP K showed no significant changes in FUS-endo signal (below).

Supporting Fig. (Response Letter only)

These findings suggest that increased hnRNP K does not impact endogenous FUS expression.

Therefore, the observed reduction in FLAG-FUS protein levels during co-expression with hnRNP K, and its subsequent effect on MP20, is likely results from interference between the promoters of two the expression vectors, rather than a direct of hnRNP K on FUS expression.

Further verification is required to determine whether FUS acts through hnRNP K. To reflect this, we updated the schematic diagram to include an arrow indicating that FUS may act directly target MP20 RNA, as well as an arrow suggesting the possibility of FUS acting on hnRNP K (Revised Fig. 10, B and C, below).

Revised Fig. 10, B and C

Minor concerns:

Immunocytochemistry experiments should include a nuclear stain (DAPI or Hoerchst), as well as a whole cell marker to distinguish subcellular compartments.

The labeling on all immunocytochemistry panels should be more specific to the images. Be sure to note which protein is FLAG-tagged when showing images of FLAG immunoreactivity.

Thank you for the suggestion. We included a single Hoechst image to all immunocytochemistry data, and the labeling on all immunocytochemistry panels was modified to be more specific (Revised Figs. 1 C, 4 B, 4 C, 4 G, 5 A, 6 B, 9 E, 9 G, S1 B, S1 J, and S2 E).

In addition, we noted which protein is FLAG-tagged when showing images of FLAG immunoreactivity Revised Figs. 1 C, 4 B, 5 A, 6 B, 9 E, 9 G, S1 B, and S2 E.

Abstract

Line 31: remove "through", and "...alternative splicing of TARDBP," not TDP-43

We have made the revisions as per your recommendation.

Line 37: this sentence ends with the word, "suppressed." What is suppressed?

Line 38: mutant FUS "inadequately inhibited the dominant-negative isoform" of TDP43? and what does it do to hnRNP K?

Line 39: the last two sentences of the abstract are quite confusing as written.

We have substantially revised the Abstract to improve clarity. Specifically, we have rewritten the sentences to better describe the relationships between hnRNP K, hnRNP A1, FUS, and the dominant-negative isoforms of TDP-43.

Introduction

gene names and RNA should be italicized.

Line 74: remove "the" before TDP-43

We have made the revisions as per your recommendation.

Line 77: re-phrase in a way that includes less "RBPs" and "regulation"

We have revised the sentence to reduce the repetition of "RBPs" and "regulation". The updated sentence now reads: Line 90: *Our work suggests that a network of RBPs regulates TDP-43, and abnormalities within any component of this regulatory network may lead to TDP-43 dysfunction, ultimately contributing to neurodegenerative diseases such as ALS and FTL.*

Figure 1

Given the literature on the cytoplasmic localization of MP18s (Weskamp 2020, Shenouda 2022), it is surprising to see nuclear localization of MP18s here.

Thank you for highlighting this point. In Weskamp et al., EGFP-sTDP43 (MP18) was overexpressed in rodent cortical neurons, where it exhibited significant nuclear localization, despite showing some degree of cytoplasmic leakage. Similarly, in our study, FLAG-MP18 overexpression in HeLaS3 cells resulted in primarily nuclear localization, with some instances of cytoplasmic leakage also observed (Revised Fig. 1 C), indicating consistency with their findings.

Regarding Shenouda et al., their study focused on a unique C-terminal isoform of MP18, referred to as C-spl, which differs from sTDP43 in having 16 unique amino acids at its C-terminus. EGFP-tagged C-spl exhibited nuclear localization in non-neuronal cell lines but was cytoplasmic in neuronal cell lines. This suggests that cytoplasmic localization of MP18 may vary between neuronal and non-neuronal cells. Since our study utilized a non-neuronal cell line (HeLaS3), the observed nuclear localization aligns with their findings.

For endogenous MP18, sTDP43 has been reported to form cytoplasmic aggregates in ALS patient neuronal tissues, with 10-50% of it still showing nuclear localization (Weskamp et al. Supplementary Figure 14). While C-spl also forms cytoplasmic aggregates in ALS motor neuron tissues, Shenouda et al. did not report its nuclear localization or relative distribution. These observations collectively suggest that MP18 is not exclusively cytoplasmic and may localize differently depending on cell type. Specifically, MP18 appears more prone to

cytoplasmic localization in neuronal cells. Future studies using neuronal cell lines or iPSC-derived neurons may further elucidate the cell type-specific localization pattern of MP isoforms.

It should be clearer which overexpressed proteins are FLAG-tagged, and which are Venus-tagged, or both?

We apologize for any confusion. To clarify, we have added schematic diagrams to each panel to clearly indicate which overexpressed proteins are FLAG-tagged, Venus-fused (Revised Fig. 1, B and C).

(d) contains both cryptic exon inclusion and exon skipping ('skiptic') events. It would be helpful to distinguish between cryptic and skiptic splicing events by separating into two separate panels. The authors could also consider including a primer diagram (such as the one currently shown for cryptic events) to clarify the method of detection for skiptic events.

Thank you for constructive comment. In response, we have added a primer diagram for the detection of alternative splicing targets in addition to the one already included for cryptic splicing events. Additionally, we have included a simplified schematic of the target exon inclusion/exclusion to the right of the RT-PCR bands to further clarify the splicing events. (Revised Fig. 2 A).

Revised Fig. 2 A

Line 107: more references required for cryptic splicing (Melamed 2019, Klim 2019, Rosa Ma 2022, Seddhigi 2024, Brown 2022)

We have included the references in Line 123 as suggested.

Line 118: discussion of MP20 localization is out of place, should occur above (line 104)

We have moved the suggested text to Line 119 as requested.

Line 129: I am not sure what the authors are trying to say ("Importantly...")

We apologize for the confusion. In response to your suggestion, we have revised the sentence

originally found in Line 129, and have relocated it to Line 159. The revised wording now aims to clarify the intended meaning.

Line 134: this is out of place, and should come before RNA binding mutations (line 122)

Due to substantial revisions made to the section regarding RNA binding mutations (Line 122), the sentence in Line 134 has been removed.

Figure 3

Line 184: remove "indigenous"

We have made the revisions as per your recommendation.

Figure 4

Line 201: is this remarkable, or expected based on result in Figure 2(b)?

Indeed, Figure 2b (now revised as Fig. 3 B) demonstrates that hnRNP K enhances MP20 splicing, however, it does not necessarily imply that there is a concomitant increase in protein levels, so it cannot be considered entirely expected. Nevertheless, I agree that the use of "Remarkably" was too strong, and we have removed the expression in Line 244 with revised version.

Line 205: shouldn't there be a decrease in MP20 after hnRNP A1 expression?

Details on the hnRNP A1 overexpression experiments can be found below in the Response to Reviewer #1 Figure 6.

Figure 5

Line 227 refers to nonexistent Figure 5(d)

The hnRNP K KD data for hnRNP A1 protein levels are shown in the right panel of Figure 5d (now revised as Fig. 6 E, Line 275), and the corresponding mRNA levels are presented in Figure S2b (now revised as Fig. S2 B, Line 275).

Figure 6

Line 241: "preventing *TARDBP* splicing to MP20"

We have removed that part as it was extensively revised.

Figure 8

Panels (d) and (g) are never referred to in the text.

We apologize for the oversight. The panels (d) and (g) (now revised as Fig. 8, D and G) have now been referenced in the text, specifically in Line 339-341, 345-347. Thank you for pointing this out, and we have made the necessary revisions accordingly.

Supplemental Figures 5-6

The text is very confusing when describing these data.

Line 352: awkward as written, "... although this region is not absolutely required for hnRNP K-dependent effects on TARDBP splicing."

Line 358-359: also very confusing and awkward. "Intriguingly" may not be the correct word. Maybe "accordingly"?

Thank you for your comments. In response, we have removed Initial Supplemental Figure 5e-f as it falls outside the main scope of this study. Supplemental Figure 5a-d and Supplemental Figure 6a-b in initial version have been combined into the Revised Fig. S4, and we have revised the figure legend and related text (Line 356-386). We have made efforts to improve clarity and ensure the figures are more easily understood.

Discussion:

The authors spend time discussion inter-domain interactions of TDP43 RRMs, but not N-terminal driven oligomerization of TDP43

We appreciate your insightful comment. As noted, while we spent time discussing the inter-domain interaction of TDP-43 RRMs, we did not sufficiently address the importance of N-terminal driven oligomerization of TDP-43. Previous studies have demonstrated that the N-terminal domain (NTD)-mediated dimerization/oligomerization of TDP-43 is physiologically critical for its splicing activity (Afroz et al., 2017; Jiang et al., 2017; Mompeán et al., 2017). Interestingly, our additional experiments revealed that MP20 shows a stronger ability to form complexes with TDP-43 than TDP-FL does with itself (Revised Fig. 2 H). This observation suggests that MP20 may preferentially associate with TDP-43, competitive disruption of oligomerization could underlie the dominant-negative activity of MP20. In contrast, MP18 also forms complexes with TDP-FL and leads to a reduction in endogenous TDP-FL levels (Revised Fig. 2 H; and Fig. 1, B and C). However, MP18 exhibits minimal dominant-negative activity. This could be explained by the partial retention of splicing activity in MP18-TDP-FL complexes. These findings highlight the interplay between TDP-43 NTD-oligomerization and MPs interactions, providing further insights into the mechanisms underlying the differential activities of TDP-MPs. Accordingly, we have revised the text at Line 434-447 as follows.

Regarding the discussion on the inter-domain interactions of TDP-43 RRMs, we have removed this section from the revised manuscript, as the connection to N-terminal driven oligomerization of TDP-43 seems more relevant, and due to word limit constraints.

"MP20 shares functional domains from the NTD to RRM2 with FL but lacks the C-terminal region, which is essential for interactions with other splicing regulatory proteins. For instance, TDP-43 splicing activity in CFTR exon 9 skipping relies on protein-protein interactions

between its C-terminal residues 321-366 and hnRNP A2 and A1 (Buratti et al., 2005; D'Ambrogio et al., 2009). Given that MP20 lacks this region, it is likely unable to regulate splicing independently.

While all MPs analyzed in this study similarly reduced FL-endo protein levels, their impact on CE inclusion varied (Fig. 1, B and C; Fig. 2 A). FL forms functional homodimers/polymers through its NTD, which are critical for splicing regulation (Afroz et al., 2017; Mompeán et al., 2017). MPs retain the NTD, allowing them to form heterocomplexes with FL. Indeed, MP20, which significantly promoted CE inclusion, demonstrated higher complex formation with FL than FL or MP18, as shown by Co-IP (Fig. 2 H). Therefore, MP20 may inhibit functional FL homodimer/polymer formation by competitively forming heterocomplexes, contributing to its dominant-negative activity. Despite reducing FL-endo levels, MP18 overexpression did not induce CE inclusion, and changes in splicing targets were similar to FL overexpression (Fig. 2 A). These findings suggest that the MP18-FL heterocomplex may partially retain the splicing function of FL homocomplexes.”

Line 424: why do these have to be unstable? Simply sequestered?

We agree with your comment that TDP-FL monomers are not necessarily unstable. To avoid any misunderstanding, we have removed the corresponding text.

Line 435: needs references

Thank you for your comment. We have removed the sentence in question, and therefore the reference is no longer necessary.

The requirement for the C-terminus in TDP43 splicing should also be discussed

Thank you for the constructive comment regarding the requirement for the C-terminus in TDP-43 splicing. We have addressed this point by discussing CFTR exon 9 skipping, a well-characterized splicing target of TDP-43. It has been shown that TDP-43 interacts with splicing factors such as hnRNP A2 and hnRNP A1 via its C-terminal region, and this protein-protein interaction is essential for mediating CFTR exon 9 skipping (Buratti et al., 2005; D'Ambrogio et al., 2009). Therefore, the C-terminus of TDP-43 is a critical domain required for interactions with other proteins required for splicing regulation. Since MP20 and MP18 lack this C-terminal region, we propose that these isoforms are unable to independently mediate splicing regulation. Indeed, it has been shown that MP18 (118) lacks the ability to induce CFTR exon 9 skipping, supporting this notion (Weskamp et al., 2020). However, our data indicate that MP18 isoforms exhibit partially FL-like splicing activity, such as the suppression of cryptic exon inclusion and the promotion of PDP1 exon skipping by MP18

(127) (Revised Fig. 2 A). This effect is likely due to heteromeric complex formation between MP18 and FL, with previous reports suggesting that FL-FL interactions are primarily mediated by the NTD, rather than the CTD (described in Revised Fig. 2 H and Discussion, at Line 434-438 as follows).

“MP20 shares functional domains from the NTD to RRM2 with FL but lacks the C-terminal region, which is essential for interactions with other splicing regulatory proteins. For instance, TDP-43 splicing activity in CFTR exon 9 skipping relies on protein-protein interactions between its C-terminal residues 321-366 and hnRNP A2 and A1 (Buratti et al., 2005; D'Ambrogio et al., 2009). Given that MP20 lacks this region, it is likely unable to regulate splicing independently. “

Line 524: is there evidence for TDP43 loss of function (i.e. cryptic splicing) in ALS-FUS?

Thank you for your comment. We have not found evidence suggesting TDP-43 loss of function in ALS-FUS. ALS-FUS typically has an earlier onset age, with an average of 35.2-46.3 years, which is much younger than the average age of ALS onset (around 60 years) (Corrado et al., 2010; Xiao et al., 2024). The pathogenesis of ALS-FUS involves not only FUS loss of function but also the gain of toxic function due to FUS cytoplasmic inclusion (An et al., 2019; Scekcic-Zahirovic et al., 2016). Therefore, we propose that the earlier onset of ALS-FUS may result in disease progression driven primarily by FUS abnormalities, potentially preceding the manifestation of TDP-43 pathology.

Consequently, we have removed the expression suggesting an involvement of TDP-43 dysfunction in ALS-FUS in the revised manuscript to avoid potential misinterpretation. Instead, we have focused on discussing the roles of nuclear and cytoplasmic FUS in the regulation of TDP-43 (Line 490-500 as follows).

“In ALS, cytoplasmic FUS leakage is linked to loss of nuclear function, while its aggregation suggests gain of toxic function (An et al., 2019; Scekcic-Zahirovic et al., 2016). Our data also showed that the P525L mutant FUS exhibited reduced nuclear localization and formed aggregates in the cytoplasm (Fig. 9 E and G). Loss of nuclear FUS likely impairs its ability to regulate hnRNP K activity via TARDBP RNA interaction, which could explain the failure of P525L FUS to suppress nuclear hnRNP K expression (Fig. 9 E).

As for cytoplasmic FUS, while WT FUS forms stress granules (SGs) containing GC-rich RNA under stress, mutant FUS SGs are enriched with unstructured AU-rich RNA (Mariani et al., 2024). Although we observed no significant difference in translational inhibition of MP20 between WT and P525L mutant FUS (Fig. S5 E-G), the altered RNA-binding

properties of the mutant may influence FUS-mediated translational control.

Our finding suggest that FUS play a multifaceted role in regulating TDP-43 providing insights into its direct effects on TDP-43 and the complex mechanism involving other RBPs (Fig. 10, A-C).”

In addition, we present a proposed model for the mechanism by which hnRNP K induces MP20 splicing and the involvement of FUS in this process, which is illustrated only in the response letter.

Supporting Fig. (Response Letter only)

Finally, during the review process, a study was published reporting that MP18 is generated through TDP-43 autoregulation and may exert a potent dominant-negative effect (Dykstra et al., 2025). This finding supports our conclusion that certain TDP-43 splice isoforms possess dominant-negative activity. Moreover, our data demonstrate that MP20 exhibits a more pronounced dominant-negative effect than MP18, highlighting the significance of our study

in advancing the understanding of TDP-43 functional regulation.

Accordingly, we have added this reference and included a corresponding sentence at Line 452–457.

Reviewer #2 (Comments to the Authors (Required)):

Retention of cryptic exons in neuronal genes has been identified as a common finding in ALS and FTD with TDP-43 proteinopathy. In this manuscript, Hasegawa-Ogawa and colleagues investigated the role of RNA binding proteins (RBPs) in the alternative splicing of TDP-43. The investigators first showed that an alternatively spliced TDP-43 isoform, TDP-43 marginal peptide 20 (or MP20) exhibited dominant negative effects on the retention of cryptic exons in TDP-43 target genes, GPSM2, ATG4B, PDP1, and BCL2L11. Next, the investigators used a candidate approach to show that RBPs HnRNPK and HnRNPA1 promote or suppress the splicing of TDP-43 M20 isoform, respectively. Finally, the authors showed that wild type FUS suppresses HnRNPK-mediated activation of M20 isoform splicing.

Overall, this is an interesting study that reveals new insights into the dominant negative role of TDP-43 spliced isoform M20 and how different RBPs, such as HnRNPK and HnRNPA1, can regulate the splicing of M20. However, there are several significant issues that need to be addressed to support the conclusions of this study. From a technical standpoint, the study requires more data to strengthen its conclusion. Without these additional data, this study is still quite preliminary and the conclusion premature. From a conceptual standpoint, the entire study was conducted using overexpression in heterologous cell lines. Hence, the physiological implications of the majority of data in this study remains unclear. Below are specific comments on how to improve this manuscript:

Response to reviewer's comments:

We sincerely thank Reviewer #2 for your constructive feedback on our manuscript. We appreciate the recognition of our findings and the helpful suggestions to address both technical points and overall interpretation. Below, we provide our responses to the specific concerns raised and outline the additional data and explanations added to improve the manuscript.

1. The entire study was conducted in HeLaS3 or HEK293 cells using overexpression of genetically engineered constructs. As such, the physiological relevance of the TDP-43 spliced isoform M20 and its reported upstream regulatory mechanisms by HnRNPK and HnRNPA1 remain unclear. For instance, the authors can provide data, using RT-qPCR combined with

direct sequencing of PCR products to demonstrate the relative abundance of TDP-43 M20 (and M18, M13) isoforms in the adult mouse brain. Moreover, they can use their M20-specific antibody to detect the presence of M20 protein using immunostaining and western blots.

Thank you for your valuable suggestions. To address the physiological relevance of the spliced isoform MP20, we examined the presence of MP20 (as well as MP18 and MP13) in the brain cortex and spinal cord of adult mice.

To compare TDP-43 splicing patterns between humans and mice, we performed RT-PCR on human cell lines (HEK293T and neuroblastoma-derived SH-SY5Y) as well as mouse brain cortex and spinal cord. Interestingly, the splicing pattern of TDP-43 differed significantly between the mouse CNS and human cells (Revised Fig. 4 E).

Revised Fig. 4 E

In mouse tissues, MP18 splicing was dominant, while in human cells, MP20+ splicing was observed not only in HEK293T and SH-SY5Y but also in iPSC-derived neural stem cells and postmitotic neurons (below). Sequencing of PCR products confirmed the presence of MP13, MP20s, and MP18s in these human cells.

Supporting Fig. (Response Letter only)

iNS: Human iPSC-derived neural stem cells
iNeurons: Human iPSC-derived postmitotic neurons

In mouse tissues, we detected MP13, MP20 (118), and MP20 (127) by sequencing MP20+ bands. Notably, MP18 splicing in mice included MP18 (118), MP18 (127), and MP18 (145),

the last of which was not observed in human cells (summary provided below). These findings are consistent with previous results by D'Alton et al. (2015), which identified MP20 (127), MP18 (118), MP18 (127), and MP18 (145), but not MP13, using 3' RACE. These differences suggest the presence of a human-specific splicing regulatory mechanism.

Supporting Fig. (Response Letter only)

Tdp-43 splicing isoforms in Mice

To confirm protein expression and localization of MP20, we performed Western blotting (WB) and immunofluorescent staining (IF) on mouse brain cortex using the MP20-specific antibody (Revised Fig. 4, F and G). WB bands corresponding to MP20 were detected in extracts from both the brain cortex and spinal cord (Revised Fig. 4 F).

Revised Fig. 4 F

In IF experiments, MP20 signals were observed predominantly in the nuclei of MAP2-positive neurons (arrowhead), with weaker signals in MAP2-negative cells (arrow). The localization pattern of MP20, which was primarily nuclear with a slight presence in the cytoplasm, was consistent with the endogenous MP20 localization observed in HeLaS3 cells (Revised Fig. 4 G).

Revised Fig. 4 G

Although MP20 RNA expression levels appear low in mice, the protein is detectable. This is consistent with a previous mass spectrometry study (D'Alton et al., 2015), which reported the presence of MP20 peptide fragments in the mouse cortex. These results suggest that while the splicing and expression profiles of MP20 differ between humans and mice, the protein is present and may play a functional role in both species.

Accordingly, we have supplemented Revised Fig. 4E-G with the additional data and revised text at Line 221-233, 506-507 as follows.

Line 221-233

“MP20 exhibits physiological expression in neurons in vivo

To address the physiological relevance of the spliced isoform MP20, we examined the presence of MP20 in the brain cortex and spinal cord of adult mice. RT-PCR analysis revealed significant differences in TDP-43 splicing patterns between the mouse CNS and human cells, with MP18 splicing being dominant in mouse tissues, while MP20+ splicing was observed in human-derived cells (non-neural HEK293T and neural SH-SY5Y cell lines) (Fig. 4 E). This species-specific difference highlights the unique splicing regulation of TDP-43 in humans. WB and immunofluorescent staining (IF) using the MP20-specific antibody confirmed the expression and localization of MP20 in mouse CNS (Fig. 4, F and G). MP20 signals were

predominantly nuclear in MAP2-positive neurons (arrowhead), with weaker signals detected in MAP2-negative cells (arrow), possible glial or other non-neuronal populations (Fig. 4 G). The localization pattern of MP20, primarily nuclear with slight cytoplasmic presence, was consistent with the endogenous MP20 localization observed in HeLaS3 cells. These findings suggest that MP20 is expressed in neurons in vivo and may play a physiological role, despite its RNA levels remaining low in mice."

Line 506-507

"Interestingly, while MP18 predominates in mice (Weskamp et al., 2020) (Fig. 4 E), humans exhibit additional short isoforms, reflecting an evolutionarily complex splicing machinery."

2. This study relied heavily on the use of TARDBP mini-gene and RT-qPCR to identify the presence of different TDP-43 spliced isoforms and characterize how different RBPs regulate TDP-43 splicing. However, the RT-qPCR approach did not have the "resolution" to distinguish TDP-43 M13 isoform (hence M20 and M13 were collectively grouped as "M20+"). To circumvent this issue, the authors should perform sequencing on the PCR products to figure out the relative abundance of each isoforms.

Thank you for your helpful suggestions. In the initial version of Figure 2e, we had already provided the relative abundance of each isoform within the MP20+ band for HEK293T cells based on sequencing of PCR products. However, we realize that this point may not have been sufficiently emphasized in our description, which may have led to some confusion. In HEK293T cells, the MP20+ band contained 68.4% MP13, 10.5% MP20 (118), and 21.1% MP20 (127) (Initial Figure 2e and Revised Fig. 3 D).

Furthermore, we conducted similar analyses in human iPSC-derived neural stem cells and postmitotic neurons, revealing that the isoform ratios remained consistent (data shown below). These results indicate that the relative abundance of these isoforms is tightly regulated in human cells.

Supporting Fig. (Response Letter only)

Relative abundance of each isoform in MP20+

Additionally, overexpression of hnRNP K significantly altered the isoform ratio, resulting in a band composed only of MP20 (127) (Revised Fig. 3 D). The previous graph in Initial Figure 2e reflected MP20+ mRNA levels on the left Y-axis, which may have also caused some misunderstanding. Revised Fig. 3 D now focuses solely on the relative abundance of each isoform within the MP20+ band, making this data clearer.

Revised Fig. 3 D

Accordingly, we have replaced data with Revised Fig. 3 D and revised text at Line 199.

3. Many key panels (including RT-qPCR, western blots) lack quantifications or proper control experiments to support the conclusion. Although the authors mentioned they used RT-qPCR, reading from Materials and Methods on this approach (page 22, lines 560-567), this reviewer still could not figure out how this approach is considered to be quantitative. In this context, it will be more convincing to quantify each PCR product (e.g., Figure 1d, 2b, 2d, 7f, and 8b) and show the consistency of these RT-PCR results.

Thank you for your important suggestions, and we apologize for any confusion caused by our initial explanation. For quantification of mRNA levels of TDP-FL and MP20 (127) isoforms, we designed specific TaqMan MGB probes and primers targeting unique sequences

for each isoform. The probe for TDP-FL was designed to target the exon 6 region absent in MP isoforms, while the probe for MP20 (127) targeted the junction between exon 6 (127 nt) and the spliced exon 7 using the downstream acceptor site. These probe sets were optimized for specificity through Thermo Fisher Scientific's Custom Synthesis Service, which uses proprietary technology to select the sequences with the highest specificity for the target regions. To clarify this approach, we have added a schematic diagram in Revised Fig. 3 A to illustrate the probe design locations in the *TARDBP* gene (revised text at Line 199-201).

Revised Fig. 3 A

For mRNA quantifications of *hnRNP K*, *hnRNP A1*, and *ACTB*, we used pre-designed TaqMan Gene Expression Assays (Thermo Fisher Scientific), which are optimized for high specificity and sensitivity in detecting target mRNA sequences. For *GPSM2* and *ATG4B*, we utilized the SYBR Green system, and confirmed that mRNA levels of *GPSM2* and *ATG4B* cryptic exons significantly increased upon TDP-FL knockdown, consistent with previous reports (Revised Fig. S1 D) (Ling et al., 2015).

Revised Fig. S1 D

All RT-qPCR data in this study were normalized to *ACTB* as a housekeeping gene, as indicated on the vertical axis of each quantification graph in the revised version. We have also revised the text in the Methods section (Line 577-591) to clarify our RT-qPCR protocol, detailing steps taken to ensure quantitative accuracy, including normalization to *ACTB* as a housekeeping gene, the use of highly specific primers and probes, and careful validation of all reagents.

Additional Information for Revised Figures:

1. Initial Figure 1d and 1e:

In Initial Figure 1d and 1e, mRNA levels of *GPSM2* and *ATG4B* cryptic exons were quantified using RT-qPCR, and exon inclusion and exclusion ratios for *PDP1* and *BCL2L11* were quantified using RT-PCR with an increased sample size (n=3). These data are now presented as Revised Fig. 2, A-C and Revised Fig. 2, D-G.

Revised Fig. 2 A-C

Revised Fig. 2 D-G

2. Initial Figure 2b

The goal of Initial Figure 2b was not to quantify each TDP-43 isoform's absolute mRNA levels but to identify RNA-binding proteins (RBPs) that significantly shift the splicing ratio of shortened TDP-43 isoforms. Therefore, RT-PCR analysis was used to assess changes in the splicing ratio of shortened TDP-43 isoforms with an increased sample size (n=3) in Revised Fig. 3 C. TDP-FL (1240bp) was excluded from comparisons due to saturation in some lanes and the significant difference in amplicon size between TDP-FL and the shortened isoforms, which affects PCR efficiency. To address this, we have annotated the bands corresponding to TDP-FL with "(saturated)" in Revised Fig. 3 B.

Furthermore, the isoform initially labeled as "shorter isoform 1" in Initial Figure 2b is now referred to as "short MP13 (sMP13)" in the revised version. This change is reflected in Revised Fig. 3 B, where the ratio of MP18, MP13/20, and sMP13 is shown in the graph of Revised Fig. 3 C. Additionally, the schematic diagram of the "shorter isoform 1" from Initial Figure S1d has been updated and now depicts sMP13 in Revised Fig. S1 F.

Revised Fig. 3 B

Revised Fig. 3 C

Revised Fig. S1 F

3. Initial Figure 2d:

The data in Initial Figure 2d focused on the splicing patterns of TDP-43 mini-gene constructs, while Initial Figure 2b (Revised Fig. 3 C) addressed the splicing patterns of endogenous TDP-43 isoforms. Although these datasets are distinct, the conclusions they support are essentially the same. To avoid redundancy and improve clarity, we have removed Initial Figure 2d from the revised version.

4. Initial Figure 7f:

For Initial Figure 7f, we have added RT-qPCR quantification data for TDP-FL and MP20 (127) mRNA levels to strengthen the conclusions (Revised Fig. 7 H).

Revised Fig. 7 H

5. Initial Figure 8b and 8f:

Similarly, RT-qPCR quantification data for TDP-FL and MP20 (127) mRNA levels have been added to provide more comprehensive support for our conclusions (Revised Fig. 8, C and F)

Revised Fig. 8 C

Revised Fig. 8 F

4. Figure 4: Panel A compared the differences in the role of HnRNPK, HnRNPE1, HnRNPE2, and HnRNPA1. However, all subsequent studies in panels b, c and e will need to include results from E1, E2 and A1 to provide the effectiveness of the PCR reaction.

Thank you for your valuable suggestions. To allow a side-by-side comparison of hnRNP K, hnRNP E1, hnRNP E2, and hnRNP A1, we additionally performed ICC and WB analysis in cells overexpressing FLAG-tagged hnRNPs. In the ICC, overexpression of hnRNP K, E1, and E2 significantly increased nuclear MP20 levels compared to control (FLAG-negative cells), while hnRNP A1 did not (Revised Fig. 5 B).

Revised Fig. 5 B

Consistent with the ICC results, WB analysis showed that hnRNP K and E1 significantly increased both SVs-endo and MP20-endo levels (Revised Fig. 5 D). While hnRNP E2 showed a tendency to increase MP20-endo, this was not statistically significant. HnRNP A1 did not affect MP20 expression in the ICC experiments.

More details on the hnRNP A1 overexpression experiments can be found below in the Response to Reviewer #1 Figure 6.

Revised Fig. 5 D

Note that MP20-endo and SVs-endo induction capacities were entirely higher for hnRNP K. Therefore, only the results for hnRNP K are shown in Revised Fig. 5 E and 5 F.

Accordingly, we have shown in Revised Fig. 5 B and 5 D with the additional data and revised text Line at 244-248, and 250-254 as follows.

Line at 244-248

“Quantification of nuclear MP20 fluorescence intensity in immunocytochemistry showed a 1.4-fold increase in the FLAG-hnRNP K-positive cells (arrowheads) compared to FLAG-negative cells (arrow), suggesting that hnRNP K enhances nuclear MP20 expression (Fig. 5 B). A similar increase in nuclear MP20 was observed in FLAG-hnRNP E1 and FLAG-hnRNP

E2-positive cells, while FLAG-hnRNP A1-positive cells showed no detectable change in MP20 levels (Fig. 5 B)."

Line at 250-254

"WB analysis in HEK293T cells further confirmed that overexpression of hnRNP K and hnRNP E1 increased MP20-endo and SVs-endo levels (Fig. 5 D). A similar trend was observed with hnRNP E2, although the increase was not statistically significant. In contrast, overexpression of hnRNP A1 led to a reduction in both SVs-endo and MP20, a result that did not align with the immunocytochemistry data (Fig. 5 D)."

5. Figure 5: The images in panel 5b are quite confusing. Even after reading the figure legend for Figure 5b, it is still unclear what these images were intended for.

Thank you for your valuable suggestion, and we apologize for the confusion caused by the figure layout and the unclear labeling. To address this, we have added a simplified schematic to clarify that FLAG-hnRNP K is overexpressed in the cells. Additionally, we have explicitly labeled each panel to ensure that their purpose is clear (Revised Fig. 6 B, below).

Revised Fig. 6 B

6. The data in Supplementary Figures 5 and 6 appeared to be incompatible with the central theme of the entire manuscript. As such, the inclusion of these two supplementary figures came across as out of place and very strange.

Thank you for your valuable feedback. We apologize for any confusion caused by the description of the experimental procedures and results, and understand your concern that

Supplementary Figure 5 and 6 appeared to be inconsistent with the central theme of our manuscript.

In response, we have removed Supplemental Figure 5e-f as it falls outside the main scope of this study. Supplemental Figure 5a-d and Supplemental Figure 6a-b have been combined into the Revised Fig. S 4, and we have revised the figure legend and related text (Line 356-386). We have made efforts to improve clarity and ensure the figures are more easily understood.

We included Revised Fig. S4 to better explain the following key observations: Initially, we hypothesized that hnRNP K promotes MP20 splicing by directly interacting with *TARDBP* RNA. To identify the responsible regions of *TARDBP* exon 6 targeted by hnRNP K using the *TARDBP* mini-gene system, but we were unable to identify such regions (Fig. S4 A-D).

However, during this process, we unexpectedly found that the 532-584 region of exon 6 is essential for the *TARDBP* splicing (Fig. S4, E and F). Deletion of this region severely disrupts splicing, likely due to excessive sensitivity to hnRNP K, suggesting that this region serves as a regulatory site for modulating hnRNP K sensitivity through other RBPs. These findings provide critical context for subsequent analyses investigating FUS as a candidate RBP involved in this regulatory mechanism.

7. The findings of wild type FUS as an upstream regulator of HnRNPK in Figure 9 seem to contradict with those reported in Figure 2, which showed no obvious effects of wild type FUS in MP20 splicing (though these data were not quantified in any meaningful way). This inconsistency raises concerns regarding the validity of the overexpression system used in this study.

Thank you for your valuable comments. We appreciate your concern regarding the apparent discrepancy between the data shown in Initial Figure 2b and Figure 9.

In the RT-PCR data presented in Initial Figure 2b (Revised Fig. 3 B), the MP20+ band includes both MP13 and MP20s isoforms, making it difficult to precisely quantify the MP20 (127) alone. As you may appreciate, this limitation in distinguishing between overlapping isoforms likely caused the unclear effects of wild-type FUS on MP20 splicing in this assay. In contrast, the data in Initial Figure 9c (Revised Fig. 9 C) obtained using RT-qPCR with isoform-specific TaqMan MGB probe, which provides a more specific and accurate quantification of MP20 (127) mRNA levels. The approach revealed a 1.8-fold increase in MP20 (127) mRNA upon FUS overexpression. The sensitivity and precision of RT-PCR allowed us to detect changes in MP20 (127) expression that were not apparent in the RT-PCR analysis of Initial Figure 2b.

This discrepancy likely arises from the inherent limitations of the RT-PCR method, rather than an issue with the overexpression system. Therefore, we hope this clarification addresses

your concerns about the validity of our experimental approach.

8. Finally, there are many areas in the manuscript where the data presentation and how they support the conclusion can be improved. For instance, the descriptions of TDP-43 RRM and RRM mutants in Page 5, lines 124-130, are very confusing. Moreover, the excessive use of adjectives and ("surprisingly", "intriguingly", "strongly", "importantly", "strikingly", etc) should be avoided.

We apologize for any confusion caused by the presentation of the data and their support for the conclusions in our manuscript. We have carefully reviewed the wording and revised it to avoid any exaggerated expressions as much as possible.

Finally, during the review process, a study was published reporting that MP18 is generated through TDP-43 autoregulation and may exert a potent dominant-negative effect (Dykstra et al., 2025). This finding supports our conclusion that certain TDP-43 splice isoforms possess dominant-negative activity. Moreover, our data demonstrate that MP20 exhibits a more pronounced dominant-negative effect than MP18, highlighting the significance of our study in advancing the understanding of TDP-43 functional regulation.

Accordingly, we have added this reference and included a corresponding sentence at Line 452–457.

- Afroz, T., E.M. Hock, P. Ernst, C. Foglieni, M. Jambeau, L.A.B. Gilhespy, F. Laferriere, Z. Maniecka, A. Plückthun, P. Mittl, P. Paganetti, F.H.T. Allain, and M. Polymenidou. 2017. Functional and dynamic polymerization of the ALS-linked protein TDP-43 antagonizes its pathologic aggregation. *Nat Commun.* 8:45.
- An, H., L. Skelt, A. Notaro, J.R. Highley, A.H. Fox, V. La Bella, V.L. Buchman, and T.A. Shelkovernikova. 2019. ALS-linked FUS mutations confer loss and gain of function in the nucleus by promoting excessive formation of dysfunctional paraspeckles. *Acta Neuropathol Commun.* 7:7.
- Avendaño-Vázquez, S.E., A. Dhir, S. Bembich, E. Buratti, N. Proudfoot, and F.E. Baralle. 2012. Autoregulation of TDP-43 mRNA levels involves interplay between transcription, splicing, and alternative polyA site selection. *Genes Dev.* 26:1679-1684.
- Ayala, Y.M., L. De Conti, S.E. Avendaño-Vázquez, A. Dhir, M. Romano, A. D'Ambrogio, J. Tollervey, J. Ule, M. Baralle, E. Buratti, and F.E. Baralle. 2011. TDP-43 regulates its mRNA levels through a negative feedback loop. *Embo j.* 30:277-288.
- Buratti, E., A. Brindisi, M. Giombi, S. Tisminetzky, Y.M. Ayala, and F.E. Baralle. 2005. TDP-43 binds heterogeneous nuclear ribonucleoprotein A/B through its C-terminal tail: an

- important region for the inhibition of cystic fibrosis transmembrane conductance regulator exon 9 splicing. *J Biol Chem.* 280:37572-37584.
- Corrado, L., R. Del Bo, B. Castellotti, A. Ratti, C. Cereda, S. Penco, G. Sorarù, Y. Carlomagno, S. Ghezzi, V. Pensato, C. Colombrita, S. Gagliardi, L. Cozzi, V. Orsetti, M. Mancuso, G. Siciliano, L. Mazzini, G.P. Comi, C. Gellera, M. Ceroni, S. D'Alfonso, and V. Silani. 2010. Mutations of FUS gene in sporadic amyotrophic lateral sclerosis. *J Med Genet.* 47:190-194.
- D'Ambrogio, A., E. Buratti, C. Stuani, C. Guarnaccia, M. Romano, Y.M. Ayala, and F.E. Baralle. 2009. Functional mapping of the interaction between TDP-43 and hnRNP A2 in vivo. *Nucleic Acids Res.* 37:4116-4126.
- Dykstra, M.M., K. Weskamp, N.B. Gómez, J. Waksmaeki, E. Tank, M.R. Glineburg, A. Snyder, E. Pinarbasi, M. Bekier, X. Li, M.R. Miller, J. Bai, S. Shahzad, N. Nedumaran, C. Wieland, C. Stewart, S. Willey, N. Grotewold, J. McBride, J.J. Moran, A.V. Suryakumar, M. Lucas, P.M. Tessier, M. Ward, P.K. Todd, and S.J. Barmada. 2025. TDP43 autoregulation gives rise to dominant negative isoforms that are tightly controlled by transcriptional and post-translational mechanisms. *Cell Rep.* 44:115113.
- Jiang, L.L., W. Xue, J.Y. Hong, J.T. Zhang, M.J. Li, S.N. Yu, J.H. He, and H.Y. Hu. 2017. The N-terminal dimerization is required for TDP-43 splicing activity. *Sci Rep.* 7:6196.
- Kamelgarn, M., J. Chen, L. Kuang, H. Jin, E.J. Kasarskis, and H. Zhu. 2018. ALS mutations of FUS suppress protein translation and disrupt the regulation of nonsense-mediated decay. *Proc Natl Acad Sci U S A.* 115:E11904-e11913.
- Lagier-Tourenne, C., M. Polymenidou, K.R. Hutt, A.Q. Vu, M. Baughn, S.C. Huelga, K.M. Clutario, S.C. Ling, T.Y. Liang, C. Mazur, E. Wancewicz, A.S. Kim, A. Watt, S. Freier, G.G. Hicks, J.P. Donohue, L. Shiue, C.F. Bennett, J. Ravits, D.W. Cleveland, and G.W. Yeo. 2012. Divergent roles of ALS-linked proteins FUS/TLS and TDP-43 intersect in processing long pre-mRNAs. *Nat Neurosci.* 15:1488-1497.
- Ling, J.P., O. Pletnikova, J.C. Troncoso, and P.C. Wong. 2015. TDP-43 repression of nonconserved cryptic exons is compromised in ALS-FTD. *Science.* 349:650-655.
- Mompeán, M., V. Romano, D. Pantoja-Uceda, C. Stuani, F.E. Baralle, E. Buratti, and D.V. Laurents. 2017. Point mutations in the N-terminal domain of transactive response DNA-binding protein 43 kDa (TDP-43) compromise its stability, dimerization, and functions. *J Biol Chem.* 292:11992-12006.
- Scekic-Zahirovic, J., O. Sindscheid, H. El Oussini, M. Jambeau, Y. Sun, S. Mersmann, M. Wagner, S. Dieterlé, J. Sinniger, S. Dirrig-Grosch, K. Drenner, M.C. Birling, J. Qiu, Y. Zhou, H. Li, X.D. Fu, C. Rouaux, T. Shelkownikova, A. Witting, A.C. Ludolph, F. Kiefer, E. Storkebaum, C. Lagier-Tourenne, and L. Dupuis. 2016. Toxic gain of function from

mutant FUS protein is crucial to trigger cell autonomous motor neuron loss. *Embo j.* 35:1077-1097.

Weskamp, K., E.M. Tank, R. Miguez, J.P. McBride, N.B. Gómez, M. White, Z. Lin, C.M. Gonzalez, A. Serio, J. Sreedharan, and S.J. Barmada. 2020. Shortened TDP43 isoforms upregulated by neuronal hyperactivity drive TDP43 pathology in ALS. *J Clin Invest.* 130:1139-1155.

Xiao, X., M. Li, Z. Ye, X. He, J. Wei, and Y. Zha. 2024. FUS gene mutation in amyotrophic lateral sclerosis: a new case report and systematic review. *Amyotroph Lateral Scler Frontotemporal Degener.* 25:1-15.

Zhang, Y.J., T. Caulfield, Y.F. Xu, T.F. Gendron, J. Hubbard, C. Stetler, H. Sasaguri, E.C. Whitelaw, S. Cai, W.C. Lee, and L. Petrucelli. 2013. The dual functions of the extreme N-terminus of TDP-43 in regulating its biological activity and inclusion formation. *Hum Mol Genet.* 22:3112-3122.

June 12, 2025

RE: JCB Manuscript #202406097R

Hirotaoka Okano
Jikei University School of Medicine

Dear Prof. Okano,

Thank you for submitting your revised manuscript entitled "Dominant-negative isoform of TDP-43 is regulated by ALS-linked RNA-binding proteins." We would be happy to publish your paper in JCB pending final revisions necessary to meet our formatting guidelines (see details below).

A. MANUSCRIPT ORGANIZATION AND FORMATTING:

1) Text limits: Character count for Articles and Tools is < 40,000, not including spaces. Count includes title page, abstract, introduction, results, discussion, and acknowledgments. Count does not include materials and methods, figure legends, references, tables, or supplemental legends.

2) Figure formatting: Articles may have up to 10 main text figures. Scale bars must be present on all microscopy images, including inset magnifications. Molecular weight or nucleic acid size markers must be included on all gel electrophoresis. Please add size markers to gels in Figures 2A/E, 3B/F, 4E, 7B/C/G, 8B/E, S3B/D/F, S4B/D/F, & S5C.

Also, please avoid pairing red and green for images and graphs to ensure legibility for color-blind readers. If red and green are paired for images, please ensure that the particular red and green hues used in micrographs are distinctive with any of the colorblind types. If not, please modify colors accordingly or provide separate images of the individual channels.

3) Statistical analysis: Error bars on graphic representations of numerical data must be clearly described in the figure legend. The number of independent data points (n) represented in a graph must be indicated in the legend. Please indicate whether 'n' refers to technical or biological replicates (i.e. number of analyzed cells, samples or animals, number of independent experiments). If independent experiments with multiple biological replicates have been performed, we recommend using distribution-reproducibility SuperPlots (please see Lord et al., JCB 2020) to better display the distribution of the entire dataset, and report statistics (such as means, error bars, and P values) that address the reproducibility of the findings.

Statistical methods should be explained in full in the materials and methods. For figures presenting pooled data the statistical measure should be defined in the figure legends. Please also be sure to indicate the statistical tests used in each of your experiments (both in the figure legend itself and in a separate methods section) as well as the parameters of the test (for example, if you ran a t-test, please indicate if it was one- or two-sided, etc.). Also, if you used parametric tests, please indicate if the data distribution was tested for normality (and if so, how). If not, you must state something to the effect that "Data distribution was assumed to be normal but this was not formally tested."

4) Abstract: We suggest adding the word "expression" between 'proper' and 'levels' in the second sentence to make this clearer.

5) Materials and methods: Should be comprehensive and not simply reference a previous publication for details on how an experiment was performed. Please provide full descriptions (at least in brief) in the text for readers who may not have access to referenced manuscripts. The text should not refer to methods "...as previously described." Please also indicate the acquisition and quantification methods for immunoblotting/western blots.

6) For all cell lines, vectors, strains, constructs/cDNAs, etc. - all genetic material: please include database / vendor ID (e.g. Addgene, ATCC, etc.) or if unavailable, please briefly describe their basic genetic features, even if described in other published work or gifted to you by other investigators (and provide references where appropriate). Please be sure to provide the sequences for all of your oligos: primers, si/shRNA, RNAi, gRNAs, etc. in the materials and methods. You must also indicate in the methods the source, species, and catalog numbers/vendor identifiers (where appropriate) for all of your antibodies, including secondary. If antibodies are not commercial, please add a reference citation if possible.

7) Microscope image acquisition: The following information must be provided about the acquisition and processing of images:

- a. Make and model of microscope
- b. Type, magnification, and numerical aperture of the objective lenses
- c. Temperature
- d. Imaging medium
- e. Fluorochromes
- f. Camera make and model
- g. Acquisition software
- h. Any software used for image processing subsequent to data acquisition. Please include details and types of operations involved (e.g., type of deconvolution, 3D reconstitutions, surface or volume rendering, gamma adjustments, etc.).

8) References: There is no limit to the number of references cited in a manuscript. References should be cited parenthetically in the text by author and year of publication. Abbreviate the names of journals according to PubMed.

9) Supplemental materials: Articles may have up to 5 supplemental figures and 10 videos. Please also note that tables, like figures, should be provided as individual, editable files. A summary of all supplemental material should appear at the end of the Materials and methods section. Please include one brief sentence per item.

10) eTOC summary: A ~40-50 word summary that describes the context and significance of the findings for a general readership should be included on the title page. The statement should be written in the present tense and refer to the work in the third person. It should begin with "First author name(s) et al..." to match our preferred style.

11) Conflict of interest statement: JCB requires inclusion of a statement in the acknowledgements regarding competing financial interests. If no competing financial interests exist, please include the following statement: "The authors declare no competing financial interests." If competing interests are declared, please follow your statement of these competing interests with the following statement: "The authors declare no further competing financial interests."

12) A separate author contribution section is required following the Acknowledgments in all research manuscripts. All authors should be mentioned and designated by their first and middle initials and full surnames. We encourage use of the CRediT nomenclature (<https://casrai.org/credit/>).

13) ORCID IDs: ORCID IDs are unique identifiers allowing researchers to create a record of their various scholarly contributions in a single place. Please note that ORCID IDs are required for all authors. At resubmission of your final files, please be sure to provide your ORCID ID and those of all co-authors.

14) JCB requires authors to submit Source Data used to generate figures containing gels and Western blots with all revised manuscripts. This Source Data consists of fully uncropped and unprocessed images for each gel/blot displayed in the main and supplemental figures. For assays performed using capillary electrophoresis and/or immunoassay-based detection, authors should instead provide the electropherogram graph(s) for each experiment, plotting fluorescence/chemiluminescence intensity vs. molecular weight/size. Since your paper includes cropped gel and/or blot images, please be sure to provide one Source Data file for each figure gels, blots, and/or capillary electrophoresis assays along with your revised manuscript files. File names for Source Data figures should be alphanumeric without any spaces or special characters (i.e., SourceDataF#, where F# refers to the associated main figure number or SourceDataFS# for those associated with Supplementary figures). For traditional gels and blots, the lanes of the gels/blots should be labeled as they are in the associated figure, the place where cropping was applied should be marked (with a box), and molecular weight/size standards should be labeled wherever possible. For capillary electrophoresis assays, each trace in the graph should be color-coded and labeled to indicate which protein, gene, or sample is being measured (please try to avoid red/green combinations to accommodate our color-blind readers).

Source Data files will be directly linked to specific figures in the published article. Source Data Figures should be provided as individual PDF files (one file per figure). Authors should endeavor to retain a minimum resolution of 300 dpi or pixels per inch. Please review our instructions for export from Photoshop, Illustrator, and PowerPoint here: <https://rupress.org/jcb/pages/submission-guidelines#revised>

15) Journal of Cell Biology now requires a data availability statement for all research article submissions. These statements will be published in the article directly above the Acknowledgments. The statement should address all data underlying the research presented in the manuscript. Please visit the JCB instructions for authors for guidelines and examples of statements at (<https://rupress.org/jcb/pages/editorial-policies#data-availability-statement>).

B. FINAL FILES:

Thank you for your attention to these final processing requirements. Please revise and format the manuscript and upload materials within 7 days. If you need an extension for whatever reason, please let us know and we can work with you to determine a suitable revision period.

Thank you for this interesting contribution, we look forward to publishing your paper in Journal of Cell Biology.

Sincerely,

Richard Youle, PhD
Monitoring Editor
Journal of Cell Biology

Dan Simon, PhD
Scientific Editor
Journal of Cell Biology

Reviewer #1 (Comments to the Authors (Required)):

The authors have comprehensively addressed reviewer concerns. The revised manuscript is significantly improved - I look forward to seeing it in print.

Reviewer #2 (Comments to the Authors (Required)):

This manuscript has been extensively revised to provide clarity about the role of RNA binding proteins, HNRNPK and FUS, in regulating the splicing of TDP-43. The authors provided a number of new data and extensively updated their figures and main text to address my previous concerns. The revised manuscript is much improved and therefore suitable for publication in JCB.